# DREAM: A Unified Framework for Drift-Corrected Federated Multi-Objective Learning

Yuan Zhou [1]  Yidan Ou [1]  Xinli Shi [1]

## Abstract

Federated Multi-Objective Learning (FMOL) enables collaborative training of conflicting objectives but faces a compounded challenge: the recursive coupling between intra-task client drift and inter-task aggregation bias. We propose DREAM, a unified framework that jointly corrects these two coupled error sources through drift-aware control variates and momentum-smoothed local updates. On the server side, DREAM formulates multi-objective aggregation as a regularized quadratic program parameterized by a task correction matrix, which provides a generalized formulation that can flexibly adapt to scalarization, prioritization, and gradient manipulation strategies. Theoretically, we establish a linear speedup convergence rate of $\mathcal{O}(1/\sqrt{NT})$ for non-convex objectives. We further provide theoretical guarantees for the conflict-avoidant direction distance. In the strongly convex setting, DREAM achieves convergence in weighted sub-optimality and admits a unified Lyapunov analysis showing linear convergence to a regularization-dependent neighborhood. Numerical experiments on representative benchmarks validate the effectiveness of DREAM in multi-objective optimization.

## 1. Introduction

Modern machine learning confronts the fundamental challenge of optimizing multiple, often conflicting, objectives simultaneously. This tension is ubiquitous: Multi-Task Learning (MTL) requires a single model to master diverse tasks (Sener & Koltun, 2018); responsible AI must balance predictive accuracy with demographic fairness (Mehrabi et al., 2021) or adversarial robustness (Madry et al., 2018); and

---

[1]The School of Cyber Science and Engineering, Southeast University, Nanjing, China. Correspondence to: Xinli Shi <xinli_shi@seu.edu.cn>.

*Proceedings of the $43^{rd}$ International Conference on Machine Learning*, Seoul, South Korea. PMLR 306, 2026. Copyright 2026 by the author(s).

safe reinforcement learning must maximize rewards under strict safety constraints (García & Fernández, 2015). Even state-of-the-art Large Language Models (LLMs) grapple with the inherent trade-offs between goals like helpfulness, harmlessness, and honesty (Bai et al., 2022). At the heart of these problems lies the need for principled compromises. **Multi-Objective Optimization** (MOO) provides the mathematical language to formalize this challenge, enabling the pursuit of solutions that optimally navigate the trade-offs among competing criteria.

**Gradient manipulation** methods have emerged as a powerful paradigm in MOO, aiming to resolve gradient conflicts algorithmically rather than relying on fixed weights (Chen et al., 2025; Zhou et al., 2022). Among these, a foundational approach is the **Multiple-Gradient Descent Algorithm (MGDA)** (Désidéri, 2012; Sener & Koltun, 2018). It dynamically finds a common descent direction at each iteration by seeking to maximize the improvement of the worst-performing task. The core philosophy of MGDA is mathematically formulated as a regularized optimization problem (Fliege et al., 2019):

$$\min_{d \in \mathbb{R}^p} \max_{s \in [S]:=\{1,\cdots,S\}} \langle \nabla f_s(x), d \rangle + \frac{1}{2}\|d\|^2. \quad (1)$$

The problem of finding the optimal direction $d$ in (1) is dual to a Quadratic Programming (QP) problem over a set of aggregation weights $\boldsymbol{\lambda}^*$. This QP, which forms the core of MGDA (Désidéri, 2012; Fliege et al., 2019), is given by:

$$\min_{\boldsymbol{\lambda}^* \in \boldsymbol{\omega}} \left\| \sum_{s \in [S]} \lambda_s \nabla f_s(x) \right\|^2, \quad (2)$$

where $\boldsymbol{\omega} := \{\boldsymbol{\lambda} \in \mathbb{R}^S \mid \lambda_s \geq 0, \ \sum_{s \in [S]} \lambda_s = 1\}$ is the probability simplex. By solving for these weights dynamically, MGDA effectively resolves inter-task conflicts and finds a descent direction $d = \sum_{s \in [S]} \lambda_s^* \nabla f_s(x)$, then converges to a **Pareto-stationary point**, a state where no common descent direction exists. Detailed derivations can be found in foundational works such as (Désidéri, 2012; Fliege et al., 2019).

Despite its theoretical elegance, MGDA is often impractical for large-scale machine learning, as it requires computing exact, full-batch gradients. Consequently, this naturally

motivates the development of Stochastic MGDA (SMGDA) (Liu & Vicente, 2024), wherein the full gradients $\nabla f_s(x)$ are replaced by their unbiased stochastic estimators $g_s(x)$ in (2) to compute weights $\boldsymbol{\lambda}^{g,*} \in \boldsymbol{\omega}$. However, it introduces a fundamental challenge: **inherent non-zero bias in the aggregated gradient direction** (Fernando et al., 2023; Liu & Vicente, 2024; Zhang et al., 2025), i.e.,

$$\mathbb{E}\Big[\sum_{s\in[S]}\lambda_s^{g,*}g_s(x)\Big] \neq \sum_{s\in[S]}\lambda_s^*\nabla f_s(x).$$

This bias stems from the non-linear dependency of the optimal weights on the input gradients, which can destabilize training and even prevent convergence (Zhou et al., 2022; Fernando et al., 2023). To mitigate this, existing approaches typically resort to increasing batch sizes (Liu & Vicente, 2024), double sampling (Xiao et al., 2023; Chen et al., 2024a), or smoothing aggregation weights (Zhou et al., 2022; Xu et al., 2025). However, these strategies inevitably impose excessive computational burdens or compromise the capability for effective conflict avoidance.

While centralized MOO has seen progress, its reliance on pooling data into a single server is increasingly untenable due to prohibitive storage and computation costs. Federated Learning (FL) emerges as a natural paradigm, enabling collaborative training across decentralized clients (McMahan et al., 2017; Li et al., 2020b). Crucially for MOO, FL accesses diverse data distributions from multiple sources, potentially unveiling a more comprehensive and robust Pareto front that no single node could achieve in isolation. This gives rise to **Federated Multi-Objective Learning** (FMOL) (Hu et al., 2022; Yang et al., 2023; Xie et al., 2024; Askin et al., 2025). However, FMOL is not merely a sum of its parts; it introduces a compounded challenge involving two coupled sources of error. While the inter-task aggregation bias discussed previously exists even with unbiased gradient estimates, FMOL further complicates the optimization landscape with intra-task client drift. Due to data heterogeneity, the accumulated local updates ($\bar{\Delta}_s, \forall s \in [S]$) contain systematic shifts rather than mere zero-mean noise (Li et al., 2020a; Zhang et al., 2021), i.e., $\mathbb{E}[\bar{\Delta}_s] \neq \mathbb{E}[g_s(x)] = \nabla f_s(x)$. When these drifted updates serve as inputs to the non-linear aggregation mechanism, the resulting update direction significantly deviates from the true Pareto front, as illustrated in Figure 1. Unfortunately, existing frameworks (Yang et al., 2023; Askin et al., 2025) often overlook this pernicious coupling and rely on bounded heterogeneity assumption to simplify the analysis.

To tackle these challenges, we introduce DREAM (DRift-corrected and momEntum-smoothed federAted Multi-objective learning). It provides a generalized optimization framework for heterogeneous FMOL by focusing on the aggregated per-task updates $\{\bar{\Delta}_s\}_{s\in[S]}$, which are affected by client drift and then used as inputs to the non-linear

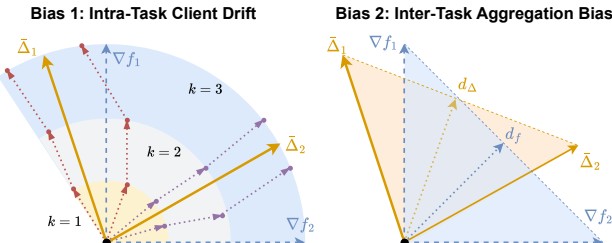

*Figure 1.* The dual-bias problem in FMOL: a cascade of errors. **LEFT**: Due to data heterogeneity, the aggregated per-task updates ($\bar{\Delta}_1$ and $\bar{\Delta}_2$), obtained from multiple clients after 3 local updates, drift away from the true global directions ($\nabla f_1$ and $\nabla f_2$). **RIGHT**: The subsequent non-linear aggregation of these drifted task updates introduces a second layer of bias, where the computed descent direction ($d_\Delta$) is a poor approximation of the ideal direction ($d_f$). **Toy Example**: take $\nabla f_1 = (1, 0.1)$, $\nabla f_2 = (-1, 0.1)$ with drifted updates $\bar{\Delta}_1 = (1, -0.05)$ and $\bar{\Delta}_2 = (-1, 0.2)$. Equal scalarization still gives $\bar{d} = 0.5\bar{\Delta}_1 + 0.5\bar{\Delta}_2 = (0, 0.075)$, satisfying $\langle \nabla f_s, \bar{d} \rangle > 0$ for both tasks, whereas MGDA on the same drifted inputs yields $d_\Delta \approx (0.01, 0.074)$ and $\langle \nabla f_2, d_\Delta \rangle < 0$, turning the update into ascent for Task 2.

weight-solving subproblem. The framework corrects and smooths these task-wise updates before server-side weight computation, thereby controlling the coupled error dynamics underlying FMOL. The main contributions of this work are threefold:

**Unified FMOL Error Analysis**. We formally characterize the dual-bias structure of FMOL, where intra-task client drift shifts the accumulated local updates away from the true directions and inter-task non-linear aggregation further biases the resulting update direction. We capture this phenomenon through coupled recursions between the gradient estimation error and the client drift, making explicit a feedback loop that prior FMOL analyses typically control through bounded heterogeneity assumptions. This analysis identifies the aggregated per-task updates as the key quantities linking local training, gradient estimation, and multi-objective weight computation.

**A Drift-Corrected and Momentum-Smoothed Framework**. Guided by the above error analysis, DREAM uses task-wise control variates to reduce the systematic shift caused by local training and a momentum term to smooth the aggregated per-task updates across rounds. On the server side, DREAM computes task weights by solving a $\rho$-regularized QP, whose strong convexity ensures the Lipschitz continuity of the weight mapping. This formulation supports both Exact-Update and a computationally efficient Step-Update strategy, and further incorporates a task correction matrix $\mathbf{A}$ to cover diverse optimization paradigms, including linear scalarization, weighted prioritization, and task-interaction modeling.

**Comprehensive Theoretical Guarantees**. For non-convex

objectives, we establish convergence to Pareto stationarity with a linear speedup rate of $\mathcal{O}(1/\sqrt{NT})$ under appropriate parameter choices. We further provide convergence guarantees for the time-averaged squared Conflict-Avoidant (CA) distance, achieving rates of $\mathcal{O}(1/\sqrt{NT})$ for Exact-Update and $\mathcal{O}(T^{-\frac{2}{5}})$ for Step-Update. In the strongly convex setting, we derive a convergence rate of $\tilde{\mathcal{O}}(1/\sqrt{T})$ for the weighted sub-optimality gap. To capture the system dynamics, we construct a Lyapunov function that couples the distance to Pareto optimality, gradient estimation error, and client drift, and augment it with a weight tracking term for the Step-Update analysis. The resulting bounds characterize the steady-state neighborhoods of the two update schemes, yielding an $\mathcal{O}(\rho^2)$ floor for Exact-Update and an $\mathcal{O}(\rho^2 + \rho^{-6})$ floor for Step-Update.

## 2. Related Work

While deterministic algorithms like MGDA, PCGrad (Yu et al., 2020), and CAGrad (Liu et al., 2021) are theoretically sound, applying them to stochastic settings introduces inherent aggregation bias due to the non-linear weight computation. Early attempts such as SMG (Liu & Vicente, 2024) addressed this by linearly increasing batch sizes, which is often impractical. Consequently, recent research has explored various strategies to mitigate this bias. One line of work, including CR-MOGM (Zhou et al., 2022) and PSMGD (Xu et al., 2025), applies smoothing directly to the aggregation weights, matching the optimal convergence rate of stochastic single-objective optimization. However, smoothing the output weights can cause weight stagnation and reduce the responsiveness of the algorithm to immediate gradient conflicts. Alternatively, MoCo (Fernando et al., 2023) applies momentum to the gradients via a tracking variable to estimate the true gradients before aggregation. Its improved variant MoCo+ (Fernando et al., 2024) adopts the recursive STORM variance reduction technique (Cutkosky & Orabona, 2019) to further enhance convergence rate and conflict avoidance. Other approaches such as MoDo (Chen et al., 2024a) and SDMGrad (Xiao et al., 2023) utilize a double-sampling strategy to construct unbiased estimates of gradient inner products. Their theoretical analyses remove the restrictive bounded function value assumption. Furthermore, SDMGrad introduces a direction-oriented regularized formulation to guide the optimization trajectory.

FMOL extends MOO to distributed settings. Early attempts like FedMGDA+ (Hu et al., 2022) combined FedAvg with MGDA, utilizing accumulated local updates as pseudo-gradients for the multi-objective solver. Subsequent works such as FedCMOO (Askin et al., 2025) and FSMGDA (Yang et al., 2023) introduce the bounded data heterogeneity assumption (i.e., $\|\nabla f_{s,i} - \nabla f_s\|^2 \leq \sigma_D^2$), primarily to bound the client drift and simplify the convergence analysis. While

Yang et al. (2023) provides convergence guarantees for the weighted sub-optimality gap under strong convexity, their analysis focuses on the objective value but lacks a rigorous characterization of the recursive interplay between client drift, gradient estimation errors, and the weight updates. Furthermore, in the federated setting where global gradients are approximated by aggregated updates, theoretical guarantees for the geometric property, specifically the CA direction distance (Chen et al., 2024a), remain largely absent. Consequently, despite the complexity of the analytical landscape, there remains a lack of work that can simultaneously provide guarantees for linear speedup, CA direction distance, and weighted sub-optimality convergence with a holistic Lyapunov analysis.

## 3. Algorithm Development

We address the MOO problem defined over a set of $S$ differentiable objective functions $\{f_s : \mathbb{R}^p \to \mathbb{R}\}_{s \in [S]}$. The goal is to find a single parameter vector that jointly minimizes all objectives. This is formally expressed as:

$$\min_{x \in \mathbb{R}^p} \mathbf{F}(x) := [f_1(x), \cdots, f_S(x)] . \tag{3}$$

We ground this problem in an FL setting with $N$ clients. The global objective for task $s \in [S]$ is an aggregation of local objectives from participating clients: $f_s(x) = \frac{1}{|\mathcal{R}_s|} \sum_{i \in \mathcal{R}_s} f_{s,i}(x)$, where $\mathcal{R}_s \subseteq [N]$ is the set of participating clients for task $s$, and $f_{s,i}$ is the corresponding local objective for client $i$. Conversely, each client $i$ may participate in a personalized subset of tasks, denoted by $\mathcal{S}_i \subseteq [S]$. This formulation constitutes the general FMOL problem.

A common approach to tackle (3) is **linear scalarization**, which transforms it into a single-objective problem by optimizing a fixed, weighted sum of the individual losses (Miettinen, 1999; Marler & Arora, 2004; Deb, 2011; Hu et al., 2022). In practice, the presence of conflicting gradients reveals its limitations (Sener & Koltun, 2018; Yang et al., 2022). Such conflicts imply that a single "utopia point" simultaneously optimizing all objectives is typically non-existent (Marler & Arora, 2004). The optimization goal thus shifts from finding this elusive utopia point to identifying solutions that represent the best possible compromises (Marler & Arora, 2004; Sener & Koltun, 2018). It motivates the fundamental concept of Pareto optimality: a solution is **Pareto optimal** if no objective can be improved without degrading at least one other. The set of all such solutions constitutes the **Pareto front**. To this end, we provide the rigorous mathematical definitions.

**Definition 3.1** (Pareto Concepts). A solution $x^*$ is **Pareto Optimal** if there does not exist another solution $x$ such that $f_s(x) \leq f_s(x^*)$ for all $s \in [S]$ with strict inequality for at least one task. A relaxed condition, **Weakly Pareto Optimal**, requires only that no $x$ satisfies $f_s(x) < f_s(x^*)$ for all

tasks simultaneously. A solution $x^*$ is **Pareto Stationary** if there is no common descent direction $d \in \mathbb{R}^p$ such that $\langle \nabla f_s(x^*), d \rangle < 0$ for all $s \in [S]$.

### The Proposed Algorithm: DREAM

The detailed procedure of DREAM is outlined in Algorithm 1. The framework coordinates the training process through two primary phases: client-side local updates, and server-side weight determination and global updates.

---

**Algorithm 1** DREAM: Drift-Corrected and Momentum-Smoothed Federated Multi-Objective Learning

---

**Initialize:** Total communication rounds $T$, local steps $K$, server learning rates $\eta_g, \eta_l, \eta_\lambda$, momentum parameter $\gamma \in (0, 1]$, updates $\{\bar{\Delta}_s^0 = \mathbf{0}_p\}_{s \in [S]}$, control variates $\{h_{s,i}^0 = \mathbf{0}_p\}_{s \in \mathcal{S}_i}, \forall i \in [N]$.
**for** each communication round $t = 0, \cdots, T-1$ **do**
  **for** each client $i \in [N]$ **in parallel do**
    **for** each task $s \in \mathcal{S}_i$ **do**
      Initialize local model $x_{s,i}^{t,0} \leftarrow x^t$.
      Repeat (4) for $k = 0, \ldots, K-1$.
    **end for**
  **end for**
  Send $\{\Delta_{s,i}^{t+1} = (x_{s,i}^{t,0} - x_{s,i}^{t,K})/(\eta_l K)\}_{s \in \mathcal{S}_i}$ to the server.
  **for** server **do**
    Aggregate $\bar{\Delta}_s^{t+1} \leftarrow \frac{1}{|\mathcal{R}_s|} \sum_{i \in \mathcal{R}_s} \Delta_{s,i}^{t+1}, \forall s \in [S]$.
    Obtain $\boldsymbol{\lambda}_\rho^{t+1}$ by solving (6) or (7).
    Update $x^{t+1} \leftarrow x^t - \eta_g \bar{\Delta}^{t+1} \mathbf{A}^\top \boldsymbol{\lambda}_\rho^{t+1}$.
    Send $x^{t+1}$ and $\{\bar{\Delta}_s^{t+1}\}_{s \in [S]}$ to clients.
  **end for**
  Each client $i \in [N]$ updates (5) in parallel, $\forall s \in \mathcal{S}_i$.
**end for**

---

**Client-Side Local Updates**. In each round $t$, participating clients $i \in \mathcal{R}_s$ perform local training to estimate the task-specific gradients. To mitigate variance and client drift, the local updates incorporate a control variate $h_{s,i}^t$ and a global momentum term $\bar{\Delta}_s^t$. The local model evolves over $K$ steps as:

$$x_{s,i}^{t,k+1} = x_{s,i}^{t,k} - \eta_l \left[ h_{s,i}^t + \gamma g_{s,i}(x_{s,i}^{t,k}) + (1-\gamma)\bar{\Delta}_s^t \right], \quad (4)$$

where $g_{s,i}$ is the local stochastic gradient, $\gamma \in (0, 1]$ is a momentum parameter. After completing local training, the client computes the update $\Delta_{s,i}^{t+1} = (x_{s,i}^{t,0} - x_{s,i}^{t,K})/(\eta_l K)$. Upon receiving the aggregated global update $\bar{\Delta}_s^{t+1} = \frac{1}{|\mathcal{R}_s|} \sum_{i \in \mathcal{R}_s} \Delta_{s,i}^{t+1}$ broadcast by the server, the client updates its control variate for the subsequent round:

$$h_{s,i}^{t+1} = h_{s,i}^t + \bar{\Delta}_s^{t+1} - \Delta_{s,i}^{t+1} \quad (5)$$
$$= \frac{\gamma}{|\mathcal{R}_s|K} \sum_{k=0}^{K-1} \sum_{i \in \mathcal{R}_s} g_{s,i}(x_{s,i}^{t,k}) - \frac{\gamma}{K} \sum_{k=0}^{K-1} g_{s,i}(x_{s,i}^{t,k}),$$

where the derivation of the second equality is given in Appendix A.3. The rationale behind this specific update design is to fundamentally enhance the quality of the aggregated inputs at the server, which is a prerequisite for identifying a reliable multi-objective update direction (Chen et al., 2024a). Inspired by Karimireddy et al. (2020); Zhou et al. (2026), the control variate mechanism effectively counteracts client drift by explicitly incorporating global gradient information into local training. However, as highlighted in the earlier discussion, correcting for drift is necessary but not sufficient for gradient estimation in stochastic MOO. Consequently, the momentum mechanism is integrated to simultaneously smooth out stochastic fluctuations. As the analysis will formally demonstrate, this input-smoothing mechanism is not merely an enhancement but a **critical requirement** for achieving guaranteed convergence in the FMOL setting.

**Server-Side Weight Determination and Global Update**. The server first aggregates the received client updates to form the task-specific global update $\bar{\Delta}_s^{t+1}$. Based on these aggregated updates, the next step is to compute weights $\boldsymbol{\lambda}_\rho$ that define the update direction in the $\mathbf{A}$-induced objective geometry. We formulate this as a **generalized regularized QP subproblem**, incorporating a **task correction matrix** $\mathbf{A} = [A_{sj}] \in \mathbb{R}^{S \times S}$ and a regularization term:

$$\min_{\boldsymbol{\lambda} \in \boldsymbol{\omega}} J_\rho(\boldsymbol{\lambda}; \bar{\Delta}^{t+1}) = \frac{1}{2} \left\| \bar{\Delta}^{t+1} \mathbf{A}^\top \boldsymbol{\lambda} \right\|^2 + \frac{\rho}{2} \|\boldsymbol{\lambda}\|^2, \quad (6)$$

where $\bar{\Delta} = [\bar{\Delta}_1, \cdots, \bar{\Delta}_S] \in \mathbb{R}^{p \times S}$, $\rho > 0$. The detailed derivations of this subproblem can be found in Appendix A.2. The regularization term $\frac{\rho}{2}\|\boldsymbol{\lambda}\|^2$ imparts strong convexity to the subproblem, ensuring the Lipschitz continuity of the weight mapping, which is essential for theoretical analysis (Chen et al., 2024a). This subproblem can be solved exactly to obtain the optimal weights $\boldsymbol{\lambda}_\rho^{t+1}$, typically using the Frank-Wolfe (FW) algorithm (Wang et al., 2016) or standard QP solvers. Alternatively, to further reduce computational overhead, it supports a step-update strategy, which updates the weights via a single projected gradient step:

$$\boldsymbol{\lambda}_\rho^{t+1} = \Pi_{\boldsymbol{\omega}} \big( \boldsymbol{\lambda}_\rho^t - \eta_\lambda \underbrace{\left[ \mathbf{A}(\bar{\Delta}^t)^\top \bar{\Delta}^t \mathbf{A}^\top + \rho \mathbf{I} \right] \boldsymbol{\lambda}_\rho^t}_{=\nabla_{\boldsymbol{\lambda}} J_\rho(\boldsymbol{\lambda}_\rho^t; \bar{\Delta}^t)} \big). \quad (7)$$

Finally, the server updates the global model using the computed global update direction

$$d^{t+1} = \bar{\Delta}^{t+1} \mathbf{A}^\top \boldsymbol{\lambda}_\rho^{t+1} = \sum_{s \in [S]} \lambda_{s,\rho}^{t+1} \sum_{j \in [S]} A_{sj} \bar{\Delta}_j^{t+1}.$$

**Generality of the Framework**. The task correction matrix $\mathbf{A}$ linearly transforms the task-update vectors before the simplex weights are applied. Setting $\mathbf{A} = \mathbf{I}$ (identity matrix) reduces the framework to the standard MGDA formulation. However, practical MTL applications often involve inherent task correlations and specific preference requirements. By

configuring the algebraic structure of $\mathbf{A}$, this framework provides a generalized formulation capable of covering diverse scenarios:

**(i) Linear Scalarization**: Setting $\mathbf{A}$ as a matrix with all entries equal to $1/S$ (denoted as $\mathbf{J} = \frac{1}{S}\mathbf{1}\mathbf{1}^\top$) degenerates the problem into single-objective optimization with equal weighting, akin to FedAvg. Beyond this extreme, this framework allows for a continuous transition between MGDA and scalarization by setting $\mathbf{A} = (1 - \beta)\mathbf{I} + \beta\mathbf{J}$ with $\beta \in [0, 1]$. This soft interpolation mechanism offers a flexible trade-off between conflict avoidance and average performance, which is more elegant and expressive than FedMGDA+ (Hu et al., 2022) that relies on imposing hard constraints on weights to bridge these two regimes.

**(ii) Weighted Prioritization**: Configuring $\mathbf{A}$ as a diagonal matrix applies a scalar transformation to each task's gradient field. This allows the framework to enforce task priorities, whether derived from specific user preferences (Lin et al., 2019; Chen et al., 2024b) or designed to counterbalance inherent magnitude disparities among objectives (Chen et al., 2018). For instance, when task losses differ by orders of magnitude, appropriate scaling prevents an excessive bias towards dominant tasks, thereby ensuring a more balanced and fair optimization process across all objectives (Javaloy & Valera, 2022).

**(iii) Task Interaction Modeling**: By configuring $\mathbf{A}$ with non-zero off-diagonal entries, the framework explicitly encodes inter-task correlations into the aggregation. This structure naturally accommodates task grouping priors (Standley et al., 2020), where dense interactions are enforced only among related tasks. Furthermore, it conceptually aligns with direction-oriented strategies (Xiao et al., 2023) by allowing the weighted average of gradients to serve as a guiding reference, thereby leveraging shared information to stabilize and steer the optimization of individual tasks.

## 4. Theoretical Analysis

The analysis is built upon the following assumptions, but does not require them to always hold simultaneously.

**Assumption 4.1** ($L$-Smoothness)**.** Each local function $f_{s,i}$ for all $s \in [S]$ and $i \in [N]$ is differentiable, and its gradient $\nabla f_{s,i}$ is Lipschitz continuous with constant $L > 0$.

**Assumption 4.2** (Stochastic Gradients)**.** Let $g_{s,i}(x, \xi)$ be the stochastic gradient of the local gradient $\nabla f_{s,i}$, where $\xi$ represents the randomness from data sampling, $s \in [S]$ and $i \in \mathcal{R}_s$. For notational simplicity, we omit $\xi$ and write $g_{s,i}(x)$. We assume $g_{s,i}$ is an unbiased estimator and has a uniformly bounded variance, i.e., $\mathbb{E}[g_{s,i}(x, \xi)] = \nabla f_{s,i}(x)$ and $\mathbb{E}[\|g_{s,i}(x, \xi) - \nabla f_{s,i}(x)\|^2] \leq \sigma^2$.

**Assumption 4.3** (Bounded Gradients)**.** The gradient of $f_s$ is uniformly bounded. Specifically, there exists a constant

$G$ such that for all $s \in [S]$: $\|\nabla f_s(x)\| \leq G$.

**Assumption 4.4** ($\mu$-Strong Convexity)**.** Each objective function $f_s$ for $s \in [S]$ is $\mu$-strongly convex with $\mu > 0$.

**Assumption 4.5** (Bounded Heterogeneity and Stochastic Noise)**.** The dissimilarity between local and global gradients is bounded uniformly, i.e., $\|\nabla f_{s,i}(x) - \nabla f_s(x)\| \leq \sigma_D, \forall s \in [S], \forall i \in \mathcal{R}_s$. The stochastic gradient noise is bounded almost surely, i.e., $\|g_{s,i}(x, \xi) - \nabla f_{s,i}(x)\| \leq \sigma_G$.

The theoretical analysis encompasses both general non-convex and strongly convex settings. The assumptions required for each result are specified when the corresponding result is stated. We note that Assumptions 4.3 and 4.4 generally do not hold simultaneously over the entire space. However, consistent with related works (Yang et al., 2023; Xu et al., 2025), we adopt Assumption 4.3 in the strongly convex analysis to facilitate derivations.

*Remark* 4.6. Chen et al. (2024a, Lemma 1) demonstrates that for smooth and strongly convex objectives, the iterate sequence $\{x^t\}$ generated by its stochastic dynamic weighting algorithms remains uniformly bounded. This boundedness implies that the gradient norms are effectively bounded along the optimization trajectory. A similar boundedness property can theoretically be established by following an analogous inductive proof. This motivates treating Assumption 4.3, when combined with strong convexity, as a trajectory-level boundedness condition rather than a global bounded-gradient requirement over the entire space.

*Remark* 4.7. Assumptions 4.1-4.4 constitute the standard theoretical foundation commonly adopted in stochastic MOO literature (Chen et al., 2025; Yang et al., 2023). Assumption 4.5 is introduced as a requisite condition to establish a uniform almost-sure upper bound on the aggregated updates. Crucially, this additional constraint is invoked selectively: it is primarily required for the analysis of the Step-Update strategy to rigorously control the higher-order moments induced by the weight tracking error. For the Exact-Update strategy, the analysis proceeds without Assumption 4.5, thereby establishing convergence under weaker conditions than prior works (Yang et al., 2023; Askin et al., 2025) that often rely on bounded heterogeneity. This distinction reflects the analytical principle of **minimal sufficiency**: we aim to establish convergence results using the weakest possible assumptions, introducing stronger constraints (like bounded heterogeneity) only when they are strictly necessary for the specific algorithmic variant.

**Coupled Errors**. Before detailing the convergence results, we identify two fundamental error sources: the gradient estimation error $\mathcal{E}_s^t = \|\nabla f_s(x^t) - \bar{\Delta}_s^{t+1}\|^2$ and the client drift $\mathcal{C}_s^t = \frac{1}{|\mathcal{R}_s|K} \sum_{i \in \mathcal{R}_s} \sum_{k=0}^{K-1} \|x_{s,i}^{t,k} - x^t\|^2$. In FMOL, these errors are perniciously coupled in a **vicious cycle**: the biased gradient estimation ($\mathcal{E}_s^t$) leads to inaccurate global updates that exacerbate client drift in the subsequent round ($\mathcal{C}_s^{t+1}$);

subsequently, this accumulated systematic shift (drift) corrupts the new gradient estimation ($\mathcal{E}_s^{t+1}$). The analysis in Appendix C formalizes this recursive interplay:

$$\mathbb{E}[\mathcal{C}_s^{t+1}] \leq \Theta(\gamma\eta_l^2)\mathbb{E}[\mathcal{C}_s^t] + \mathcal{O}(\eta_l^2)\mathbb{E}[\mathcal{E}_s^t] + \cdots ,$$
$$\mathbb{E}[\mathcal{E}_s^{t+1}] \leq (1 - \Theta(\gamma))\mathbb{E}[\mathcal{E}_s^t] + \mathcal{O}(\gamma)\mathbb{E}[\mathcal{C}_s^{t+1}] + \cdots .$$

Prior frameworks typically circumvent this coupled difficulty by invoking the bounded data heterogeneity assumption (Assumption 4.5). It artificially simplifies the analysis by capping the gradient variance component within $\mathbb{E}[\mathcal{E}_s^{t+1}]$ with a constant $\mathcal{O}(\sigma^2 + \sigma_D^2)$, essentially severing the feedback loop from $\mathbb{E}[\mathcal{C}_s^{t+1}]$. In contrast, our analysis explicitly controls this coupling via the proposed correction mechanism without relying on such simplifications. Establishing these lays the foundation for the subsequent analysis across both non-convex and strongly convex settings.

**Geometric Impact of Regularization**. The regularization term in the subproblem fundamentally alters the optimization geometry. For Exact-Update, the descent condition follows a relaxed inequality (Lemma D.1 in Appendix D):

$$-\langle \bar{\boldsymbol{\Delta}}^{t+1}\mathbf{A}^\top \boldsymbol{\lambda}, d^{t+1}\rangle \leq -\|d^{t+1}\|^2 + \rho.$$

This inequality indicates that $\rho$ acts as a controlled ascent tolerance, permitting slight deviations from the strict descent cone. While this relaxation introduces a bias that vanishes as $\rho \to 0$, it serves as a necessary trade-off to ensure the Lipschitz continuity of the weights, thereby stabilizing the optimization trajectory. For the Step-Update strategy, the descent condition becomes more complex due to the inexact weight solution (Lemma D.2 in Appendix D):

$$-\langle \bar{\boldsymbol{\Delta}}^{t+1}\mathbf{A}^\top \boldsymbol{\lambda}, d^{t+1}\rangle \leq -\left[1 - \eta_\lambda\rho - \mathcal{O}(\eta_\lambda)\right]\|d^{t+1}\|^2$$
$$- \frac{1}{2\eta_\lambda}\left(\|\boldsymbol{\lambda}_\rho^{t+2} - \boldsymbol{\lambda}\|^2 - \|\boldsymbol{\lambda}_\rho^{t+1} - \boldsymbol{\lambda}\|^2\right) + \mathcal{O}(\rho).$$

The effective descent magnitude is also dampened by the weight learning rate $\eta_\lambda$. Moreover, the additional term explicitly captures the evolution of the weight tracking error, representing the penalty incurred by avoiding the exact QP solution. Consequently, an appropriate selection of $\eta_\lambda$ is required to balance the descent magnitude against this tracking error. This introduces additional complexity to the error bound and analysis, serving as the necessary trade-off for the significant reduction in computational overhead.

**Main Convergence Results**

To facilitate the theoretical analysis, we adopt the standard full-participation setting where all clients contribute to every task, i.e., $|\mathcal{R}_s| = N$ for all $s \in [S]$. We assume that the task correction matrix $\mathbf{A}$ is entry-wise nonnegative and satisfies $\nu_A := \min_{s\in[S]} \sum_{j\in[S]} A_{sj} > 0$. This structural condition ensures that the $\mathbf{A}$-induced scalarized objectives preserve

the smoothness and convexity inequalities used in the proofs. We begin by establishing the convergence guarantees to Pareto stationary points for general non-convex objectives.

**Theorem 4.8** (General Non-Convex Convergence). *Under Assumptions 4.1, 4.2, 4.3, and additionally Assumption 4.5 if $\mathbb{I}_{step} = 1$, assuming the $\mathbf{A}$-weighted objectives are lower bounded, with the step size conditions in Appendix E, the sequence generated by Algorithm 1 satisfies*

$$\min_{0\leq r\leq T-1} \mathbb{E}\|\nabla\mathbf{F}(x^r)\mathbf{A}^\top\boldsymbol{\lambda}_0^*(x^r)\|^2 \leq \mathcal{O}\left(\frac{1}{\eta_g T} + \frac{1}{\gamma T} + \frac{\gamma\eta_l^2}{T}\right)$$
$$+ \mathcal{O}\left(\gamma\eta_l^2\sigma^2 + \eta_l^2 G^2 + \frac{\gamma\sigma^2}{N} + \rho\right) + \mathbb{I}_{step}\cdot\mathcal{O}\left(\frac{1}{\eta_\lambda T}\right),$$

*where $\boldsymbol{\lambda}_0^*(x^r)$ are optimal weights for the unregularized problem, i.e., $\arg\min_{\boldsymbol{\lambda}\in\boldsymbol{\omega}} \frac{1}{2}\left\|\nabla\mathbf{F}(x^r)\mathbf{A}^\top\boldsymbol{\lambda}\right\|^2$, and $\mathbb{I}_{step}$ is 1 for Step-Update and 0 otherwise.*

Convergence requires satisfying specific stability conditions (see Appendix E for more details). Qualitatively, the server learning rate is bounded by the momentum parameter ($\eta_g \lesssim \gamma$), while the weight update step $\eta_\lambda$ is constrained by both the regularization strength ($\eta_\lambda\rho \lesssim 1$) and a problem-dependent constant. By appropriately tuning parameters, we derive the convergence rate to Pareto stationary point.

**Corollary 4.9.** *With parameters $\gamma = \Theta(N^{\frac{1}{2}}T^{-\frac{1}{2}})$, $\eta_l = \Theta(T^{-\frac{1}{2}})$, $\eta_g = \Theta(N^{\frac{1}{2}}T^{-\frac{1}{2}})$ and $\rho = \Theta(N^{-\frac{1}{2}}T^{-\frac{1}{2}})$, Theorem 4.8 yields a convergence rate of $\mathcal{O}(1/\sqrt{NT})$. For the Step-Update strategy, this rate holds provided that $\eta_\lambda$ is chosen as a sufficiently small constant.*

The convergence rate is dominated by the term $\mathcal{O}(1/\sqrt{NT})$, provided that the number of communication rounds $T$ is sufficiently large. This result implies a **linear speedup** in the number of clients: doubling $N$ improves convergence by a factor of $1/\sqrt{2}$, for fixed $T$. Such parallel scalability is particularly desirable in large-scale FL. To our knowledge, this is the first linear-speedup result for FMOL that does not rely on bounded data heterogeneity for Exact-Update. Moreover, it highlights the critical role of the momentum mechanism for guaranteeing convergence without large batch sizes. If momentum is disabled (e.g., $\gamma = 1$), then the variance bound in Theorem 4.8 becomes non-vanishing and prevents convergence, even as $T \to \infty$.

*Remark* 4.10. In the special case of deterministic gradients (i.e., $\sigma^2 = 0$), all variance-related terms in the upper bound vanish. In this scenario, by setting $\eta_l = \mathcal{O}(T^{-1/2})$, it leads to a faster convergence rate of $\mathcal{O}(1/T)$, which aligns with the convergence rate for deterministic non-convex MOO algorithms in Fliege et al. (2019); Yang et al. (2023).

Beyond this convergence rate, we analyze the geometric alignment of the descent direction. The following theorem bounds the CA distance, defined as the Euclidean distance between the actual direction $d^{t+1}$ and the ideal direction.

**Theorem 4.11** (CA Direction Distance). *Under Assumptions 4.1, 4.2, 4.3, and additionally Assumption 4.5 if $\mathbb{I}_{step} = 1$, with the step size conditions similar to Theorem 4.8 (with minor coefficient adjustments, see Appendix F), the time-averaged squared CA distance satisfies:*

$$\min_{0 \le r \le T-1} \mathbb{E}\|\nabla\mathbf{F}(x^r)\mathbf{A}^\top\boldsymbol{\lambda}_0^*(x^r) - d^{r+1}\|^2 \le \mathcal{O}\Big(\frac{1}{\eta_g T} + \frac{1}{\gamma T}\Big)$$
$$+ \mathcal{O}\Big(\eta_l^2 + \frac{\gamma\sigma^2}{N} + \rho\Big) + \mathbb{I}_{step} \cdot \mathcal{O}\Big(\frac{1}{\eta_\lambda T} + \frac{\eta_g^2}{\eta_\lambda^2\rho^2}\Big).$$

By instantiating the parameters, we derive the specific convergence rates for the CA distance.

**Corollary 4.12.** *For Exact-Update, the algorithm achieves a vanishing CA distance at the optimal rate of $\mathcal{O}(1/\sqrt{NT})$, by adopting the same parameter schedule as in Corollary 4.9. For Step-Update, by setting $\eta_g = \Theta(T^{-\frac{3}{5}})$, $\gamma = \Theta(T^{-\frac{2}{5}})$, and $\rho = \Theta(T^{-\frac{2}{5}})$, the CA distance converges at a rate of $\mathcal{O}(T^{-\frac{2}{5}})$.*

For the Exact-Update case, the CA distance bound shares the same structure as the convergence result, implying that the update direction aligns with the ideal CA direction as fast as stationarity is achieved. In contrast, the Step-Update case involves a trade-off between tracking error and regularization bias, resulting in a convergence rate of $\mathcal{O}(T^{-\frac{2}{5}})$. Notably, this reduced rate is superior to the $\mathcal{O}(T^{-\frac{1}{3}})$ bias reduction rate achieved by MoCo+ (Fernando et al., 2024). This demonstrates that DREAM's regularized formulation maintains high directional accuracy even when computational efficiency is prioritized.

We further analyze the convergence behavior when the objective functions are strongly convex, where stronger convergence guarantees can be established. Let $x^*$ denote the unique Pareto optimal solution associated with a specific weight vector $\boldsymbol{\lambda}_0^*(x^*)$. Under strong convexity, it satisfies the first-order optimality condition $\nabla\mathbf{F}(x^*)\mathbf{A}^\top\boldsymbol{\lambda}_0^*(x^*) = \mathbf{0}$. The analysis for this setting is centered around a carefully constructed Lyapunov function, designed to capture the evolution of the entire system, which is defined as:

$$\varphi^t = \mathbb{E}\|x^t - x^*\|^2 + \gamma\sum_{s \in [S]}\big[\mathbb{E}[\mathcal{E}_s^t] + \gamma(1+\gamma)L^2\mathbb{E}[\mathcal{C}_s^t]\big].$$

By proving the geometric contraction of $\varphi$, we establish the following theorem.

**Theorem 4.13** (Strongly Convex Convergence). *Under Assumptions 4.1, 4.2, 4.4, with appropriate step sizes (see Appendix G), the weighted sub-optimality gap satisfies:*

$$\min_{0 \le r \le T-1} \mathbb{E}[\mathbf{F}(x^r) - \mathbf{F}(x^*)]\mathbf{A}^\top\boldsymbol{\lambda}_\rho^{r+1}$$
$$\le \mathcal{O}\Big(\exp\Big(-\frac{\eta_g\mu T}{4}\Big)\Big) + \mathcal{O}\Big(\eta_g + \frac{\gamma^2\eta_l^2}{\eta_g} + \frac{\gamma^3\sigma^2}{\eta_g}\Big).$$

*Simultaneously, the Lyapunov function exhibits general geometric contraction*

$$\varphi^{t+1} \le \Big(1 - \frac{\eta_g\mu\nu_A}{4}\Big)\varphi^t + \mathcal{O}\big(\gamma^2\eta_l^2 + \gamma^3\big)$$
$$+ \frac{\eta_g\|\nabla\mathbf{F}(x^*)\mathbf{A}^\top\|_2^2}{\mu\nu_A}\mathbb{E}\underbrace{\|\boldsymbol{\lambda}_\rho^{t+1} - \boldsymbol{\lambda}_0^*(x^*)\|^2}_{\le 2,\ \boldsymbol{\lambda} \in \boldsymbol{\omega}},$$

*where $\nu_A := \min_{s \in [S]}(\sum_{j \in [S]} A_{sj})$.*

*Furthermore, by incorporating the additional Assumptions 4.3, 4.5 and letting $\rho > 0$, we can refine the last weight error term. Specifically, for Exact-Update, the error bound is explicitly characterized by $\mathcal{O}(\eta_g\rho^2 + \eta_g^2)$; while for Step-Update, by augmenting the Lyapunov function with a weight tracking error term ($\mathbb{E}\|\boldsymbol{\lambda}_\rho^{t+1} - \boldsymbol{\lambda}_{\rho,*}^{t+1}\|^2$, see Appendix G), we establish a similar contraction with the error bound $\mathcal{O}(\eta_g\rho^2 + \eta_g^2 + \frac{\eta_g}{\eta_\lambda^2\rho^4})$.*

The theorem above provides general bounds. By appropriately tuning parameters, we can achieve convergence of the weighted sub-optimality gap. However, due to the intricate stability conditions, the Lyapunov function converges to a steady-state error floor rather than zero.

**Corollary 4.14.** *By setting $\eta_g = \Theta(\log T/T)$, the constraint $\eta_g \lesssim \gamma^2$ necessitates scaling the momentum as $\gamma = \Theta(\sqrt{\log T/T})$. With $\eta_l = \Theta(1/\sqrt{T})$, the weighted sub-optimality gap converges at a rate of $\tilde{\mathcal{O}}(1/\sqrt{T})$. Simultaneously, the Lyapunov function converges to a steady-state error neighborhood determined by the regularization parameter $\rho = \Theta(1)$. Specifically, Exact-Update converges to a neighborhood dominated by $\mathcal{O}(\rho^2)$, while Step-Update incurs an error floor of order $\mathcal{O}(\rho^2 + \rho^{-6})$.*

While existing works in the strongly convex setting primarily establish convergence for the weighted sub-optimality gap (Yang et al., 2023; Xu et al., 2025), our analysis goes further by providing a holistic view through the Lyapunov function. It encapsulates the distance to Pareto optimality, gradient estimation error, client drift, and the weight tracking error for the Step-Update strategy. The general Lyapunov contraction in Theorem 4.13 applies to both update strategies, as the weight approximation error is implicitly absorbed into the constant terms owing to the boundedness of the probability simplex. However, explicitly characterizing the impact of specific weight update schemes necessitates the strong convexity provided by $\rho > 0$. This refined analysis reveals that while Exact-Update is affected solely by the regularization bias $\mathcal{O}(\rho^2)$, Step-Update may significantly amplify the error floor due to accumulated tracking errors. Specifically, to maintain tracking stability, the weight update step $\eta_\lambda$ must be tightly constrained by $\rho$ (i.e., $\eta_\lambda \lesssim \rho$). This tight constraint becomes particularly problematic when $\rho$ is small (as desired for low regularization bias), as it slows down the weight adaptation and inflates the tracking error

term $\mathcal{O}(1/(\eta_\lambda^2\rho^4))$ to $\mathcal{O}(1/\rho^6)$. This explicitly characterizes the theoretical cost of substituting the exact QP solver with a single projected gradient step.

*Remark* 4.15. FSMGDA (Yang et al., 2023) claims the state-of-the-art convergence rate of $\tilde{\mathcal{O}}(1/T)$ for the weighted sub-optimality gap in strongly convex FMOL. However, this result hinges on a restrictive $(\alpha, \beta)$-Lipschitz continuity condition for stochastic gradients:

$$\mathbb{E}[\|\nabla f(\mathbf{x}, \xi) - \nabla f(\mathbf{y}, \xi')\|^2] \leq \alpha\|\mathbf{x} - \mathbf{y}\|^2 + \beta\sigma^2.$$

Crucially, achieving their reported rate requires setting $\beta = \mathcal{O}(\eta_g)$, which implicitly necessitates mega-batch sampling strategies that significantly inflate computational costs to control variance. In contrast, DREAM achieves convergence without such restrictive assumptions. Moreover, were we to adopt a similar mechanism (e.g., decaying $\sigma^2$), the noise-induced error floor would likewise vanish, enabling DREAM to achieve the same convergence rate.

## 5. Numerical Experiments

We evaluate DREAM on diverse benchmarks ranging from heterogeneous settings to large-scale scenarios. Specifically, we report results on Multi-MNIST and CIFAR-100 in the main text. Experimental setup and additional evaluations on the large-scale CelebA dataset (40 attributes) are detailed in Appendix H due to space constraints.

**Performance on Multi-MNIST**. We evaluate DREAM against representative baselines in a heterogeneous setting with $K = 40$ local steps. Figure 2(a) visualizes the optimization trajectories in the loss space. FSMGDA and FedCMOO exhibit clear performance skew, with FSMGDA biased toward Task 1 and FedCMOO favoring Task 2. In contrast, both variants of DREAM move toward a more balanced trade-off and attain solutions closer to the Pareto front. Figure 2(b) further shows that this trade-off can be controlled within the same framework. By setting the task correction matrix to diagonal forms ($T_1 = \text{diag}\{[2, 0.5]\}$ and $T_2 = \text{diag}\{[0.5, 2]\}$), DREAM can explicitly shift the optimization preference toward a target objective.

Figures 2(c) and 2(d) evaluate robustness as the number of local steps $K$ increases. FSMGDA suffers a noticeable drop in average accuracy, while FedCMOO maintains a relatively stable average accuracy but leaves a large and persistent gap between the two tasks. DREAM maintains both high average accuracy and a small task gap across different $K$ values. This comparison also isolates the effect of client-side correction: at $K = 40$ in Figure 2(c), removing the explicit control-variate correction reduces DREAM to an FSMGDA-style update without direct drift compensation, leading to a strong bias toward Task 1 and an average accuracy drop of about 6%. These results empirically confirm that the drift-aware correction mechanism mitigates the directional bias induced by client drift under intensive local computation.

Beyond drift correction, the momentum mechanism provides a complementary stabilization effect. Figure 2(e) shows that disabling momentum smoothing ($\gamma = 1$) yields a larger and more oscillatory descent-direction norm, indicating noisier inputs to the weight-solving subproblem. Figure 2(f) further shows that momentum smoothing stabilizes the task weights across rounds. Together with Figure 2(c)–(d), these results support the correct-then-smooth design: the control variate suppresses client drift, while momentum smoothing reduces stochastic oscillations before multi-objective aggregation.

**Performance on CIFAR-100**. To validate DREAM on a more complex vision benchmark, we conduct experiments on the CIFAR-100 dataset. Since each image in CIFAR-100 is associated with a "fine" label (100 classes) and a "coarse" label (20 superclasses), we formulate this as a multi-objective problem where the model must simultaneously perform fine-grained and coarse-grained classification. We compare the Step-Update variant of DREAM against FSMGDA and FedAvg (with fixed scalarization weights of $0.5/0.5$) in an i.i.d. setting. Figure 3 reports the test accuracy of the fine-grained and coarse-grained classification tasks, showing a clear performance advantage for DREAM. In terms of final test accuracy, DREAM achieves approximately 40.2% on the fine-grained task and 55.3% on the coarse-grained task. These results surpass FSMGDA by margins of $+2.4\%$ and $+3.5\%$, respectively. Similarly, compared to the static weighting of FedAvg, DREAM provides improvements of $+2.6\%$ and $+4.3\%$. This consistent improvement across both hierarchical objectives indicates that DREAM can improve shared representation learning and trade-off navigation on a more complex vision benchmark, yielding stronger performance than scalarization and standard MOO baselines.

## 6. Conclusion

In this paper, we introduced DREAM, a generalized optimization framework for controlling the recursive coupling between intra-task client drift and inter-task aggregation bias in heterogeneous FMOL. By combining task-wise control variates with momentum-smoothed local updates, DREAM improves the aggregated per-task updates used by a unified regularized QP formulation driven by a task correction matrix. We provided theoretical guarantees for convergence and CA direction distance, establishing a linear speedup rate of $\mathcal{O}(1/\sqrt{NT})$ under non-convex settings. For strongly convex objectives, we derived convergence guarantees for the weighted sub-optimality gap through a Lyapunov analysis. Numerical experiments further validate that DREAM improves task balance, robustness to local training, and overall

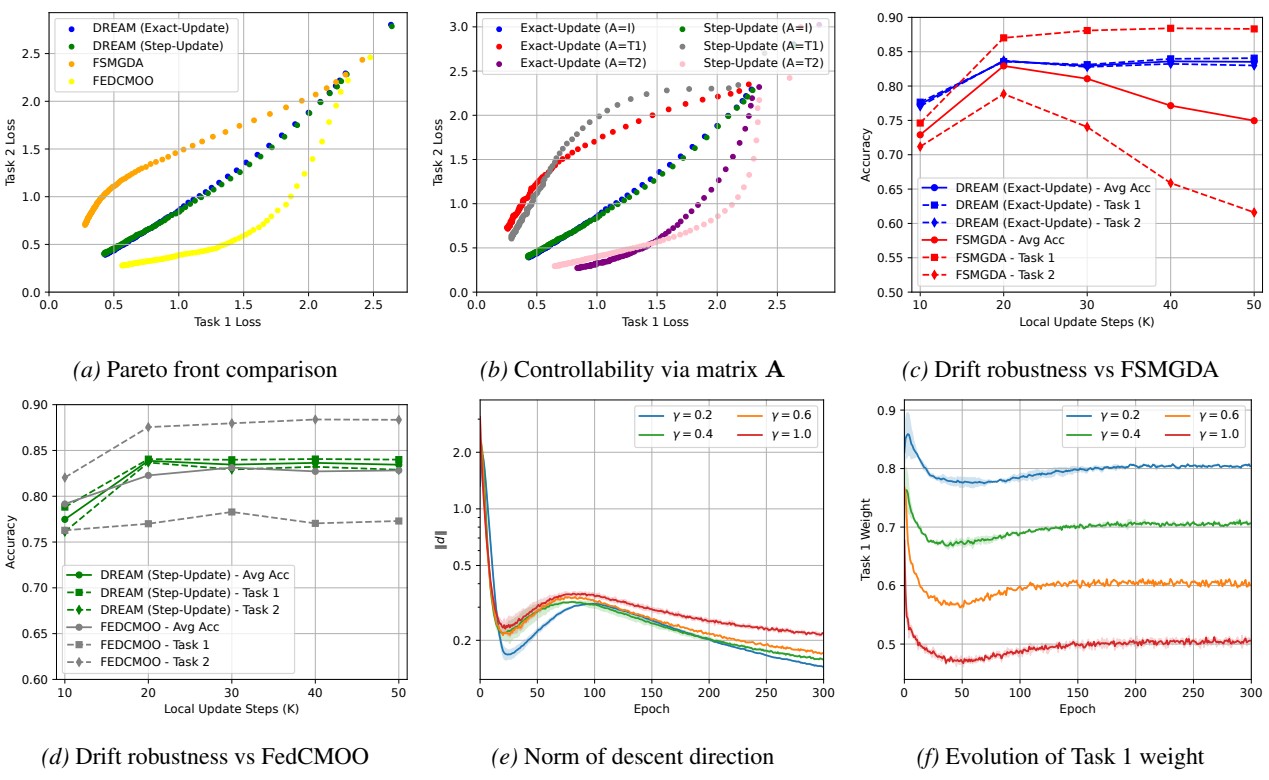

*(a)* Pareto front comparison     *(b)* Controllability via matrix **A**     *(c)* Drift robustness vs FSMGDA

*(d)* Drift robustness vs FedCMOO     *(e)* Norm of descent direction     *(f)* Evolution of Task 1 weight

*Figure 2.* Performance comparison on the Multi-MNIST dataset. For visualization clarity, the curves for $\gamma = 0.2, 0.4, 0.6$ in (f) are vertically shifted by $+0.3, +0.2, +0.1$, respectively.

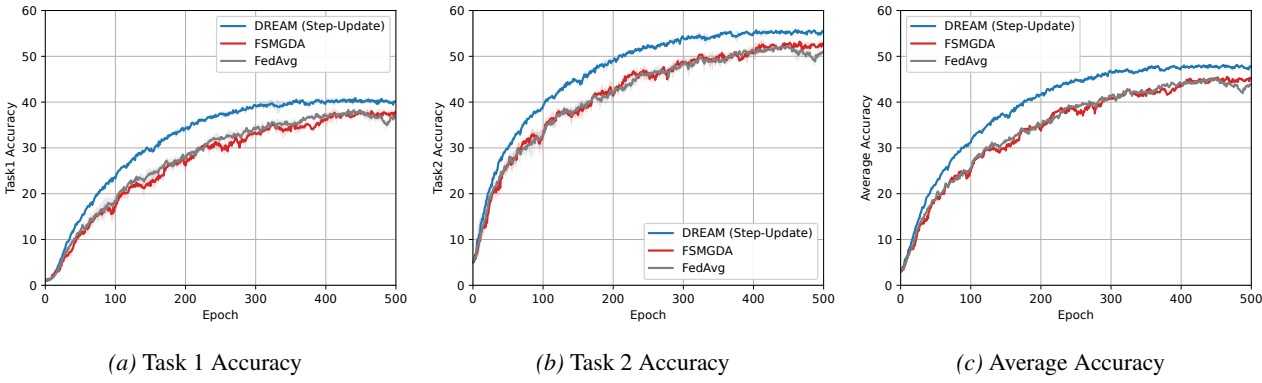

*(a)* Task 1 Accuracy     *(b)* Task 2 Accuracy     *(c)* Average Accuracy

*Figure 3.* Performance comparison on the CIFAR-100 dataset.

performance over representative FMOL baselines. Future work includes extending this framework to asynchronous or personalized settings.

## Acknowledgements

This work was supported by the National Natural Science Foundation of China under Grant Nos. 62473098, in part by the Australian Research Council under Grant DE250100961, in part by the SEU Innovation Capability Enhancement Plan for Doctoral Students under Grant CXJH_SEU_25243.

## Impact Statement

This work focuses on the theoretical and algorithmic aspects of federated multi-objective optimization. Its primary contribution is to improve the understanding of convergence behavior and stability under heterogeneous data distributions. While the proposed method may indirectly benefit large-scale distributed learning by improving optimization efficiency, it does not introduce application-specific mechanisms and does not have immediate societal impact.

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

# Contents of the Supplementary Material

# A. Algorithm Reformulation and Preliminaries

## A.1. Summary of Notations and Definitions

For ease of reference, we summarize the mathematical definitions used throughout the convergence analysis.

*Table 1.* Summary of Notations

| Category | Notation | Description |
|---|---|---|
| **System Model** | $N, S$ | Number of clients and tasks (objectives). |
| | $\mathcal{R}_s$ | Set of clients participating in task $s$ (size $|\mathcal{R}_s|$). |
| | $T, K$ | Total communication rounds and local update steps. |
| **Optimization** | $\mathbf{F}(x)$ | Vector of global objective functions $[f_1(x), \dots, f_S(x)]$. |
| | $\nabla \mathbf{F}(x)$ | Jacobian matrix of objectives, $[\nabla f_1(x), \cdots, \nabla f_S(x)] \in \mathbb{R}^{p \times S}$. |
| | $\mathbf{A}$ | Task correction matrix used to transform task updates. |
| | $\boldsymbol{\lambda}$ | Aggregation weights in probability simplex $\boldsymbol{\omega}$. |
| | $\mathbb{I}_{step}$ | Indicator: 1 for Step-Update strategy, 0 for Exact-Update. |
| **Updates** | $x^t, x_{s,i}^{t,k}$ | Global model at round $t$, local model at step $k$ for task $s$. |
| | $\Delta_{s,i}^{t+1}, \bar{\Delta}_s^{t+1}$ | Local update and aggregated global update for task $s$. |
| | $c_{s,i}^t, c_s^t$ | Control variates: local and global correction terms. |
| | $d^{t+1}$ | Global descent direction computed by the server. |
| **Hyperparams** | $\eta_l, \eta_g, \eta_\lambda$ | Learning rates for local client, global server, and weight update. |
| | $\gamma, \rho$ | Momentum parameter and regularization coefficient. |
| **Constants** | $L, \mu$ | Smoothness constant and strong convexity constant. |
| | $G, \sigma^2$ | Global gradient norm bound and stochastic variance bound. |
| | $\nu_A$ | Minimum row sum of $\mathbf{A}$, i.e., $\min_s \sum_j A_{sj}$. |
| **Errors** | $\mathcal{E}_s^t$ | Gradient estimation error, $\|\nabla f_s(x^t) - \bar{\Delta}_s^{t+1}\|^2$. |
| | $\mathcal{C}_s^t$ | Client drift error (Distance from global model), $\frac{1}{|\mathcal{R}_s|K} \sum_{i \in \mathcal{R}_s} \sum_{k=0}^{K-1} \|x_{s,i}^{t,k} - x^t\|^2$. |

For any gradient proxy matrix $\mathbf{G} \in \mathbb{R}^{p \times S}$ (instantiated as the aggregated update $\bar{\boldsymbol{\Delta}} = [\bar{\Delta}_1, \cdots, \bar{\Delta}_S] \in \mathbb{R}^{p \times S}$ or the true Jacobian $\nabla \mathbf{F}(x)$), the $\rho$-regularized multi-objective function is defined as

$$J_\rho(\boldsymbol{\lambda}; \mathbf{G}) = \frac{1}{2}\|\mathbf{G}\mathbf{A}^\top \boldsymbol{\lambda}\|^2 + \frac{\rho}{2}\|\boldsymbol{\lambda}\|^2 = \frac{1}{2}\boldsymbol{\lambda}^\top[\mathbf{A}\mathbf{G}^\top \mathbf{G}\mathbf{A}^\top + \rho\mathbf{I}]\boldsymbol{\lambda},$$

which is $\rho$-strongly convex with respect to $\boldsymbol{\lambda}$ for any $\rho > 0$. The weighting vector $\boldsymbol{\lambda}$ lies in the probability simplex $\boldsymbol{\omega}$, and the task correction matrix $\mathbf{A}$ is structured as follows:

$$\boldsymbol{\lambda} = \begin{bmatrix} \lambda_1 \\ \cdots \\ \lambda_S \end{bmatrix} \in \boldsymbol{\omega} = \left\{ \boldsymbol{\lambda} \in \mathbb{R}^S \mid \lambda_s \geq 0, \sum_{s \in [S]} \lambda_s = 1 \right\}, \quad \mathbf{A} = \begin{bmatrix} \mathbf{A}_1 \\ \cdots \\ \mathbf{A}_S \end{bmatrix} = \begin{bmatrix} A_{11} & \cdots & A_{1S} \\ \vdots & \ddots & \vdots \\ A_{S1} & \cdots & A_{SS} \end{bmatrix} \in \mathbb{R}^{S \times S}.$$

We distinguish below between the algorithmic weights maintained during training and the theoretical optimal weights used as reference points for convergence analysis:

- **Current Algorithmic Weights $\boldsymbol{\lambda}_\rho$.** The weight vector maintained by the algorithm at the current step.

- **Subproblem Optimal Weights $\boldsymbol{\lambda}_{\rho,*}$.** The optimal solution to the regularized subproblem defined by the current aggregated updates $\bar{\boldsymbol{\Delta}}$ (used for Exact-Update):

$$\boldsymbol{\lambda}_{\rho,*} = \arg\min_{\boldsymbol{\lambda} \in \boldsymbol{\omega}} J_\rho(\boldsymbol{\lambda}; \bar{\boldsymbol{\Delta}}) = \arg\min_{\boldsymbol{\lambda} \in \boldsymbol{\omega}} \frac{1}{2}\|\bar{\boldsymbol{\Delta}}\mathbf{A}^\top \boldsymbol{\lambda}\|^2 + \frac{\rho}{2}\|\boldsymbol{\lambda}\|^2.$$

- **Regularized Reference Weights** $\boldsymbol{\lambda}_\rho^*(x)$. The theoretical optimal weights with respect to the true Jacobian $\nabla\mathbf{F}(x)$. This map represents the ideal regularized weights that the algorithm tracks:

$$\boldsymbol{\lambda}_\rho^*(x) = \arg\min_{\boldsymbol{\lambda}\in\boldsymbol{\omega}} J_\rho(\boldsymbol{\lambda}; \nabla\mathbf{F}(x)) = \arg\min_{\boldsymbol{\lambda}\in\boldsymbol{\omega}} \frac{1}{2} \left\|\nabla\mathbf{F}(x)\mathbf{A}^\top\boldsymbol{\lambda}\right\|^2 + \frac{\rho}{2}\|\boldsymbol{\lambda}\|^2.$$

- **Pareto-Stationary Reference Weights** $\boldsymbol{\lambda}_0^*(x)$. The optimal weights for the unregularized problem ($\rho = 0$) with true gradients. These weights determine the Pareto descent direction used in the convergence criterion:

$$\boldsymbol{\lambda}_0^*(x) = \arg\min_{\boldsymbol{\lambda}\in\boldsymbol{\omega}} \frac{1}{2} \left\|\nabla\mathbf{F}(x)\mathbf{A}^\top\boldsymbol{\lambda}\right\|^2.$$

- **Optimal Pareto Weights** $\boldsymbol{\lambda}_0^*(x^*)$. The optimal weight vector associated with a Pareto stationary point $x^*$ (unique Pareto optimal solution in the context of strong convexity). By definition of stationarity in the generalized framework, the descent direction vanishes at $x^*$, implying

$$\nabla\mathbf{F}(x^*)\mathbf{A}^\top\boldsymbol{\lambda}_0^*(x^*) = \mathbf{0}.$$

Moreover, it is crucial to quantify the gap between the regularized solution $\boldsymbol{\lambda}_\rho^*(x^*)$ tracked by the algorithm and the true Pareto solution $\boldsymbol{\lambda}_0^*(x^*)$. When the unregularized solution is non-unique, $\boldsymbol{\lambda}_0^*(x^*)$ denotes the limiting unregularized optimizer selected by $\boldsymbol{\lambda}_\rho^*(x^*)$ as $\rho \downarrow 0$. Viewing the regularized problem as a perturbation of the unregularized one where the objective Hessian is shifted by $\rho\mathbf{I}$, we invoke standard sensitivity results for quadratic programming (Robinson, 1980). Since the unregularized problem is convex and the constraints are polyhedral, the optimal solution map is locally Lipschitz continuous with respect to the perturbation parameter $\rho$. Therefore, there exists a problem-dependent constant $C_{bias} > 0$ such that:

$$\|\boldsymbol{\lambda}_\rho^*(x^*) - \boldsymbol{\lambda}_0^*(x^*)\| \leq C_{bias}|\rho - 0| = \mathcal{O}(\rho).$$

This implies that the regularization-induced error can be efficiently controlled by $\rho$.

### A.2. Derivation of the Generalized Dual Problem

To formally derive the generalized update rule incorporating the task correction matrix $\mathbf{A}$ and the regularization term, we start from the primal perspective of finding a common descent direction in the corrected gradient space. Let $\mathbf{G} \in \mathbb{R}^{p\times S}$ be the gradient matrix. We seek a descent direction $d \in \mathbb{R}^p$ that minimizes a linear combination of the gradients projected by $\mathbf{A}$. Specifically, let $(\mathbf{A}\mathbf{G}^\top)_s$ denote the $s$-th row of the transformed gradient matrix $\mathbf{A}\mathbf{G}^\top$, representing the gradient of the $s$-th "corrected objective". We formulate the following optimization problem with slack variables $\boldsymbol{\xi} \in \mathbb{R}^S$ to allow for soft descent constraints:

$$\min_{d,\alpha,\boldsymbol{\xi}} \quad \alpha + \frac{1}{2}\|d\|^2 + \frac{1}{2\rho}\|\boldsymbol{\xi}\|^2$$
$$\text{s.t.} \quad (\mathbf{A}\mathbf{G}^\top)d \leq (\alpha + \xi_s)\mathbf{1}, \quad \forall s \in [S],$$

where, $\alpha$ represents the common descent amount, and the term $\frac{1}{2\rho}\|\boldsymbol{\xi}\|^2$ is controlled by the regularization parameter $\rho > 0$. This primal formulation extends the standard MGDA framework (which corresponds to $\mathbf{A} = \mathbf{I}$ and $\boldsymbol{\xi} = \mathbf{0}$) by introducing structural flexibility and robustness. Specifically, the matrix $\mathbf{A}$ transforms the raw task gradients into a corrected space, enabling the enforcement of task priorities or correlations directly within the descent condition. Furthermore, the slack variables $\boldsymbol{\xi}$ relax the strict common descent constraint found in classical MGDA. Instead of requiring the direction $d$ to improve all objectives equally (bounded by $\alpha$), we allow for task-specific deviations $\xi_s$, penalized by the term $\frac{1}{2\rho}\|\boldsymbol{\xi}\|^2$. This relaxation prevents the optimization from being overly constrained by conflicting gradients, and leads to a regularized dual problem that promotes smoother and more stable weight solutions.

We introduce the Lagrange multiplier vector $\boldsymbol{\lambda} \in \mathbb{R}^S$ ($\lambda_s \geq 0, \forall s \in [S]$) associated with the inequality constraints. The Lagrangian function is given by:

$$\begin{aligned}\mathcal{L}(d,\alpha,\boldsymbol{\xi},\boldsymbol{\lambda}) &= \alpha + \frac{1}{2}\|d\|^2 + \frac{1}{2\rho}\|\boldsymbol{\xi}\|^2 + \sum_{s\in[S]} \lambda_s \left([(\mathbf{A}\mathbf{G}^\top)d]_s - \alpha - \xi_s\right)\\ &= \alpha\left(1 - \mathbf{1}^\top\boldsymbol{\lambda}\right) + \left(\frac{1}{2}\|d\|^2 + d^\top\mathbf{G}\mathbf{A}^\top\boldsymbol{\lambda}\right) + \left(\frac{1}{2\rho}\|\boldsymbol{\xi}\|^2 - \boldsymbol{\lambda}^\top\boldsymbol{\xi}\right).\end{aligned}$$

Since the primal problem is convex, the optimal solution satisfies the zero-gradient conditions of the Lagrangian with respect to the primal variables:

Optimality w.r.t. $\alpha$:

$$\nabla_\alpha \mathcal{L} = 1 - \sum_{s \in [S]} \lambda_s = 0 \implies \boldsymbol{\lambda} \in \boldsymbol{\omega}. \qquad \text{(Simplex Constraint)}$$

This derives the simplex constraint on the weights naturally.

Optimality w.r.t. $d$:

$$\nabla_d \mathcal{L} = d + \mathbf{G}\mathbf{A}^\top \boldsymbol{\lambda} = 0 \implies d = -\mathbf{G}\mathbf{A}^\top \boldsymbol{\lambda}.$$

This confirms that the optimal direction is a linear combination of gradients weighted by $\mathbf{A}^\top \boldsymbol{\lambda}$. Substituting $d$ back into the Lagrangian:

$$\frac{1}{2}\|d\|^2 + (d)^\top(-d) = -\frac{1}{2}\|\mathbf{G}\mathbf{A}^\top \boldsymbol{\lambda}\|^2.$$

Optimality w.r.t. $\boldsymbol{\xi}$:

$$\nabla_{\boldsymbol{\xi}} \mathcal{L} = \frac{1}{\rho}\boldsymbol{\xi} - \boldsymbol{\lambda} = 0 \implies \boldsymbol{\xi} = \rho\boldsymbol{\lambda}.$$

Substituting $\boldsymbol{\xi}$ back:

$$\frac{1}{2\rho}\|\rho\boldsymbol{\lambda}\|^2 - \boldsymbol{\lambda}^\top(\rho\boldsymbol{\lambda}) = \frac{\rho}{2}\|\boldsymbol{\lambda}\|^2 - \rho\|\boldsymbol{\lambda}\|^2 = -\frac{\rho}{2}\|\boldsymbol{\lambda}\|^2.$$

Combining these terms, the dual problem is to maximize the concave function over $\boldsymbol{\lambda} \in \boldsymbol{\omega}$:

$$\max_{\boldsymbol{\lambda} \in \boldsymbol{\omega}} \quad -\frac{1}{2}\|\mathbf{G}\mathbf{A}^\top \boldsymbol{\lambda}\|^2 - \frac{\rho}{2}\|\boldsymbol{\lambda}\|^2.$$

This is equivalent to minimizing the convex quadratic objective defined in our framework:

$$\min_{\boldsymbol{\lambda} \in \boldsymbol{\omega}} \quad J_\rho(\boldsymbol{\lambda}; \mathbf{G}) = \frac{1}{2}\|\mathbf{G}\mathbf{A}^\top \boldsymbol{\lambda}\|^2 + \frac{\rho}{2}\|\boldsymbol{\lambda}\|^2.$$

### A.3. Algorithm Reformulation for Analysis

In single-objective FL, several works have shown that client drift can be effectively suppressed by incorporating a global correction term into local updates (Cheng et al., 2024; Karimireddy et al., 2020; Mishchenko et al., 2022; Zhou et al., 2026). The core idea, as pioneered by SCAFFOLD (Karimireddy et al., 2020), is to modify each local update to align it with the global descent direction. For a specific task $s$, this involves defining an average local update and its global counterpart:

$$c_{s,i}^{t+1} = \frac{1}{K}\sum_{k=0}^{K-1} g_{s,i}(x_{s,i}^{t,k}), \qquad c_s^{t+1} = \frac{1}{|\mathcal{R}_s|}\sum_{i \in \mathcal{R}_s} c_{s,i}^{t+1}.$$

By replacing the local gradient with a corrected direction $g_{s,i} - c_{s,i}^t + c_s^t$, the update step is steered towards the global average, making it a better estimator.

The control variate $h_{s,i}$ in DREAM achieves a similar corrective effect. From the local update rule (4), i.e.,

$$x_{s,i}^{t,k+1} = x_{s,i}^{t,k} - \eta_l \left[ h_{s,i}^t + \gamma g_{s,i}(x_{s,i}^{t,k}) + (1-\gamma)\bar{\Delta}_s^t \right],$$

we can establish the following relationships:

$$\Delta_{s,i}^{t+1} = (x_{s,i}^{t,0} - x_{s,i}^{t,K})/(\eta_l K) = h_{s,i}^t + \gamma c_{s,i}^{t+1} + (1-\gamma)\bar{\Delta}_s^t,$$
$$\bar{\Delta}_s^{t+1} = \sum_{i \in \mathcal{R}_s} \Delta_{s,i}^{t+1}/|\mathcal{R}_s| = \gamma c_s^{t+1} + (1-\gamma)\bar{\Delta}_s^t.$$

The second equality holds because the update of control variate (5), i.e., $h_{s,i}^{t+1} = h_{s,i}^t + \bar{\Delta}_s^{t+1} - \Delta_{s,i}^{t+1}$ ensures that the sum of client-side control variates is conserved and remains zero across rounds, i.e.,

$$\sum_{i \in \mathcal{R}_s} h_{s,i}^{t+1} = \sum_{i \in \mathcal{R}_s} h_{s,i}^t = \cdots = \sum_{i \in \mathcal{R}_s} h_{s,i}^0 = \mathbf{0}_p.$$

By substituting expressions of $\bar{\Delta}_s^{t+1} - \Delta_{s,i}^{t+1}$ back into (5), we arrive at an insightful relationship:

$$h_{s,i}^{t+1} + \gamma g_{s,i}(x_{s,i}^{t,k}) = \gamma \left[ c_s^{t+1} - c_{s,i}^{t+1} + g_{s,i}(x_{s,i}^{t,k}) \right].$$

This derivation reveals that the control variate $h_{s,i}$ effectively implements the SCAFFOLD correction term $(c_s - c_{s,i})$, scaled by the momentum parameter $\gamma$. This structural insight allows us to analyze the drift reduction and momentum smoothing effects in a unified manner. From a practical standpoint, DREAM streamlines the control variate management compared to SCAFFOLD by requiring only clients to maintain their internal state $h_{s,i}$.

**Reformulated Updates for Analysis**. Based on this insight, we present the equivalent mathematical representation of DREAM used throughout the convergence proofs:

$$\text{Local Model Update: } x_{s,i}^{t,k+1} = x_{s,i}^{t,k} - \eta_l \left[ \gamma g_{s,i}(x_{s,i}^{t,k}) + \gamma(c_s^t - c_{s,i}^t) + (1-\gamma)\bar{\Delta}_s^t \right], \quad x_{s,i}^{t,0} = x^t, \tag{8a}$$

$$\text{Local Update Calculation: } \Delta_{s,i}^{t+1} = \frac{1}{\eta_l K}(x_{s,i}^{t,0} - x_{s,i}^{t,K}) = \gamma(c_{s,i}^{t+1} + c_s^t - c_{s,i}^t) + (1-\gamma)\bar{\Delta}_s^t, \tag{8b}$$

$$\text{Aggregated Global Update: } \bar{\Delta}_s^{t+1} = \frac{1}{|\mathcal{R}_s|} \sum_{i \in \mathcal{R}_s} \Delta_{s,i}^{t+1} = \frac{\gamma}{|\mathcal{R}_s|} \sum_{i \in \mathcal{R}_s} c_{s,i}^{t+1} + (1-\gamma)\bar{\Delta}_s^t = \gamma c_s^{t+1} + (1-\gamma)\bar{\Delta}_s^t, \tag{8c}$$

$$\text{Local Control Variate Update: } c_{s,i}^{t+1} = \frac{1}{K} \sum_{k=0}^{K-1} g_{s,i}(x_{s,i}^{t,k}), \tag{8d}$$

$$\text{Global Control Variate Update: } c_s^{t+1} = c_s^t + \frac{1}{|\mathcal{R}_s|} \sum_{i \in \mathcal{R}_s} (c_{s,i}^{t+1} - c_{s,i}^t) = \frac{1}{|\mathcal{R}_s|} \sum_{i \in \mathcal{R}_s} c_{s,i}^{t+1}, \tag{8e}$$

$$\text{Weights Update: } \boldsymbol{\lambda}_\rho^{t+1} = \begin{cases} \boldsymbol{\lambda}_{\rho,*}^{t+1} = \arg\min_{\boldsymbol{\lambda} \in \boldsymbol{\omega}} \underbrace{\frac{1}{2} \boldsymbol{\lambda}^\top [\mathbf{A}(\bar{\mathbf{\Delta}}^{t+1})^\top \bar{\mathbf{\Delta}}^{t+1} \mathbf{A}^\top + \rho\mathbf{I}]\boldsymbol{\lambda}}_{=J_\rho(\boldsymbol{\lambda};\bar{\mathbf{\Delta}}^{t+1})}, \\ \Pi_{\boldsymbol{\omega}}\left(\boldsymbol{\lambda}_\rho^t - \eta_\lambda \underbrace{[\mathbf{A}(\bar{\mathbf{\Delta}}^t)^\top \bar{\mathbf{\Delta}}^t \mathbf{A}^\top + \rho\mathbf{I}] \boldsymbol{\lambda}_\rho^t}_{=\nabla_{\boldsymbol{\lambda}} J_\rho(\boldsymbol{\lambda}_\rho^t;\bar{\mathbf{\Delta}}^t)}\right), \end{cases} \tag{8f}$$

$$\text{Global Descent Direction: } d^{t+1} = \sum_{s \in [S]} \lambda_{s,\rho}^{t+1} \sum_{j \in [S]} A_{sj} \bar{\Delta}_j^{t+1} = \bar{\mathbf{\Delta}}^{t+1} \mathbf{A}^\top \boldsymbol{\lambda}_\rho^{t+1}, \tag{8g}$$

$$\text{Global Model Update: } x^{t+1} = x^t - \eta_g d^{t+1}. \tag{8h}$$

## B. Technical Lemmas

In this section, we gather the essential technical lemmas and detailed algorithmic formulations that form the foundation of our convergence analysis.

**Lemma B.1** (Basic Vector Inequalities). *For any $x, y \in \mathbb{R}^p$, the following inequalities hold:*

$$\|x + y\|^2 \leq (1+\eta)\|x\|^2 + (1+\eta^{-1})\|y\|^2, \quad \forall \eta > 0, \tag{9a}$$

$$\langle x, y \rangle \leq \frac{\eta}{2}\|x\|^2 + \frac{1}{2\eta}\|y\|^2, \quad \forall \eta > 0, \tag{9b}$$

$$-\|x\|^2 \leq \|x - y\|^2 - \frac{1}{2}\|y\|^2. \tag{9c}$$

**Lemma B.2** (Aggregation Inequalities). *For any vectors $x_i \in \mathbb{R}^p$, $i \in [N]$, the following inequalities hold:*

$$\left\| \sum_{i \in [N]} x_i \right\|^2 \leq N \sum_{i \in [N]} \|x_i\|^2, \tag{10a}$$

$$\left\| \sum_{i \in [N]} w_i x_i \right\|^2 \leq \sum_{i \in [N]} w_i \|x_i\|^2, \quad w_i \geq 0, \sum_{i \in [N]} w_i = 1, \tag{10b}$$

$$\sum_{i \in [N]} \|x_i - \bar{x}\|^2 \leq \sum_{i \in [N]} \|x_i\|^2, \quad \bar{x} := \frac{1}{N} \sum_{i \in [N]} x_i. \tag{10c}$$

Inequality (9a) follows from a generalized version of Young's inequality, and (9b) combines Cauchy-Schwarz inequality with Young's bound to decouple inner product terms. Inequality (9c) is derived from the triangle inequality and Young's inequality, providing a useful lower bound for the negative squared norm. Inequality (10a) provides a standard bound on the norm of a sum, while (10b) offers a tighter bound for convex combinations via Jensen's inequality. Inequality (10c) captures a classical variance decomposition that measures the dispersion of vectors around their average.

**Lemma B.3** ($L$-Smoothness). *If a function $f : \mathbb{R}^p \to \mathbb{R}$ is $L$-smooth, then for any $x, y \in \mathbb{R}^p$, the following inequalities hold:*

$$\|\nabla f(x) - \nabla f(y)\| \leq L\|x - y\|, \tag{11a}$$

$$f(x) - f(y) \leq \langle \nabla f(y), x - y \rangle + \frac{L}{2}\|x - y\|^2. \tag{11b}$$

**Lemma B.4** ($\mu$-Strong Convexity). *If a function $f : \mathbb{R}^p \to \mathbb{R}$ is $\mu$-strongly convex, then for any $x, y \in \mathbb{R}^p$, the following inequality holds:*

$$f(y) \geq f(x) + \langle \nabla f(x), y - x \rangle + \frac{\mu}{2}\|y - x\|^2, \tag{12a}$$

$$\langle \nabla f(x) - \nabla f(y), x - y \rangle \geq \mu\|y - x\|^2. \tag{12b}$$

**Lemma B.5** (Variance bound under martingale difference structure (Karimireddy et al., 2020)). *Let $\{\Xi_1, \ldots, \Xi_\tau\}$ be random vectors in $\mathbb{R}^p$ with conditional expectations defined as $\xi_i := \mathbb{E}[\Xi_i \mid \Xi_1, \ldots, \Xi_{i-1}]$. Suppose that the deviation sequence $\{\Xi_i - \xi_i\}$ forms a martingale difference sequence, and further assume the variance is uniformly bounded as*

$$\mathbb{E}[\|\Xi_i - \xi_i\|^2] \leq \sigma^2.$$

*Then the second moment of the sum admits the following bound:*

$$\mathbb{E}\left[\left\|\frac{1}{\tau}\sum_{i=1}^\tau \Xi_i\right\|^2\right] \leq 2\left\|\frac{1}{\tau}\sum_{i=1}^\tau \xi_i\right\|^2 + \frac{2\sigma^2}{\tau}.$$

This lemma is a powerful tool for analyzing stochastic processes with dependencies. In the context of FL with multiple local steps, the sequence of stochastic gradients $\{g_{s,i}(x_{s,i}^{t,k})\}_{k=0}^{K-1}$ computed on a client does not consist of independent random variables, because $x_{s,i}^{t,k+1}$ depends on $g_{s,i}(x_{s,i}^{t,k})$. It allows us to bound the variance of the average of these dependent stochastic gradients, which is crucial for analyzing the drift induced by local SGD steps.

**Lemma B.6** (Uniform Boundedness of Aggregated Updates). *Under Assumptions 4.2, 4.3 and 4.5, the expected squared norm of the aggregated update $\bar{\Delta}_s^{t+1}$ for any task $s \in [S]$ is uniformly bounded almost surely:*

$$\|\bar{\Delta}_s^{t+1}\| \leq G + \sigma_D + \sigma_G := M.$$

*Proof.* Under Assumptions 4.3 (Bounded Global Gradient, $\|\nabla f_s(x)\| \leq G$) and 4.5 (Bounded Heterogeneity $\|\nabla f_{s,i}(x) - \nabla f_s(x)\| \leq \sigma_D$, and Bounded Noise $\|g_{s,i}(x) - \nabla f_{s,i}(x)\| \leq \sigma_G$), we can bound the norm of any local stochastic gradient $g_{s,i}(x)$ almost surely:

$$\begin{aligned}
\|g_{s,i}(x)\| &= \|g_{s,i}(x) - \nabla f_{s,i}(x) + \nabla f_{s,i}(x) - \nabla f_s(x) + \nabla f_s(x)\| \\
&\leq \|g_{s,i}(x) - \nabla f_{s,i}(x)\| + \|\nabla f_{s,i}(x) - \nabla f_s(x)\| + \|\nabla f_s(x)\| \\
&\leq G + \sigma_D + \sigma_G.
\end{aligned}$$

Since the Euclidean norm is convex, the norm of an average is less than or equal to the average of the norms. Using (8d) and (8e), we have

$$\|c_s^{t+1}\| = \left\|\frac{1}{|\mathcal{R}_s|K}\sum_{i \in \mathcal{R}_s}\sum_{k=0}^{K-1} g_{s,i}(x_{s,i}^{t,k})\right\| \leq \frac{1}{|\mathcal{R}_s|K}\sum_{i \in \mathcal{R}_s}\sum_{k=0}^{K-1}\|g_{s,i}(x_{s,i}^{t,k})\| \leq G + \sigma_D + \sigma_G.$$

Substituting this back into (8c):

$$\|\bar{\Delta}_s^{t+1}\| \leq \gamma(G + \sigma_D + \sigma_G) + (1 - \gamma)\|\bar{\Delta}_s^t\|.$$

Unrolling this recursion from $t$ down to $0$ ($\bar{\Delta}_s^0 = \mathbf{0}$):

$$\|\bar{\Delta}_s^{t+1}\| \leq \gamma(G + \sigma_D + \sigma_G) \sum_{j=0}^{t} (1 - \gamma)^j = \gamma(G + \sigma_D + \sigma_G) \cdot \frac{1 - (1 - \gamma)^{t+1}}{\gamma} \leq G + \sigma_D + \sigma_G,$$

which completes the proof. $\qquad\square$

This lemma establishes a uniform almost-sure bound on the aggregated updates, which is pivotal for the subsequent analysis for Step-Update strategy.

**Lemma B.7** (Lipschitz Continuity of Regularized Weights). *Let* $\boldsymbol{\lambda}_\rho^*(\mathbf{G}) = \arg\min_{\boldsymbol{\lambda} \in \boldsymbol{\omega}} \{ J_\rho(\boldsymbol{\lambda}; \mathbf{G}) = \frac{1}{2} \|\mathbf{G}\mathbf{A}^\top \boldsymbol{\lambda}\|^2 + \frac{\rho}{2} \|\boldsymbol{\lambda}\|^2 = \frac{1}{2} \boldsymbol{\lambda}^\top [\mathbf{A}\mathbf{G}^\top \mathbf{G}\mathbf{A}^\top + \rho\mathbf{I}]\boldsymbol{\lambda} \}$ *be the optimal weight vector for a given matrix* $\mathbf{G} \in \mathbb{R}^{p \times S}$, *where* $\rho > 0$ *is the regularization parameter. Then, for any two matrices* $\mathbf{G}_1, \mathbf{G}_2$, *the following inequality holds:*

$$\|\boldsymbol{\lambda}_\rho^*(\mathbf{G}_1) - \boldsymbol{\lambda}_\rho^*(\mathbf{G}_2)\| \leq \frac{(\|\mathbf{G}_1\|_2 + \|\mathbf{G}_2\|_2)\|\mathbf{A}\|_2^2}{\rho} \|\mathbf{G}_1 - \mathbf{G}_2\|_2.$$

*Proof.* Let $\boldsymbol{\lambda}_1 = \boldsymbol{\lambda}_\rho^*(\mathbf{G}_1)$ and $\boldsymbol{\lambda}_2 = \boldsymbol{\lambda}_\rho^*(\mathbf{G}_2)$. By the strong convexity of the objective function (with modulus $\rho$), we have the following variational inequalities for the optimal solutions:

$$\langle \nabla J_\rho(\boldsymbol{\lambda}_1; \mathbf{G}_1) = (\mathbf{A}\mathbf{G}_1^\top \mathbf{G}_1 \mathbf{A}^\top + \rho\mathbf{I})\boldsymbol{\lambda}_1, \boldsymbol{\lambda}_2 - \boldsymbol{\lambda}_1 \rangle \geq 0.$$

$$\langle \nabla J_\rho(\boldsymbol{\lambda}_2; \mathbf{G}_2) = (\mathbf{A}\mathbf{G}_2^\top \mathbf{G}_2 \mathbf{A}^\top + \rho\mathbf{I})\boldsymbol{\lambda}_2, \boldsymbol{\lambda}_1 - \boldsymbol{\lambda}_2 \rangle \geq 0.$$

Summing these two inequalities:

$$\langle (\mathbf{A}\mathbf{G}_2^\top \mathbf{G}_2 \mathbf{A}^\top + \rho\mathbf{I})\boldsymbol{\lambda}_2 - (\mathbf{A}\mathbf{G}_1^\top \mathbf{G}_1 \mathbf{A}^\top + \rho\mathbf{I})\boldsymbol{\lambda}_1, \boldsymbol{\lambda}_1 - \boldsymbol{\lambda}_2 \rangle$$
$$= \langle (\mathbf{A}\mathbf{G}_2^\top \mathbf{G}_2 \mathbf{A}^\top \boldsymbol{\lambda}_2 - \mathbf{A}\mathbf{G}_1^\top \mathbf{G}_1 \mathbf{A}^\top \boldsymbol{\lambda}_1) - \rho(\boldsymbol{\lambda}_1 - \boldsymbol{\lambda}_2), \boldsymbol{\lambda}_1 - \boldsymbol{\lambda}_2 \rangle \geq 0.$$

Rearranging terms to isolate $\rho\|\boldsymbol{\lambda}_1 - \boldsymbol{\lambda}_2\|^2$:

$$\rho\|\boldsymbol{\lambda}_1 - \boldsymbol{\lambda}_2\|^2 \leq \langle \mathbf{A}\mathbf{G}_2^\top \mathbf{G}_2 \mathbf{A}^\top \boldsymbol{\lambda}_2 - \mathbf{A}\mathbf{G}_1^\top \mathbf{G}_1 \mathbf{A}^\top \boldsymbol{\lambda}_1, \boldsymbol{\lambda}_1 - \boldsymbol{\lambda}_2 \rangle$$
$$= \langle \mathbf{A}\mathbf{G}_2^\top \mathbf{G}_2 \mathbf{A}^\top (\boldsymbol{\lambda}_2 - \boldsymbol{\lambda}_1) + (\mathbf{A}\mathbf{G}_2^\top \mathbf{G}_2 \mathbf{A}^\top - \mathbf{A}\mathbf{G}_1^\top \mathbf{G}_1 \mathbf{A}^\top)\boldsymbol{\lambda}_1, \boldsymbol{\lambda}_1 - \boldsymbol{\lambda}_2 \rangle$$
$$= -\underbrace{\|\boldsymbol{\lambda}_1 - \boldsymbol{\lambda}_2\|_{\mathbf{A}\mathbf{G}_2^\top \mathbf{G}_2 \mathbf{A}^\top}^2}_{\geq 0} + \underbrace{\langle (\mathbf{A}\mathbf{G}_2^\top \mathbf{G}_2 \mathbf{A}^\top - \mathbf{A}\mathbf{G}_1^\top \mathbf{G}_1 \mathbf{A}^\top)\boldsymbol{\lambda}_1, \boldsymbol{\lambda}_1 - \boldsymbol{\lambda}_2 \rangle}_{\leq \|(\mathbf{A}\mathbf{G}_2^\top \mathbf{G}_2 \mathbf{A}^\top - \mathbf{A}\mathbf{G}_1^\top \mathbf{G}_1 \mathbf{A}^\top)\boldsymbol{\lambda}_1\| \|\boldsymbol{\lambda}_1 - \boldsymbol{\lambda}_2\|}$$
$$\leq \|\mathbf{A}(\mathbf{G}_2^\top \mathbf{G}_2 - \mathbf{G}_1^\top \mathbf{G}_1)\mathbf{A}^\top\|_2 \underbrace{\|\boldsymbol{\lambda}_1\|}_{\leq 1, \text{ since } \boldsymbol{\lambda}_1 \in \boldsymbol{\omega}} \|\boldsymbol{\lambda}_1 - \boldsymbol{\lambda}_2\|$$
$$\leq \|\mathbf{A}(\mathbf{G}_2^\top \mathbf{G}_2 - \mathbf{G}_1^\top \mathbf{G}_1)\mathbf{A}^\top\|_2 \|\boldsymbol{\lambda}_1 - \boldsymbol{\lambda}_2\|,$$

where the second inequality holds by using Cauchy-Schwarz inequality and the positive semi-definiteness of matrix $\mathbf{A}\mathbf{G}_2^\top \mathbf{G}_2 \mathbf{A}^\top$. Dividing both sides of this inequality by $\rho\|\boldsymbol{\lambda}_1 - \boldsymbol{\lambda}_2\|$:

$$\|\boldsymbol{\lambda}_1 - \boldsymbol{\lambda}_2\| \leq \frac{1}{\rho} \|\mathbf{A}(\mathbf{G}_2^\top \mathbf{G}_2 - \mathbf{G}_1^\top \mathbf{G}_1)\mathbf{A}^\top\|_2$$
$$= \frac{1}{\rho} \|\mathbf{A}[\mathbf{G}_2^\top(\mathbf{G}_2 - \mathbf{G}_1) + (\mathbf{G}_2^\top - \mathbf{G}_1^\top)\mathbf{G}_1]\mathbf{A}^\top\|_2$$
$$= \frac{1}{\rho} \|\mathbf{A}\mathbf{G}_2^\top(\mathbf{G}_2 - \mathbf{G}_1)\mathbf{A}^\top + \mathbf{A}(\mathbf{G}_2^\top - \mathbf{G}_1^\top)\mathbf{G}_1\mathbf{A}^\top\|_2$$
$$\leq \frac{1}{\rho} \|\mathbf{A}\mathbf{G}_2^\top(\mathbf{G}_2 - \mathbf{G}_1)\mathbf{A}^\top\|_2 + \frac{1}{\rho} \|\mathbf{A}(\mathbf{G}_2^\top - \mathbf{G}_1^\top)\mathbf{G}_1\mathbf{A}^\top\|_2$$
$$\leq \frac{\|\mathbf{A}\mathbf{G}_1^\top\|_2 + \|\mathbf{A}\mathbf{G}_2^\top\|_2}{\rho} \|\mathbf{A}(\mathbf{G}_1^\top - \mathbf{G}_2^\top)\|_2 \leq \frac{(\|\mathbf{G}_1\|_2 + \|\mathbf{G}_2\|_2)\|\mathbf{A}\|_2^2}{\rho} \|\mathbf{G}_1 - \mathbf{G}_2\|_2,$$

where the inequality follows from the triangle inequality $\|A + B\|_2 \leq \|A\|_2 + \|B\|_2$, the sub-multiplicative property of the spectral norm $\|AB\|_2 \leq \|A\|_2 \|B\|_2$, and the fact that $\|A^\top\|_2 = \|A\|_2$. This completes the proof. $\qquad\square$

This lemma characterizes the sensitivity of the optimal weights to perturbations in the input gradients. The Lipschitz constant is proportional to $1/\rho$, highlighting the trade-off introduced by the regularization term: a larger $\rho$ stabilizes the weights but introduces a larger regularization bias.

**Lemma B.8** (Regularization Bias Bound). *Let* $\boldsymbol{\lambda}_\rho^*(\mathbf{G}) = \arg\min_{\boldsymbol{\lambda}\in\boldsymbol{\omega}}\{J_\rho(\boldsymbol{\lambda};\mathbf{G}) = \frac{1}{2}\|\mathbf{G}\mathbf{A}^\top\boldsymbol{\lambda}\|^2 + \frac{\rho}{2}\|\boldsymbol{\lambda}\|^2 = \frac{1}{2}\boldsymbol{\lambda}^\top[\mathbf{A}\mathbf{G}^\top\mathbf{G}\mathbf{A}^\top + \rho\mathbf{I}]\boldsymbol{\lambda}\}$ *be the optimal weight vector for a given matrix* $\mathbf{G} \in \mathbb{R}^{p\times S}$, *where* $\rho \geq 0$. *Then, the squared norm of the regularized descent direction is bounded by the unregularized counterpart as follows:*

$$\left\|\mathbf{G}\mathbf{A}^\top\boldsymbol{\lambda}_\rho^*(\mathbf{G})\right\|^2 \leq \left\|\mathbf{G}\mathbf{A}^\top\boldsymbol{\lambda}_0^*(\mathbf{G})\right\|^2 + \rho\left[\|\boldsymbol{\lambda}_0^*(\mathbf{G})\|^2 - \|\boldsymbol{\lambda}_\rho^*(\mathbf{G})\|^2\right] \leq \left\|\mathbf{G}\mathbf{A}^\top\boldsymbol{\lambda}_0^*(\mathbf{G})\right\|^2 + \rho\left(1 - \frac{1}{S}\right).$$

*Proof.* For notation simplicity, we denote $\boldsymbol{\lambda}_\rho^* = \boldsymbol{\lambda}_\rho^*(\mathbf{G})$ and $\boldsymbol{\lambda}_0^* = \boldsymbol{\lambda}_0^*(\mathbf{G})$. Since $\boldsymbol{\lambda}_\rho^*$ is the global minimizer of $J_\rho(\boldsymbol{\lambda};\mathbf{G})$ over the probability simplex $\boldsymbol{\omega}$, and $\boldsymbol{\lambda}_0^*$ is also a feasible point, we have the following inequality based on optimality:

$$J_\rho(\boldsymbol{\lambda}_\rho^*;\mathbf{G}) = \frac{1}{2}\|\mathbf{G}\mathbf{A}^\top\boldsymbol{\lambda}_\rho^*\|^2 + \frac{\rho}{2}\|\boldsymbol{\lambda}_\rho^*\|^2 \leq J_\rho(\boldsymbol{\lambda}_0^*;\mathbf{G}) = \frac{1}{2}\|\mathbf{G}\mathbf{A}^\top\boldsymbol{\lambda}_0^*\|^2 + \frac{\rho}{2}\|\boldsymbol{\lambda}_0^*\|^2.$$

We are interested in bounding the first term on the left-hand side. Rearranging the inequality and multiplying both sides by 2 yields the claimed inequality:

$$\|\mathbf{G}\mathbf{A}^\top\boldsymbol{\lambda}_\rho^*\|^2 \leq \|\mathbf{G}\mathbf{A}^\top\boldsymbol{\lambda}_0^*\|^2 + \rho(\|\boldsymbol{\lambda}_0^*\|^2 - \|\boldsymbol{\lambda}_\rho^*\|^2).$$

Since $\boldsymbol{\lambda} \in \boldsymbol{\omega} = \{\boldsymbol{\lambda} \in \mathbb{R}^S \mid \lambda_s \geq 0, \sum_{s\in[S]}\lambda_s = 1\}$, the Euclidean norm $\|\boldsymbol{\lambda}\|^2$ is bounded.

- Upper bound: Attained at vertices (e.g., $[1,0,\ldots,0]$), where $\|\boldsymbol{\lambda}\|^2 = 1$.

- Lower bound: Attained at the center (i.e., $[1/S,\ldots,1/S]$), where $\|\boldsymbol{\lambda}\|^2 = 1/S$.

Therefore, the difference is strictly bounded:

$$\|\boldsymbol{\lambda}_0^*\|^2 - \|\boldsymbol{\lambda}_\rho^*\|^2 \leq \max_{\boldsymbol{\lambda}\in\boldsymbol{\omega}}\|\boldsymbol{\lambda}\|^2 - \min_{\boldsymbol{\lambda}\in\boldsymbol{\omega}}\|\boldsymbol{\lambda}\|^2 = 1 - \frac{1}{S}.$$

This concludes the proof. $\qquad\square$

This bound quantifies the systematic error introduced by solving the regularized subproblem instead of the original one. It confirms that the regularization bias vanishes as $\rho \to 0$. In the convergence analysis, this term appears as an irreducible error floor, necessitating a careful choice of $\rho$ to balance the approximation accuracy with the weight stability.

## C. Analysis of Gradient Estimation Error and Client Drift

This section provides the detailed proofs for bounding the stochastic error terms arising from the federated setting. We first establish the recursive inequalities for $\mathbb{E}[\mathcal{E}_s^t]$ and $\mathbb{E}[\mathcal{C}_s^t]$, then derive the explicit bound for the time-averaged estimation error by unrolling these coupled recursive relations.

### C.1. Proof of Lemma C.1

**Lemma C.1** (Recursive Bound for Estimation Error). *Under Assumptions 4.1 and 4.2, for the sequence generated by DREAM, the expected gradient estimation error evolves according to the following inequality:*

$$\mathbb{E}\left[\mathcal{E}_s^{t+1}\right] \leq \left[1 - \frac{\gamma(1+\gamma)}{2}\right]\mathbb{E}\left[\mathcal{E}_s^t\right] + \gamma(1+\gamma)L^2\mathbb{E}\left[\mathcal{C}_s^{t+1}\right] + \left[\frac{2}{\gamma} - (1+\gamma)\right]\eta_g^2L^2\mathbb{E}\|d^{t+1}\|^2 + \frac{2\gamma^2\sigma^2}{K|\mathcal{R}_s|}. \tag{13}$$

*Furthermore, the initial error is bounded by:* $\mathbb{E}\left[\mathcal{E}_s^0\right] \leq (1-\gamma)\|\nabla f_s(x^0)\|^2 + \gamma(1+\gamma)L^2\mathbb{E}\left[\mathcal{C}_s^0\right] + \frac{2\gamma^2\sigma^2}{K|\mathcal{R}_s|}.$

*Proof.* The objective of this proof is to establish a recursive bound for the gradient estimation error. The core strategy is to decompose the error term, bound its constituent parts, and then reassemble these bounds into the final recursive form.

We begin by expanding the definition of $\bar{\Delta}_s^{t+1}$ using its formulation in (8c). This allows us to separate the influence of the momentum term $\bar{\Delta}_s^t$ from the newly computed control variate $c_s^{t+1}$:

$$\nabla f_s(x^t) - \bar{\Delta}_s^{t+1} \overset{(8c)}{=} \gamma[\nabla f_s(x^t) - c_s^{t+1}] + (1-\gamma)[\nabla f_s(x^t) - \bar{\Delta}_s^t]$$

$$\overset{(8d),(8e)}{=} \frac{\gamma}{K|\mathcal{R}_s|} \sum_{i \in \mathcal{R}_s} \sum_{k=0}^{K-1} \left[\nabla f_{s,i}(x^t) - g_{s,i}(x_{s,i}^{t,k})\right] + (1-\gamma)[\nabla f_s(x^t) - \bar{\Delta}_s^t].$$

We then expand the squared norm of the above expression:

$$\mathbb{E}\left[\mathcal{E}_s^t\right] = \mathbb{E}\|\nabla f_s(x^t) - \bar{\Delta}_s^{t+1}\|^2$$

$$= 2\gamma(1-\gamma)\mathbb{E}\left\langle \frac{1}{K|\mathcal{R}_s|} \sum_{i \in \mathcal{R}_s} \sum_{k=0}^{K-1} [\nabla f_{s,i}(x^t) - \nabla f_{s,i}(x_{s,i}^{t,k})], \nabla f_s(x^t) - \bar{\Delta}_s^t \right\rangle$$

$$+ \gamma^2 \mathbb{E}\left\| \frac{1}{K|\mathcal{R}_s|} \sum_{i \in \mathcal{R}_s} \sum_{k=0}^{K-1} [\nabla f_{s,i}(x^t) - g_{s,i}(x_{s,i}^{t,k})] \right\|^2 + (1-\gamma)^2 \mathbb{E}\left\| \nabla f_s(x^t) - \bar{\Delta}_s^t \right\|^2.$$

Since $c_s^{t+1}$ is computed from stochastic gradients in round $t$, while $\nabla f_s(x^t) - \bar{\Delta}_s^t$ is determined by the history up to round $t-1$, we can first take the expectation over the randomness of round $t$. This allows replacing the stochastic term $c_s^{t+1}$ with its expectation inside the inner product.

To proceed, we bound each of these terms. First, the cross-term is managed by applying Young's inequality (9b) and Jensen's inequality (10b). This allows us to decouple the inner product:

$$2 \cdot \left\langle \frac{1}{K|\mathcal{R}_s|} \sum_{i \in \mathcal{R}_s} \sum_{k=0}^{K-1} [\nabla f_{s,i}(x^t) - \nabla f_{s,i}(x_{s,i}^{t,k})], \nabla f_s(x^t) - \bar{\Delta}_s^t \right\rangle$$

$$\overset{(9b)}{\leq} \left\|\nabla f_s(x^t) - \bar{\Delta}_s^t\right\|^2 + \left\| \frac{1}{K|\mathcal{R}_s|} \sum_{i \in \mathcal{R}_s} \sum_{k=0}^{K-1} [\nabla f_{s,i}(x^t) - \nabla f_{s,i}(x_{s,i}^{t,k})] \right\|^2$$

$$\overset{(10b)}{\leq} \left\|\nabla f_s(x^t) - \bar{\Delta}_s^t\right\|^2 + \frac{1}{K|\mathcal{R}_s|} \sum_{i \in \mathcal{R}_s} \sum_{k=0}^{K-1} \underbrace{\left\|\nabla f_{s,i}(x^t) - \nabla f_{s,i}(x_{s,i}^{t,k})\right\|^2}_{\overset{(11a)}{\leq} L^2\|x^t - x_{s,i}^{t,k}\|^2}$$

$$\leq \left\|\nabla f_s(x^t) - \bar{\Delta}_s^t\right\|^2 + L^2 \mathcal{C}_s^t.$$

Next, we bound the second variance term, which encapsulates the error introduced by the current round's updates. This error stems from two sources: the deviation of local models from the global model (client drift) and the inherent noise of stochastic gradients. To analyze this, the term is decomposed by adding and subtracting the true local gradients $\nabla f_{s,i}(x_{s,i}^{t,k})$:

$$\mathbb{E}\left\| \frac{1}{K|\mathcal{R}_s|} \sum_{i \in \mathcal{R}_s} \sum_{k=0}^{K-1} [\nabla f_{s,i}(x^t) - g_{s,i}(x_{s,i}^{t,k})] \right\|^2$$

$$= \mathbb{E}\left\| \frac{1}{K|\mathcal{R}_s|} \sum_{i \in \mathcal{R}_s} \sum_{k=0}^{K-1} [\nabla f_{s,i}(x^t) - \nabla f_{s,i}(x_{s,i}^{t,k}) + \nabla f_{s,i}(x_{s,i}^{t,k}) - g_{s,i}(x_{s,i}^{t,k})] \right\|^2$$

$$\overset{\text{Lemma}B.5}{\leq} 2\mathbb{E}\left\| \frac{1}{K|\mathcal{R}_s|} \sum_{i \in \mathcal{R}_s} \sum_{k=0}^{K-1} [\nabla f_{s,i}(x^t) - \nabla f_{s,i}(x_{s,i}^{t,k})] \right\|^2 + \frac{2\sigma^2}{K|\mathcal{R}_s|} \overset{(10b)}{\leq} 2L^2 \mathbb{E}\left[\mathcal{C}_s^t\right] + \frac{2\sigma^2}{K|\mathcal{R}_s|}.$$

Substituting these derived bounds back into the initial expansion of $\mathbb{E}\left[\mathcal{E}_s^t\right]$ yields an intermediate inequality:

$$\mathbb{E}\left[\mathcal{E}_s^t\right] \leq (1-\gamma)\mathbb{E}\left\|\nabla f_s(x^t) - \bar{\Delta}_s^t\right\|^2 + \gamma(1+\gamma)L^2\mathbb{E}\left[\mathcal{C}_s^t\right] + \frac{2\gamma^2\sigma^2}{K|\mathcal{R}_s|}. \tag{14}$$

The crucial step to establish the recursive connection to the previous round's error is to handle the first term on the right-hand side of (14). This is achieved by adding and subtracting $\nabla f_s(x^{t-1})$ inside the norm and applying Young's inequality (9a) with $\eta = \gamma/2$, which yields

$$(1-\gamma)\mathbb{E}\left\|\nabla f_s(x^t) - \bar{\Delta}_s^t\right\|^2$$

$$=(1-\gamma)\mathbb{E}\left\|\nabla f_s(x^{t-1}) - \bar{\Delta}_s^t + \nabla f_s(x^t) - \nabla f_s(x^{t-1})\right\|^2$$

$$\overset{(9a)}{\leq}(1-\gamma)\left(1+\frac{\gamma}{2}\right)\mathbb{E}\left\|\nabla f_s(x^{t-1}) - \bar{\Delta}_s^t\right\|^2 + (1-\gamma)\left(1+\frac{2}{\gamma}\right)\mathbb{E}\underbrace{\left\|\nabla f_s(x^t) - \nabla f_s(x^{t-1})\right\|^2}_{\leq L^2\|x^t-x^{t-1}\|^2 \overset{(8h)}{=} \eta_g^2 L^2\|d^t\|^2}$$

$$\leq\frac{2-\gamma-\gamma^2}{2}\mathbb{E}\left[\mathcal{E}_s^{t-1}\right] + \frac{2-\gamma-\gamma^2}{\gamma}\eta_g^2 L^2\mathbb{E}\|d^t\|^2.$$

Finally, substituting this recursive bound back into (14) yields the claimed result (13). To obtain the specific bound for the initial error $\mathcal{E}_s^0$, one can apply the result of (14) at $t = 0$. $\qquad\square$

## C.2. Proof of Lemma C.2

**Lemma C.2** (Recursive Bound for Client Drift). *Under Assumptions 4.1 and 4.2, for the sequence generated by DREAM, if $48\gamma(K-1)K\eta_l^2 L^2 \leq 1$ holds, the client drift is governed by the following recursive relation:*

$$\mathbb{E}\left[\mathcal{C}_s^{t+1}\right] - 96\gamma\eta_l^2 K^2 L^2\mathbb{E}\left[\mathcal{C}_s^t\right]$$

$$\leq 96\gamma\eta_g^2\eta_l^2 K^2 L^2\mathbb{E}\|d^{t+1}\|^2 + 6(1-\gamma)\eta_l^2 K^2\mathbb{E}\left[\mathcal{E}_s^t\right] + 6\gamma\eta_l^2 K^2\sigma^2\left(1+\frac{16}{K}\right) + 48\eta_l^2 K^2\mathbb{E}\|\nabla f_s(x^t)\|^2. \qquad (15)$$

*Furthermore, the initial client drift is bounded by:* $\mathbb{E}\left[\mathcal{C}_s^0\right] \leq 6\gamma\eta_l^2 K^2\left[\sigma^2 + \frac{8}{|\mathcal{R}_s|}\sum_{i\in\mathcal{R}_s}\mathbb{E}\|\nabla f_{s,i}(x^0)\|^2\right].$

*Proof.* Recall the local update rule for $x_{s,i}^{t,k}$ in (8a). By unrolling the iteration, we have

$$x_{s,i}^{t,k} - x_{s,i}^{t,0} \overset{(8a)}{=} x_{s,i}^{t,k-1} - x_{s,i}^{t,0} - \eta_l\left[\gamma g_{s,i}(x_{s,i}^{t,k-1}) + \gamma(c_s^t - c_{s,i}^t) + (1-\gamma)\bar{\Delta}_s^t\right].$$

By taking the squared norm and applying Young's inequality (9a) with $\eta = 1/(K-1)$, we can bound the deviation at step $k$ by the deviation at step $k-1$ and the norm of the update term:

$$\mathbb{E}\|x_{s,i}^{t,k} - x_{s,i}^{t,0}\|^2 \overset{(9a)}{\leq} \frac{K}{K-1}\mathbb{E}\left\|x_{s,i}^{t,k-1} - x_{s,i}^{t,0}\right\|^2 + K\eta_l^2\mathbb{E}\|\gamma g_{s,i}(x_{s,i}^{t,k-1}) + \gamma(c_s^t - c_{s,i}^t) + (1-\gamma)\bar{\Delta}_s^t\|^2$$

$$\overset{(10b)}{\leq} \frac{K}{K-1}\mathbb{E}\|x_{s,i}^{t,k-1} - x_{s,i}^{t,0}\|^2 + \gamma K\eta_l^2\mathbb{E}\|g_{s,i}(x_{s,i}^{t,k-1}) + c_s^t - c_{s,i}^t\|^2 + (1-\gamma)K\eta_l^2\mathbb{E}\|\bar{\Delta}_s^t\|^2.$$

To further bound the second term, which contains the drift-corrected local stochastic gradient, we apply Assumptions 4.1 ($L$-smoothness) and 4.2 (bounded variance). This allows us to separate the effects of the true gradient and the stochastic noise:

$$\mathbb{E}\|g_{s,i}(x_{s,i}^{t,k-1}) + c_s^t - c_{s,i}^t\|^2 \leq 2\mathbb{E}\|\nabla f_{s,i}(x_{s,i}^{t,k-1}) + c_s^t - c_{s,i}^t\|^2 + 2\sigma^2$$

$$\leq 4\mathbb{E}\|\nabla f_{s,i}(x_{s,i}^{t,k-1}) - \nabla f_{s,i}(x_{s,i}^{t,0})\|^2 + 4\mathbb{E}\|\nabla f_{s,i}(x_{s,i}^{t,0}) + c_s^t - c_{s,i}^t\|^2 + 2\sigma^2$$

$$\leq 4L^2\mathbb{E}\|x_{s,i}^{t,k-1} - x_{s,i}^{t,0}\|^2 + 4\mathbb{E}\|\nabla f_{s,i}(x_{s,i}^{t,0}) + c_s^t - c_{s,i}^t\|^2 + 2\sigma^2.$$

Substituting this bound back into the one-step inequality yields a recursive relation for the local model's deviation:

$$\mathbb{E}\|x_{s,i}^{t,k} - x_{s,i}^{t,0}\|^2 \leq \left(\frac{K}{K-1} + 4\gamma K\eta_l^2 L^2\right)\mathbb{E}\|x_{s,i}^{t,k-1} - x_{s,i}^{t,0}\|^2$$

$$+ 4\gamma K\eta_l^2\mathbb{E}\|\nabla f_{s,i}(x_{s,i}^{t,0}) + c_s^t - c_{s,i}^t\|^2 + (1-\gamma)K\eta_l^2\mathbb{E}\|\bar{\Delta}_s^t\|^2 + 2\gamma K\eta_l^2\sigma^2.$$

By recursively applying the above one-step inequality from $r = 0$ to $k - 1$, we further have

$$\mathbb{E}\|x_{s,i}^{t,k} - x_{s,i}^{t,0}\|^2 \leq K\eta_l^2 \sum_{r=0}^{k-1} \left(\frac{K}{K-1} + 4\gamma K\eta_l^2 L^2\right)^r \left[4\gamma\mathbb{E}\|\nabla f_{s,i}(x_{s,i}^{t,0}) + c_s^t - c_{s,i}^t\|^2 + (1-\gamma)\mathbb{E}\|\bar{\Delta}_s^t\|^2 + 2\gamma\sigma^2\right]$$

$$\leq \eta_l^2 K^2 \left(\frac{K}{K-1} + 4\gamma K\eta_l^2 L^2\right)^{K-1} \left[4\gamma\mathbb{E}\|\nabla f_{s,i}(x_{s,i}^{t,0}) + c_s^t - c_{s,i}^t\|^2 + (1-\gamma)\mathbb{E}\|\bar{\Delta}_s^t\|^2 + 2\gamma\sigma^2\right]$$

$$\leq 12\gamma\eta_l^2 K^2 \mathbb{E}\|\nabla f_{s,i}(x_{s,i}^{t,0}) + c_s^t - c_{s,i}^t\|^2 + 3(1-\gamma)\eta_l^2 K^2 \mathbb{E}\|\bar{\Delta}_s^t\|^2 + 6\gamma\eta_l^2 K^2\sigma^2$$

$$\leq 24\gamma\eta_l^2 K^2 \mathbb{E}\|\nabla f_{s,i}(x_{s,i}^{t,0}) - \nabla f_s(x_s^t) + c_s^t - c_{s,i}^t\|^2 + 24\gamma\eta_l^2 K^2 \mathbb{E}\|\nabla f_s(x^t)\|^2$$
$$+ 3(1-\gamma)\eta_l^2 K^2 \mathbb{E}\|\bar{\Delta}_s^t\|^2 + 6\gamma\eta_l^2 K^2\sigma^2,$$

where the third inequality holds under the condition $4\gamma K\eta_l^2 L^2 \leq \frac{1}{12(K-1)}$, since $\lim_{x\to\infty}(1 + \frac{1}{x} + \frac{1}{12x})^x = e^{1+\frac{1}{12}} < 3$. Averaging this bound over all participating clients $i \in \mathcal{R}_s$ and applying inequality (10c) gives a bound on the average client deviation:

$$\frac{1}{|\mathcal{R}_s|} \sum_{i \in \mathcal{R}_s} \mathbb{E}\|x_{s,i}^{t,k} - x_{s,i}^{t,0}\|^2 \overset{(10c)}{\leq} \frac{24\gamma\eta_l^2 K^2}{|\mathcal{R}_s|} \sum_{i \in \mathcal{R}_s} \mathbb{E}\|\nabla f_{s,i}(x_{s,i}^{t,0}) - c_{s,i}^t\|^2$$
$$+ 3(1-\gamma)\eta_l^2 K^2 \mathbb{E}\|\bar{\Delta}_s^t\|^2 + 6\gamma\eta_l^2 K^2 \left(4\mathbb{E}\|\nabla f_s(x^t)\|^2 + \sigma^2\right). \tag{16}$$

By applying Lemma B.5, together with the bounded variance assumption and the smoothness of $f_{s,i}$, this bound follows directly:

$$\frac{1}{|\mathcal{R}_s|} \sum_{i \in \mathcal{R}_s} \mathbb{E}\|\nabla f_{s,i}(x_{s,i}^{t+1,0}) - c_{s,i}^{t+1}\|^2 \overset{(8d)}{=} \frac{1}{|\mathcal{R}_s|} \sum_{i \in \mathcal{R}_s} \mathbb{E}\left\|\nabla f_{s,i}(x_{s,i}^{t+1,0}) - \frac{1}{K}\sum_{k=0}^{K-1} g_{s,i}(x_{s,i}^{t,k})\right\|^2$$

$$\leq \frac{2}{|\mathcal{R}_s|} \sum_{i \in \mathcal{R}_s} \mathbb{E}\left\|\nabla f_{s,i}(x_{s,i}^{t,0}) - \frac{1}{K}\sum_{k=0}^{K-1} g_{s,i}(x_{s,i}^{t,k})\right\|^2 + \frac{2}{|\mathcal{R}_s|} \sum_{i \in \mathcal{R}_s} \mathbb{E}\|\nabla f_{s,i}(x_{s,i}^{t+1,0}) - \nabla f_{s,i}(x_{s,i}^{t,0})\|^2$$

$$\overset{\text{Lemma} B.5}{\leq} \frac{4}{|\mathcal{R}_s|} \sum_{i \in \mathcal{R}_s} \mathbb{E}\left\|\frac{1}{K}\sum_{k=0}^{K-1}[\nabla f_{s,i}(x_{s,i}^{t,k}) - \nabla f_{s,i}(x_{s,i}^{t,0})]\right\|^2 + \frac{4\sigma^2}{K} + \frac{2L^2}{|\mathcal{R}_s|} \sum_{i \in \mathcal{R}_s} \mathbb{E}\|\underbrace{x^{t+1} - x^t}_{\overset{(8h)}{=}-d^{t+1}}\|^2$$

$$\leq 4L^2\mathbb{E}\left[\mathcal{C}_s^t\right] + 2\eta_g^2 L^2 \mathbb{E}\|d^{t+1}\|^2 + \frac{4\sigma^2}{K}.$$

Substituting the above bound into (16) at iteration $t+1$ and averaging over $k = 0, \cdots, K-1$, we obtain:

$$\mathbb{E}\left[\mathcal{C}_s^{t+1}\right] \leq 96\gamma\eta_l^2 K^2 L^2 \mathbb{E}\left[\mathcal{C}_s^t\right] + 48\gamma\eta_g^2 \eta_l^2 K^2 L^2 \mathbb{E}\|d^{t+1}\|^2 + 3(1-\gamma)\eta_l^2 K^2 \mathbb{E}\|\bar{\Delta}_s^{t+1}\|^2$$
$$+ 6\gamma\eta_l^2 K^2\sigma^2 \left(1 + \frac{16}{K}\right) + 24\gamma\eta_l^2 K^2 \mathbb{E}\|\nabla f_s(x^{t+1})\|^2.$$

Then, we use a standard decomposition and Young's inequality:

$$\mathbb{E}\|\bar{\Delta}_s^{t+1}\|^2 \leq 2\mathbb{E}\|\nabla f_s(x^t) - \bar{\Delta}_s^{t+1}\|^2 + 2\mathbb{E}\|\nabla f_s(x^t)\|^2 = 2\mathbb{E}\left[\mathcal{E}_s^t\right] + 2\mathbb{E}\|\nabla f_s(x^t)\|^2,$$

$$\mathbb{E}\|\nabla f_s(x^{t+1})\|^2 \leq 2\mathbb{E}\underbrace{\|\nabla f_s(x^{t+1}) - \nabla f_s(x^t)\|^2}_{\leq L^2\|x^{t+1}-x^t\|^2 \overset{(8h)}{=} \eta_g^2 L^2\|d^{t+1}\|^2} + 2\mathbb{E}\|\nabla f_s(x^t)\|^2 \leq 2\eta_g^2 L^2 \mathbb{E}\|d^{t+1}\|^2 + 2\mathbb{E}\|\nabla f_s(x^t)\|^2.$$

Substituting these two bounds back into the expression for $\mathbb{E}\left[\mathcal{C}_s^{t+1}\right]$ yields the final recursive inequality (15):

$$\mathbb{E}\left[\mathcal{C}_s^{t+1}\right] \leq 96\gamma\eta_l^2 K^2 L^2 \mathbb{E}\left[\mathcal{C}_s^t\right] + 96\gamma\eta_g^2 \eta_l^2 K^2 L^2 \mathbb{E}\|d^{t+1}\|^2 + 6(1-\gamma)\eta_l^2 K^2 \mathbb{E}\left[\mathcal{E}_s^t\right]$$
$$+ 6\gamma\eta_l^2 K^2\sigma^2 \left(1 + \frac{16}{K}\right) + \underbrace{(6 + 42\gamma)}_{\leq 48}\eta_l^2 K^2 \mathbb{E}\|\nabla f_s(x^t)\|^2.$$

Finally, to obtain a bound for the initial client drift, we can set $t = 0$ in (16). $\qquad\square$

These two lemmas formalize the error dynamics. They reveal a **potential vicious cycle** where errors could amplify each other: the gradient estimation error ($\mathcal{E}_s^t$) contributes to the client drift of the subsequent round ($\mathcal{C}_s^{t+1}$) (Lemma C.2), which in turn degrades future gradient estimates (Lemma C.1). However, a key counteracting force is also present: the momentum-induced contraction factor $(2 - \gamma - \gamma^2)/2$ in Lemma C.1. This factor, being strictly less than 1, acts as a powerful contraction factor. It demonstrates that the integrated "correct-then-smooth" mechanism provably dampens the propagation of gradient estimation error across rounds. While the client drift itself is also controlled under a contractive mapping in Lemma C.2 with sufficiently small step sizes, the momentum-induced contraction provides a more fundamental stabilization by directly suppressing the primary error source. The stability and convergence of the entire system hinge on this contraction overpowering the error amplification from the feedback loop.

## C.3. Proof of Proposition C.3

**Proposition C.3.** *Under Assumptions 4.1-4.3, by choosing parameters such that $48\eta_l^2 K^2 L^2 \leq \min\{\frac{1}{4\gamma}, \frac{1}{1-\gamma}\}$, the time-averaged sum of the gradient estimation errors is bounded:*

$$\frac{1}{T}\sum_{t=0}^{T-1}\mathbb{E}\left[\mathcal{E}_s^t\right] \leq \frac{4\mathbb{E}\left[\mathcal{E}_s^0\right]}{\gamma T} + \frac{4L^2\mathbb{E}\left[\mathcal{C}_s^0\right]}{T} + \frac{8\eta_g^2 L^2}{\gamma^2 T}\sum_{t=0}^{T-1}\mathbb{E}\|d^{t+1}\|^2$$
$$+ 48\eta_l^2 K^2 L^2\left[\gamma\sigma^2\left(1+\frac{16}{K}\right)+8G^2\right]+\frac{8\gamma\sigma^2}{K|\mathcal{R}_s|}. \tag{17}$$

*Proof.* We begin by averaging the recursive inequality (15) for client drift from Lemma C.2 over $t = 0, \cdots, T-1$:

$$\frac{1}{T}\sum_{t=0}^{T-1}\mathbb{E}\left[\mathcal{C}_s^{t+1}\right] - \frac{96\gamma\eta_l^2 K^2 L^2}{T}\sum_{t=0}^{T-1}\mathbb{E}\left[\mathcal{C}_s^t\right]$$

$$= \frac{1-96\gamma\eta_l^2 K^2 L^2}{T}\sum_{t=0}^{T-1}\mathbb{E}\left[\mathcal{C}_s^{t+1}\right] - \frac{96\gamma\eta_l^2 K^2 L^2}{T}\left[\mathbb{E}\left[\mathcal{C}_s^0\right] - \mathbb{E}\left[\mathcal{C}_s^T\right]\right]$$

$$\leq \frac{96\gamma\eta_g^2\eta_l^2 K^2 L^2}{T}\sum_{t=0}^{T-1}\mathbb{E}\|d^{t+1}\|^2 + \frac{6(1-\gamma)\eta_l^2 K^2}{T}\sum_{t=0}^{T-1}\mathbb{E}\left[\mathcal{E}_s^t\right] + \frac{48\eta_l^2 K^2}{T}\sum_{t=0}^{T-1}\mathbb{E}\|\nabla f_s(x^t)\|^2 + 6\gamma\eta_l^2 K^2\sigma^2\left(1+\frac{16}{K}\right).$$

This allows us to relate the total client drift to the total gradient estimation error. By choosing parameters such that $96\gamma\eta_l^2 K^2 L^2 \leq \frac{1}{2}$, we can absorb the term $\mathbb{E}\left[\mathcal{C}_s^t\right]$, yielding:

$$\frac{1}{T}\sum_{t=0}^{T-1}\mathbb{E}\left[\mathcal{C}_s^{t+1}\right] \leq \frac{\mathbb{E}\left[\mathcal{C}_s^0\right]}{T} + \frac{192\gamma\eta_g^2\eta_l^2 K^2 L^2}{T}\sum_{t=0}^{T-1}\mathbb{E}\|d^{t+1}\|^2 + \frac{12(1-\gamma)\eta_l^2 K^2}{T}\sum_{t=0}^{T-1}\mathbb{E}\left[\mathcal{E}_s^t\right]$$

$$+ \frac{96\eta_l^2 K^2}{T}\sum_{t=0}^{T-1}\mathbb{E}\|\nabla f_s(x^t)\|^2 + 12\gamma\eta_l^2 K^2\sigma^2\left(1+\frac{16}{K}\right).$$

Next, averaging (13) from Lemma C.1 over $t = 0, \cdots, T-1$ and substituting the above inequality gives:

$$\frac{1}{T}\sum_{t=0}^{T-1}\mathbb{E}\left[\mathcal{E}_s^{t+1}\right] \leq \frac{2-\gamma-\gamma^2}{2T}\sum_{t=0}^{T-1}\mathbb{E}\left[\mathcal{E}_s^t\right] + \frac{\gamma(1+\gamma)L^2}{T}\sum_{t=0}^{T-1}\mathbb{E}\left[\mathcal{C}_s^{t+1}\right]$$

$$+ \frac{2\eta_g^2 L^2}{\gamma T}\sum_{t=0}^{T-1}\mathbb{E}\|d^{t+1}\|^2 - \frac{(1+\gamma)\eta_g^2 L^2}{T}\sum_{t=0}^{T-1}\mathbb{E}\|d^{t+1}\|^2 + \frac{2\gamma^2\sigma^2}{K|\mathcal{R}_s|}$$

$$\leq \frac{2-\gamma-\gamma^2}{2T}\sum_{t=0}^{T-1}\mathbb{E}\left[\mathcal{E}_s^t\right] + \underbrace{12(1-\gamma)\eta_l^2 K^2 L^2}_{\leq\frac{1}{4}}\cdot\frac{\gamma(1+\gamma)}{T}\sum_{t=0}^{T-1}\mathbb{E}\left[\mathcal{E}_s^t\right] + \frac{\gamma(1+\gamma)L^2\mathbb{E}\left[\mathcal{C}_s^0\right]}{T}$$

$$+ \frac{2\eta_g^2 L^2}{\gamma T}\sum_{t=0}^{T-1}\mathbb{E}\|d^{t+1}\|^2 - \underbrace{(1-192\gamma^2\eta_l^2 K^2 L^2)}_{\geq 0}\cdot\frac{(1+\gamma)\eta_g^2 L^2}{T}\sum_{t=0}^{T-1}\mathbb{E}\|d^{t+1}\|^2$$

$$+ \frac{96\gamma(1+\gamma)\eta_l^2 K^2 L^2}{T} \sum_{t=0}^{T-1} \mathbb{E}\|\nabla f_s(x^t)\|^2 + 12(1+\gamma)\gamma^2\eta_l^2 K^2 L^2 \sigma^2 \left(1 + \frac{16}{K}\right) + \frac{2\gamma^2\sigma^2}{K|\mathcal{R}_s|}.$$

Set $192\gamma^2\eta_l^2 K^2 L^2 \leq 1$ and $48(1-\gamma)\eta_l^2 K^2 L^2 \leq 1$ to absorb some terms, we can obtain a new bound on the cumulative gradient estimation error after rearranging:

$$\frac{1}{T} \sum_{t=0}^{T-1} \mathbb{E}\left[\mathcal{E}_s^t\right] \leq \frac{4\mathbb{E}\left[\mathcal{E}_s^0\right]}{\gamma T} + \frac{4L^2 \mathbb{E}\left[\mathcal{C}_s^0\right]}{T} + \frac{8\eta_g^2 L^2}{\gamma^2 T} \sum_{t=0}^{T-1} \mathbb{E}\|d^{t+1}\|^2$$

$$+ \frac{384\eta_l^2 K^2 L^2}{T} \sum_{t=0}^{T-1} \mathbb{E}\|\nabla f_s(x^t)\|^2 + 48\gamma\eta_l^2 K^2 L^2 \sigma^2 \left(1 + \frac{16}{K}\right) + \frac{8\gamma\sigma^2}{K|\mathcal{R}_s|}.$$

Using the bounded gradient assumption, we can obtain (17). □

# D. Descent Direction Analysis

In this section, we analyze the geometric properties of the global descent direction $d$ generated by the server. We establish the critical one-step descent inequalities in Proposition D.3 for the global objective function $\mathbf{F}(x^t)\mathbf{A}^\top\boldsymbol{\lambda}$.

## D.1. Proof of Lemma D.1

**Lemma D.1** (Descent Inequality for Exact-Update). *Consider the sequence generated by* (8) *using the exact regularized QP solver. For any task $s \in [S]$, the inner product between the modified gradient and the descent direction satisfies the following inequality:*

$$-\left\langle \sum_{j\in[S]} A_{sj}\bar{\Delta}_j^{t+1}, d^{t+1} \right\rangle \leq -\|d^{t+1}\|^2 - \rho\left(\|\boldsymbol{\lambda}_{\rho,*}^{t+1}\|^2 - \lambda_{s,\rho,*}^{t+1}\right) \leq -\|d^{t+1}\|^2 + \rho.$$

*For any reference weight vector $\boldsymbol{\lambda} \in \omega$, it implies*

$$-\left\langle \sum_{s\in[S]} \lambda_s \sum_{j\in[S]} A_{sj}\bar{\Delta}_j^{t+1} = \bar{\boldsymbol{\Delta}}^{t+1}\mathbf{A}^\top\boldsymbol{\lambda}, d^{t+1} \right\rangle \leq -\|d^{t+1}\|^2 + \rho. \tag{18}$$

*Proof.* Consider the Lagrangian function associated with the constrained optimization problem for $\boldsymbol{\lambda}_{\rho,*}^{t+1}$:

$$\mathcal{L}(\boldsymbol{\lambda}, \nu, \boldsymbol{\xi}) = \frac{1}{2}\left\| \sum_{s\in[S]} \lambda_s \sum_{j\in[S]} A_{sj}\bar{\Delta}_j^{t+1} \right\|^2 + \frac{\rho}{2} \sum_{s\in[S]} \lambda_s^2 - \nu\left( \sum_{s\in[S]} \lambda_s - 1 \right) - \sum_{s\in[S]} \xi_s\lambda_s,$$

where $\nu \in \mathbb{R}$ is the Lagrange multiplier for the equality constraint $\sum_{s\in[S]} \lambda_s = 1$, and $\xi_s \geq 0$ are the multipliers for the non-negativity constraints $\lambda_s \geq 0, \forall s \in [S]$. The first-order optimality condition (KKT condition) with respect to $\lambda_s$ is given by:

$$\frac{\partial\mathcal{L}}{\partial\lambda_s} = \left\langle \sum_{j\in[S]} A_{sj}\bar{\Delta}_j^{t+1}, \underbrace{\sum_{s\in[S]} \lambda_{s,\rho,*}^{t+1} \sum_{j\in[S]} A_{sj}\bar{\Delta}_j^{t+1}}_{=d^{t+1}} \right\rangle + \rho\lambda_{s,\rho,*}^{t+1} - \nu - \xi_s = 0. \tag{19}$$

To determine the value of $\nu$, we multiply (19) by $\lambda_{s,\rho}^{t+1}$ and sum over all $s \in [S]$:

$$\left\langle \underbrace{\sum_{s\in[S]} \lambda_{s,\rho,*}^{t+1} \sum_{j\in[S]} A_{sj}\bar{\Delta}_j^{t+1}}_{=d^{t+1}}, d^{t+1} \right\rangle + \rho \sum_{s\in[S]} (\lambda_{s,\rho,*}^{t+1})^2 = \nu \sum_{s\in[S]} \lambda_{s,\rho,*}^{t+1} + \sum_{s\in[S]} \xi_s\lambda_{s,\rho,*}^{t+1}.$$

Using the property $\sum_{s\in[S]} \lambda_{s,\rho,*}^{t+1} = 1$, and the complementary slackness condition $\xi_s\lambda_{s,\rho,*}^{t+1} = 0$, the equation simplifies to:

$$\|d^{t+1}\|^2 + \rho\|\boldsymbol{\lambda}_{\rho,*}^{t+1}\|^2 = \nu.$$

Substituting $\nu$ back into (19), we obtain:

$$\left\langle \sum_{j\in[S]} A_{sj}\bar{\Delta}_j^{t+1}, d^{t+1} \right\rangle + \rho\lambda_{s,\rho,*}^{t+1} = \|d^{t+1}\|^2 + \rho\|\boldsymbol{\lambda}_{\rho,*}^{t+1}\|^2 + \xi_s.$$

Since $\xi_s \geq 0$ (dual feasibility), we can drop it and reverse the inequality direction to obtain:

$$\left\langle \sum_{j\in[S]} A_{sj}\bar{\Delta}_j^{t+1}, d^{t+1} \right\rangle \geq \|d^{t+1}\|^2 + \rho\left(\|\boldsymbol{\lambda}_{\rho,*}^{t+1}\|^2 - \lambda_{s,\rho,*}^{t+1}\right).$$

Since $\boldsymbol{\lambda}_{\rho,*}^{t+1} \in \boldsymbol{\omega}$, we have $\lambda_{s,\rho,*}^{t+1} \leq 1, \forall s \in [S]$, and $\|\boldsymbol{\lambda}_{\rho,*}^{t+1}\|^2 \geq 1/S$. In the worst case, the term $(\|\boldsymbol{\lambda}_{\rho,*}^{t+1}\|^2 - \lambda_{s,\rho,*}^{t+1})$ is lower bounded by $-1$ (conceptually, though strictly tighter). Then, we can loosely bound the regularization term by $\rho$. This completes the proof. $\qquad\square$

The regularization term $\rho$ introduces a relaxation to the strict common descent condition found in standard MGDA. In the unregularized case ($\rho = 0$), the inequality holds strictly as $-\left\langle \sum_{j\in[S]} A_{sj}\bar{\Delta}_j^{t+1}, d^{t+1} \right\rangle \leq -\|d^{t+1}\|^2$, ensuring that the global direction $d$ is a descent direction for every task individually. The introduction of $\rho > 0$ adds a bias term, implying that for certain tasks, the update might not be strictly descending (allowing slight ascent up to $\mathcal{O}(\rho)$). The bias is controlled and vanishes as $\rho \to 0$. However, this regularization term discourages the solution from collapsing into sparse simplex corners (where one task dominates, a phenomenon often observed in standard MGDA). Instead, it promotes a more uniform weight distribution, encouraging the model to learn shared representations that are beneficial across a broader set of tasks. This relaxation is also the necessary trade-off for obtaining Lipschitz continuous weights $\boldsymbol{\lambda}_{\rho,*}$, which is crucial for controlling the variance and ensuring robustness.

### D.2. Proof of Lemma D.2

**Lemma D.2** (Descent Inequality for Step-Update). *Consider the sequence generated by* (8) *using the one-step projected gradient descent update for the weights. For any reference weight vector $\boldsymbol{\lambda} \in \boldsymbol{\omega}$, the following inequality holds:*

$$-\left\langle \bar{\boldsymbol{\Delta}}^{t+1}\mathbf{A}^\top\boldsymbol{\lambda}, d^{t+1} \right\rangle \leq -\left[(1-\eta_\lambda\rho) - \frac{\eta_\lambda}{2}\|\mathbf{A}\|_2^2\|\bar{\boldsymbol{\Delta}}^{t+1}\|_F^2\right]\|d^{t+1}\|^2 + \frac{\rho}{2}\left(\|\boldsymbol{\lambda}\|^2 - \|\boldsymbol{\lambda}_\rho^{t+1} - \boldsymbol{\lambda}\|^2\right)$$

$$-\frac{1}{2\eta_\lambda}\left(\|\boldsymbol{\lambda}_\rho^{t+2} - \boldsymbol{\lambda}\|^2 - \|\boldsymbol{\lambda}_\rho^{t+1} - \boldsymbol{\lambda}\|^2\right). \tag{20}$$

*Proof.* We start with the non-expansiveness property of the projection operator $\Pi_{\boldsymbol{\omega}}$. For any weight vector $\boldsymbol{\lambda} \in \boldsymbol{\omega}$:

$$\|\boldsymbol{\lambda}_\rho^{t+2} - \boldsymbol{\lambda}\|^2 \stackrel{(8f)}{=} \left\|\Pi_{\boldsymbol{\omega}}\left(\boldsymbol{\lambda}_\rho^{t+1} - \eta_\lambda\left[\mathbf{A}(\bar{\boldsymbol{\Delta}}^{t+1})^\top\bar{\boldsymbol{\Delta}}^{t+1}\mathbf{A}^\top + \rho\mathbf{I}\right]\boldsymbol{\lambda}_\rho^{t+1}\right) - \boldsymbol{\lambda}\right\|^2$$

$$\leq \left\|\boldsymbol{\lambda}_\rho^{t+1} - \eta_\lambda\left[\mathbf{A}(\bar{\boldsymbol{\Delta}}^{t+1})^\top\bar{\boldsymbol{\Delta}}^{t+1}\mathbf{A}^\top + \rho\mathbf{I}\right]\boldsymbol{\lambda}_\rho^{t+1} - \boldsymbol{\lambda}\right\|^2$$

$$= \left\|(1-\eta_\lambda\rho)\boldsymbol{\lambda}_\rho^{t+1} - \boldsymbol{\lambda}\right\|^2 - 2\eta_\lambda\left\langle(1-\eta_\lambda\rho)\boldsymbol{\lambda}_\rho^{t+1} - \boldsymbol{\lambda}, \mathbf{A}(\bar{\boldsymbol{\Delta}}^{t+1})^\top\bar{\boldsymbol{\Delta}}^{t+1}\mathbf{A}^\top\boldsymbol{\lambda}_\rho^{t+1}\right\rangle + \eta_\lambda^2\left\|\mathbf{A}(\bar{\boldsymbol{\Delta}}^{t+1})^\top\bar{\boldsymbol{\Delta}}^{t+1}\mathbf{A}^\top\boldsymbol{\lambda}_\rho^{t+1}\right\|^2$$

$$= -2\eta_\lambda(1-\eta_\lambda\rho)\|\underbrace{\bar{\boldsymbol{\Delta}}^{t+1}\mathbf{A}^\top\boldsymbol{\lambda}_\rho^{t+1}}_{\stackrel{(8g)}{=}d^{t+1}}\|^2 + 2\eta_\lambda\langle\bar{\boldsymbol{\Delta}}^{t+1}\mathbf{A}^\top\boldsymbol{\lambda}, \underbrace{\bar{\boldsymbol{\Delta}}^{t+1}\mathbf{A}^\top\boldsymbol{\lambda}_\rho^{t+1}}_{\stackrel{(8g)}{=}d^{t+1}}\rangle + \eta_\lambda^2\|\mathbf{A}(\bar{\boldsymbol{\Delta}}^{t+1})^\top\underbrace{\bar{\boldsymbol{\Delta}}^{t+1}\mathbf{A}^\top\boldsymbol{\lambda}_\rho^{t+1}}_{\stackrel{(8g)}{=}d^{t+1}}\|^2$$

$$+ \|(1-\eta_\lambda\rho)(\boldsymbol{\lambda}_\rho^{t+1} - \boldsymbol{\lambda}) + \eta_\lambda\rho\boldsymbol{\lambda}\|^2$$

$$\stackrel{(10b)}{\leq} (1-\eta_\lambda\rho)\|\boldsymbol{\lambda}_\rho^{t+1} - \boldsymbol{\lambda}\|^2 + \eta_\lambda\rho\|\boldsymbol{\lambda}\|^2 - \eta_\lambda\left[2(1-\eta_\lambda\rho) - \eta_\lambda\|\bar{\boldsymbol{\Delta}}^{t+1}\mathbf{A}^\top\|_2^2\right]\|d^{t+1}\|^2 + 2\eta_\lambda\langle\bar{\boldsymbol{\Delta}}^{t+1}\mathbf{A}^\top\boldsymbol{\lambda}, d^{t+1}\rangle$$

$$\leq \|\boldsymbol{\lambda}_\rho^{t+1} - \boldsymbol{\lambda}\|^2 + \eta_\lambda\rho\left(\|\boldsymbol{\lambda}\|^2 - \|\boldsymbol{\lambda}_\rho^{t+1} - \boldsymbol{\lambda}\|^2\right) - \eta_\lambda\left[2(1-\eta_\lambda\rho) - \eta_\lambda\|\mathbf{A}\|_2^2\|\bar{\boldsymbol{\Delta}}^{t+1}\|_F^2\right]\|d^{t+1}\|^2$$

$$+ 2\eta_\lambda\langle\bar{\boldsymbol{\Delta}}^{t+1}\mathbf{A}^\top\boldsymbol{\lambda}, d^{t+1}\rangle,$$

where the last inequality holds due to

$$\|\bar{\boldsymbol{\Delta}}^{t+1}\mathbf{A}^\top\|_2^2 \leq \|\bar{\boldsymbol{\Delta}}^{t+1}\mathbf{A}^\top\|_F^2 \leq \|\mathbf{A}\|_2^2\|\bar{\boldsymbol{\Delta}}^{t+1}\|_F^2.$$

Rearranging the inequality to isolate $-\langle\bar{\boldsymbol{\Delta}}^{t+1}\mathbf{A}^\top\boldsymbol{\lambda}, d^{t+1}\rangle$ completes the proof. $\qquad\square$

### D.3. Proof of Proposition D.3

**Proposition D.3.** *Under Assumptions 4.1-4.3, consider the sequence generated by (8). For notational simplicity when telescoping the descent inequalities, we assume the $\mathbf{A}$-weighted objectives considered below are lower bounded and normalize the corresponding lower bounds to $0$.*

- **Exact-Update.** *If the weights are updated using the exact regularized QP solver, the following inequality holds:*

$$
\frac{1}{T}\sum_{t=0}^{T-1}\mathbb{E}\left\|\nabla\mathbf{F}(x^t)\mathbf{A}^\top\boldsymbol{\lambda}_0^*(x^t)\right\|^2 \leq \frac{6\|\mathbf{A}\|_2^2}{T}\sum_{t=0}^{T-1}\sum_{s\in[S]}\mathbb{E}[\mathcal{E}_s^t] + \frac{8\mathbf{F}(x^0)\mathbf{A}^\top\boldsymbol{\lambda}}{\eta_g T}
$$
$$
- \frac{2\left(1-2\eta_g L\|\mathbf{A}\|_\infty\right)}{T}\sum_{t=0}^{T-1}\mathbb{E}\|d^{t+1}\|^2 + 8\rho. \tag{21}
$$

- **Step-Update.** *If the weights are updated using the one-step projected gradient descent and the additional Assumption 4.5 holds, we have the following inequality:*

$$
\frac{1}{T}\sum_{t=0}^{T-1}\mathbb{E}\left\|\nabla\mathbf{F}(x^t)\mathbf{A}^\top\boldsymbol{\lambda}_0^*(x^t)\right\|^2 \leq \frac{6\|\mathbf{A}\|_2^2}{T}\sum_{t=0}^{T-1}\sum_{s\in[S]}\mathbb{E}[\mathcal{E}_s^t] + \frac{8\mathbf{F}(x^0)\mathbf{A}^\top\boldsymbol{\lambda}}{\eta_g T} + \frac{8}{\eta_\lambda T} \tag{22}
$$
$$
- \frac{2\left[1-4\eta_\lambda\rho-2\eta_\lambda SM^2\|\mathbf{A}\|_2^2-2\eta_g L\|\mathbf{A}\|_\infty\right]}{T}\sum_{t=0}^{T-1}\mathbb{E}\|d^{t+1}\|^2 + 4\rho.
$$

*Proof.* For any given task $s\in[S]$, we analyze the decrease in its objective function $f_s$ from $x^t$ to $x^{t+1}$. We start by applying the $L$-smoothness property of $f_s$:

$$
f_s(x^{t+1}) - f_s(x^t) \overset{(11b)}{\leq} \left\langle\nabla f_s(x^t), \underbrace{x^{t+1}-x^t}_{\overset{(8h)}{=}-\eta_g d^{t+1}}\right\rangle + \frac{L}{2}\left\|\underbrace{x^{t+1}-x^t}_{\overset{(8h)}{=}-\eta_g d^{t+1}}\right\|^2
$$
$$
= \langle\nabla f_s(x^t)-\bar{\Delta}_s^{t+1}, -\eta_g d^{t+1}\rangle + \langle\bar{\Delta}_s^{t+1}, -\eta_g d^{t+1}\rangle + \frac{L\eta_g^2}{2}\|d^{t+1}\|^2.
$$

Multiplying the inequality for each task $j$ by the preference term $A_{sj}$ and summing over $s$ and $j$ with weights $\boldsymbol{\lambda}$, we arrive at the aggregated descent bound:

$$
\sum_{s\in[S]}\lambda_s\sum_{j\in[S]}A_{sj}\left[f_j(x^{t+1})-f_j(x^t)\right] = \left[\mathbf{F}(x^{t+1})-\mathbf{F}(x^t)\right]\mathbf{A}^\top\boldsymbol{\lambda}
$$
$$
\leq -\eta_g\sum_{s\in[S]}\lambda_s\left\langle\sum_{j\in[S]}A_{sj}\left[\nabla f_j(x^t)-\bar{\Delta}_j^{t+1}\right], d^{t+1}\right\rangle
$$
$$
-\eta_g\left\langle\underbrace{\sum_{s\in[S]}\lambda_s\sum_{j\in[S]}A_{sj}\bar{\Delta}_j^{t+1}}_{=\bar{\Delta}^{t+1}\mathbf{A}^\top\boldsymbol{\lambda}}, d^{t+1}\right\rangle + \frac{L\eta_g^2}{2}\underbrace{\sum_{s\in[S]}\lambda_s\sum_{j\in[S]}A_{sj}}_{\leq\sum_{s\in[S]}\lambda_s\max_{s\in[S]}(\sum_{j\in[S]}A_{sj})=\|\mathbf{A}\|_\infty}\|d^{t+1}\|^2
$$
$$
\leq \frac{\eta_g}{2}\|\mathbf{A}\|_2^2\sum_{s\in[S]}\mathcal{E}_s^t + \frac{\eta_g}{2}\left(1+\eta_g L\|\mathbf{A}\|_\infty\right)\|d^{t+1}\|^2 - \eta_g\left\langle\bar{\Delta}^{t+1}\mathbf{A}^\top\boldsymbol{\lambda}, d^{t+1}\right\rangle. \tag{23}
$$

The last inequality of (23) is deduced by applying Young's inequality (9b) with $\eta=\eta_g$ to bound the inner product:

$$
-\eta_g\sum_{s\in[S]}\lambda_s\left\langle\sum_{j\in[S]}A_{sj}\left[\nabla f_j(x^t)-\bar{\Delta}_j^{t+1}\right], d^{t+1}\right\rangle
$$
$$
\overset{(9b)}{\leq} \frac{\eta_g}{2}\sum_{s\in[S]}\underbrace{\lambda_s}_{\leq 1}\left\|\sum_{j\in[S]}A_{sj}\left[\nabla f_j(x^t)-\bar{\Delta}_j^{t+1}\right]\right\|^2 + \frac{\eta_g}{2}\sum_{s\in[S]}\underbrace{\lambda_s}_{=1}\|d^{t+1}\|^2
$$

$$\leq \frac{\eta_g}{2} \sum_{s \in [S]} \left\| \sum_{j \in [S]} A_{sj} \left[ \nabla f_j(x^t) - \bar{\Delta}_j^{t+1} \right] \right\|^2 + \frac{\eta_g}{2} \|d^{t+1}\|^2$$

$$\leq \frac{\eta_g}{2} \|\mathbf{A}\|_2^2 \sum_{s \in [S]} \underbrace{\left\| \nabla f_s(x^t) - \bar{\Delta}_s^{t+1} \right\|^2}_{=\mathcal{E}_s^t} + \frac{\eta_g}{2} \|d^{t+1}\|^2,$$

where the last inequality holds due to

$$\|XY\|_F \leq \|X\|_2 \|Y\|_F = \|X^\top\|_2 \|Y\|_F, \text{ with } X = \mathbf{A}^\top, \ Y = \nabla \mathbf{F}(x^t) - \bar{\Delta}^{t+1}.$$

Then, the gradient estimation error is decoupled from the descent direction.

**Exact Regularized Update.** Substituting (18) in Lemma D.1 back into (23) gives the following one-step progress:

$$\left[ \mathbf{F}(x^{t+1}) - \mathbf{F}(x^t) \right] \mathbf{A}^\top \boldsymbol{\lambda} \leq \frac{\eta_g}{2} \|\mathbf{A}\|_2^2 \sum_{s \in [S]} \mathcal{E}_s^t + \frac{\eta_g}{2} \left( 1 + \eta_g L \|\mathbf{A}\|_\infty \right) \|d^{t+1}\|^2 - \eta_g \left\langle \bar{\Delta}^{t+1} \mathbf{A}^\top \boldsymbol{\lambda}, d^{t+1} \right\rangle$$

$$\overset{(18)}{\leq} \frac{\eta_g}{2} \|\mathbf{A}\|_2^2 \sum_{s \in [S]} \mathcal{E}_s^t - \frac{\eta_g}{4} \left( 1 - 2\eta_g L \|\mathbf{A}\|_\infty \right) \|d^{t+1}\|^2 - \frac{\eta_g}{4} \|d^{t+1}\|^2 + \eta_g \rho$$

$$\overset{(24)}{\leq} \frac{3\eta_g}{4} \|\mathbf{A}\|_2^2 \sum_{s \in [S]} \mathcal{E}_s^t - \frac{\eta_g}{4} \left( 1 - 2\eta_g L \|\mathbf{A}\|_\infty \right) \|d^{t+1}\|^2 - \frac{\eta_g}{8} \left\| \nabla \mathbf{F}(x^t) \mathbf{A}^\top \boldsymbol{\lambda}_\rho^{t+1} \right\|^2 + \eta_g \rho.$$

where the last inequality is obtained by using

$$-\|d^{t+1}\|^2 = -\left\| \bar{\Delta}^{t+1} \mathbf{A}^\top \boldsymbol{\lambda}_\rho^{t+1} \right\|^2$$

$$\overset{(9c)}{\leq} \left\| \left[ \nabla \mathbf{F}(x^t) - \bar{\Delta}^{t+1} \right] \mathbf{A}^\top \boldsymbol{\lambda}_\rho^{t+1} \right\|^2 - \frac{1}{2} \left\| \nabla \mathbf{F}(x^t) \mathbf{A}^\top \boldsymbol{\lambda}_\rho^{t+1} \right\|^2$$

$$= \left\| \sum_{s \in [S]} \lambda_{s,\rho}^{t+1} \sum_{j \in [S]} A_{sj} [\nabla f_j(x^t) - \bar{\Delta}_j^{t+1}] \right\|^2 - \frac{1}{2} \left\| \nabla \mathbf{F}(x^t) \mathbf{A}^\top \boldsymbol{\lambda}_\rho^{t+1} \right\|^2$$

$$\overset{(10b)}{\leq} \sum_{s \in [S]} \lambda_{s,\rho}^{t+1} \left\| \sum_{j \in [S]} A_{sj} [\nabla f_j(x^t) - \bar{\Delta}_j^{t+1}] \right\|^2 - \frac{1}{2} \left\| \nabla \mathbf{F}(x^t) \mathbf{A}^\top \boldsymbol{\lambda}_\rho^{t+1} \right\|^2$$

$$\leq \|\mathbf{A}\|_2^2 \sum_{s \in [S]} \mathcal{E}_s^t - \frac{1}{2} \left\| \nabla \mathbf{F}(x^t) \mathbf{A}^\top \boldsymbol{\lambda}_\rho^{t+1} \right\|^2. \tag{24}$$

Rearranging the inequality to isolate $\|\nabla \mathbf{F}(x^t) \mathbf{A}^\top \boldsymbol{\lambda}_\rho^{t+1}\|^2$ gives

$$\left\| \nabla \mathbf{F}(x^t) \mathbf{A}^\top \boldsymbol{\lambda}_\rho^{t+1} \right\|^2 \leq 6 \|\mathbf{A}\|_2^2 \sum_{s \in [S]} \mathcal{E}_s^t - 2 \left( 1 - 2\eta_g L \|\mathbf{A}\|_\infty \right) \|d^{t+1}\|^2 - \frac{8}{\eta_g} \left[ \mathbf{F}(x^{t+1}) - \mathbf{F}(x^t) \right] \mathbf{A}^\top \boldsymbol{\lambda} + 8\rho.$$

Recalling the definition of $\boldsymbol{\lambda}_0^*(x^t)$, it can be obtained that

$$\left\| \nabla \mathbf{F}(x^t) \mathbf{A}^\top \boldsymbol{\lambda}_0^*(x^t) \right\|^2 = \min_{\boldsymbol{\lambda} \in \omega} \left\| \nabla \mathbf{F}(x^t) \mathbf{A}^\top \boldsymbol{\lambda} \right\|^2 \leq \left\| \nabla \mathbf{F}(x^t) \mathbf{A}^\top \boldsymbol{\lambda}_\rho^{t+1} \right\|^2.$$

Substituting it back, and averaging over $t = 0, \cdots, T - 1$, then taking expectation gives (21).

**One-step Projected Gradient Update.** The proof for the step-update variant follows a similar logical framework to that of the exact-update. Specifically, by substituting (20) in Lemma D.2 back into (23) and using (24), we obtain:

$$\left[ \mathbf{F}(x^{t+1}) - \mathbf{F}(x^t) \right] \mathbf{A}^\top \boldsymbol{\lambda} + \frac{\eta_g}{2\eta_\lambda} \left[ \|\boldsymbol{\lambda}_\rho^{t+2} - \boldsymbol{\lambda}\|^2 - \|\boldsymbol{\lambda}_\rho^{t+1} - \boldsymbol{\lambda}\|^2 \right]$$

$$\leq \frac{\eta_g}{2} \|\mathbf{A}\|_2^2 \sum_{s \in [S]} \mathcal{E}_s^t - \frac{\eta_g}{2} \left[ 1 - 2\eta_\lambda \rho - \eta_\lambda \|\mathbf{A}\|_2^2 \|\bar{\Delta}^{t+1}\|_F^2 - \eta_g L \|\mathbf{A}\|_\infty \right] \|d^{t+1}\|^2 + \frac{\eta_g \rho}{2} \underbrace{\left[ \|\boldsymbol{\lambda}\|^2 - \|\boldsymbol{\lambda}_\rho^{t+1} - \boldsymbol{\lambda}\|^2 \right]}_{\leq 1}$$

$$\leq \frac{\eta_g}{2}\|\mathbf{A}\|_2^2 \sum_{s\in[S]} \mathcal{E}_s^t - \frac{\eta_g}{4}\left[1 - 4\eta_\lambda\rho - 2\eta_\lambda\|\mathbf{A}\|_2^2\|\bar{\mathbf{\Delta}}^{t+1}\|_F^2 - 2\eta_g L\|\mathbf{A}\|_\infty\right]\|d^{t+1}\|^2 - \frac{\eta_g}{4}\|d^{t+1}\|^2 + \frac{\eta_g\rho}{2}$$

$$\overset{(24)}{\leq} \frac{3\eta_g}{4}\|\mathbf{A}\|_2^2 \sum_{s\in[S]} \mathcal{E}_s^t - \frac{\eta_g}{4}\left[1 - 4\eta_\lambda\rho - 2\eta_\lambda\|\mathbf{A}\|_2^2\|\bar{\mathbf{\Delta}}^{t+1}\|_F^2 - 2\eta_g L\|\mathbf{A}\|_\infty\right]\|d^{t+1}\|^2 - \frac{\eta_g}{8}\left\|\nabla\mathbf{F}(x^t)\mathbf{A}^\top\boldsymbol{\lambda}_\rho^{t+1}\right\|^2 + \frac{\eta_g\rho}{2}.$$

Rearranging this inequality and introducing $\boldsymbol{\lambda}_0^*(x^t)$, it following that

$$\left\|\nabla\mathbf{F}(x^t)\mathbf{A}^\top\boldsymbol{\lambda}_0^*(x^t)\right\|^2 \leq \left\|\nabla\mathbf{F}(x^t)\mathbf{A}^\top\boldsymbol{\lambda}_\rho^{t+1}\right\|^2$$

$$\leq 6\|\mathbf{A}\|_2^2 \sum_{s\in[S]} \mathcal{E}_s^t - 2\left[1 - 4\eta_\lambda\rho - 2\eta_\lambda\|\mathbf{A}\|_2^2\|\bar{\mathbf{\Delta}}^{t+1}\|_F^2 - 2\eta_g L\|\mathbf{A}\|_\infty\right]\|d^{t+1}\|^2$$

$$- \frac{8}{\eta_g}\left[\mathbf{F}(x^{t+1}) - \mathbf{F}(x^t)\right]\mathbf{A}^\top\boldsymbol{\lambda} - \frac{4}{\eta_\lambda}\left[\|\boldsymbol{\lambda}_\rho^{t+2} - \boldsymbol{\lambda}\|^2 - \|\boldsymbol{\lambda}_\rho^{t+1} - \boldsymbol{\lambda}\|^2\right] + 4\rho.$$

Taking the expectation, substituting the bound $\mathbb{E}\|\bar{\mathbf{\Delta}}^{t+1}\|_F^2 \leq \sum_{s\in[S]}\mathbb{E}\|\bar{\Delta}_s^{t+1}\|^2 \leq SM^2$ (Lemma B.6) into the telescoping sum over $t = 0,\cdots,T-1$, we obtain (22). $\square$

Proposition D.3 provides a fundamental one-step progress bound, making the convergence guarantee contingent on the magnitude of $\mathcal{E}_s^t$. By unrolling the recursive dynamics from Lemmas C.1 and C.2 under stability-ensuring parameter choices, we can establish an explicit bound on the time-averaged gradient estimation error. Then, we can substitute (17) into the time-averaged inequality of Proposition D.3. By carefully selecting the server learning rate $\eta_g$, the negative term associated with $\|d^{t+1}\|^2$ in (21) can be made to dominate its positive counterpart arising from the error bound. This allows us to telescope the descent inequality and derive the main theorem.

## E. Convergence Analysis in the Non-Convex Setting

This section provides the complete proof for Theorem 4.8, establishing the convergence rate of DREAM for general non-convex objectives. By combining the descent inequality from Appendix D with the accumulated error bounds from Appendix C, and carefully telescoping the inequalities and applying the stability conditions for the learning rates, the convergence results can be obtained.

**Theorem E.1** (DREAM in the Non-Convex Setting). *Under Assumptions 4.1-4.3, consider the sequence generated by Algorithm 1. Assume that all clients participate in every task, i.e., $|\mathcal{R}_s| = N$ for all $s \in [S]$.*

- ***Exact-Update.*** *If the weights are updated using the exact regularized QP solver, and step sizes satisfy*

$$48\eta_l^2 K^2 L^2 \leq \min\left\{\frac{1}{4\gamma^2}, \frac{1}{1-\gamma}\right\}, \quad 24S\eta_g^2 L^2\|\mathbf{A}\|_2^2 \leq (1 - 2\eta_g L\|\mathbf{A}\|_\infty)\gamma^2,$$

  *there exists $0 \leq r \leq T-1$ such that*

$$\mathbb{E}\left\|\nabla\mathbf{F}(x^r)\mathbf{A}^\top\boldsymbol{\lambda}_0^*(x^r)\right\|^2 \leq \mathcal{O}\left(\frac{1}{\eta_g T} + \frac{1}{\gamma T} + \frac{\gamma\eta_l^2}{T}\right) + \mathcal{O}\left(\eta_l^2 + \frac{\gamma}{N} + \rho\right).$$

- ***Step-Update.*** *If the weights are updated using the one-step projected gradient descent, additional Assumption 4.5 holds, and step sizes satisfy*

$$48\eta_l^2 K^2 L^2 \leq \min\left\{\frac{1}{4\gamma^2}, \frac{1}{1-\gamma}\right\}, \quad 24S\eta_g^2 L^2\|\mathbf{A}\|_2^2 \leq [1 - 2\eta_g L\|\mathbf{A}\|_\infty - 4\eta_\lambda\rho - 2\eta_\lambda SM^2\|\mathbf{A}\|_2^2]\gamma^2,$$

  *there exists $0 \leq r \leq T-1$ such that*

$$\mathbb{E}\left\|\nabla\mathbf{F}(x^r)\mathbf{A}^\top\boldsymbol{\lambda}_0^*(x^r)\right\|^2 \leq \mathcal{O}\left(\frac{1}{\eta_g T} + \frac{1}{\gamma T} + \frac{\gamma\eta_l^2}{T} + \frac{1}{\eta_\lambda T}\right) + \mathcal{O}\left(\eta_l^2 + \frac{\gamma}{N} + \rho\right).$$

*Proof.* This proof can be easily accomplished by combining the results of Proposition D.3 and Proposition C.3.

**Analysis for Exact-Update Strategy**

Substituting (17) with $|\mathcal{R}_s| = N$ into (21) with $\boldsymbol{\lambda} = \frac{1}{S} \cdot \mathbf{1}$, we can obtain

$$
\frac{1}{T} \sum_{t=0}^{T-1} \mathbb{E} \left\| \nabla \mathbf{F}(x^t) \mathbf{A}^\top \boldsymbol{\lambda}_0^*(x^t) \right\|^2 \leq \frac{8\mathbf{F}(x^0)\mathbf{A}^\top \mathbf{1}}{\eta_g ST} + \frac{24\|\mathbf{A}\|_2^2 \sum_{s \in [S]} \mathbb{E}\left[\mathcal{E}_s^0\right]}{\gamma T} + \frac{24L^2\|\mathbf{A}\|_2^2 \sum_{s \in [S]} \mathbb{E}\left[\mathcal{C}_s^0\right]}{T}
$$

$$
- \frac{2}{T} \underbrace{\left( 1 - 2\eta_g L \|\mathbf{A}\|_\infty - \frac{24S\eta_g^2 L^2 \|\mathbf{A}\|_2^2}{\gamma^2} \right)}_{\geq 0} \sum_{t=0}^{T-1} \mathbb{E}\|d^{t+1}\|^2
$$

$$
+ 288S\eta_l^2 K^2 L^2 \|\mathbf{A}\|_2^2 \left[ \gamma\sigma^2 \left(1 + \frac{16}{K}\right) + 8G^2 \right] + \frac{48\gamma S\sigma^2 \|\mathbf{A}\|_2^2}{KN} + 8\rho
$$

$$
= \mathcal{O}\left( \frac{1}{\eta_g T} + \frac{1}{\gamma T} + \frac{\gamma\eta_l^2}{T} \right) + \mathcal{O}\left( \eta_l^2 + \frac{\gamma}{N} + \rho \right),
$$

where $\mathbb{E}\left[\mathcal{C}_s^0\right] \leq 6\gamma\eta_l^2 K^2 [\sigma^2 + \frac{8}{|\mathcal{R}_s|} \sum_{i \in \mathcal{R}_s} \mathbb{E}\|\nabla f_{s,i}(x^0)\|^2]$ is used.

**Analysis for Step-Update Strategy**

Similarly, substituting (17) with $|\mathcal{R}_s| = N$ into (22) with $\boldsymbol{\lambda} = \frac{1}{S} \cdot \mathbf{1}$, we have

$$
\frac{1}{T} \sum_{t=0}^{T-1} \mathbb{E} \left\| \nabla \mathbf{F}(x^t) \mathbf{A}^\top \boldsymbol{\lambda}_0^*(x^t) \right\|^2
$$

$$
\leq \frac{8\mathbf{F}(x^0)\mathbf{A}^\top \mathbf{1}}{\eta_g ST} + \frac{24\|\mathbf{A}\|_2^2 \sum_{s \in [S]} \mathbb{E}\left[\mathcal{E}_s^0\right]}{\gamma T} + \frac{24L^2\|\mathbf{A}\|_2^2 \sum_{s \in [S]} \mathbb{E}\left[\mathcal{C}_s^0\right]}{T} + \frac{8}{\eta_\lambda T}
$$

$$
- \frac{2}{T} \underbrace{\left[ 1 - 2\eta_g L \|\mathbf{A}\|_\infty - \frac{24S\eta_g^2 L^2 \|\mathbf{A}\|_2^2}{\gamma^2} - 4\eta_\lambda \rho - 2\eta_\lambda SM^2 \|\mathbf{A}\|_2^2 \right]}_{\geq 0} \sum_{t=0}^{T-1} \mathbb{E}\|d^{t+1}\|^2
$$

$$
+ 288S\eta_l^2 K^2 L^2 \|\mathbf{A}\|_2^2 \left[ \gamma\sigma^2 \left(1 + \frac{16}{K}\right) + 8G^2 \right] + \frac{48\gamma S\sigma^2 \|\mathbf{A}\|_2^2}{KN} + 4\rho
$$

$$
= \mathcal{O}\left( \frac{1}{\eta_g T} + \frac{1}{\gamma T} + \frac{\gamma\eta_l^2}{T} + \frac{1}{\eta_\lambda T} \right) + \mathcal{O}\left( \eta_l^2 + \frac{\gamma}{N} + \rho \right),
$$

which completes the proof. $\qquad \square$

The general bounds derived in Theorem 4.8 can be simplified to the explicit rates by setting specific parameter schedules. For the Exact-Update case, substituting the parameters $\gamma = \mathcal{O}(N^{\frac{1}{2}}T^{-\frac{1}{2}})$, $\eta_g = \mathcal{O}(N^{\frac{1}{2}}T^{-\frac{1}{2}})$, $\rho = \mathcal{O}(N^{-\frac{1}{2}}T^{-\frac{1}{2}})$, into the derived bound yields $\mathcal{O}(1/\sqrt{NT})$. This confirms the linear speedup w.r.t. the number of clients $N$. For the Step-Update case, by choosing a suitable weight learning rate $\eta_\lambda$, it preserves the same state-of-the-art convergence rate.

## F. Analysis of CA Direction Distance

We provide the proof for Theorem 4.11, quantifying the distance between the actual update direction and the ideal CA direction. This analysis highlights the impact of the regularization parameter $\rho$ and the weight tracking error on the directional accuracy.

**Theorem F.1** (Time-Averaged CA Direction Distance). *Under Assumptions 4.1-4.3, consider the sequence generated by Algorithm 1. Assume that all clients participate in every task, i.e., $|\mathcal{R}_s| = N$ for all $s \in [S]$.*

- *Exact-Update. If the weights are updated using the exact regularized QP solver, and step sizes satisfy*

$$
48\eta_l^2 K^2 L^2 \leq \min\left\{ \frac{1}{4\gamma^2}, \frac{1}{1-\gamma} \right\}, \quad 28S\eta_g^2 L^2 \|\mathbf{A}\|_2^2 \leq (1 - 2\eta_g L \|\mathbf{A}\|_\infty)\gamma^2,
$$

*the time-averaged CA direction distance is of the same order as the convergence rate established in Theorem 4.8:*

$$\frac{1}{T}\sum_{t=0}^{T-1}\mathbb{E}\|\nabla\mathbf{F}(x^t)\mathbf{A}^\top\boldsymbol{\lambda}_0^*(x^t) - d^{t+1}\|^2 \leq \mathcal{O}\left(\frac{1}{\eta_g T} + \frac{1}{\gamma T} + \frac{\gamma\eta_l^2}{T}\right) + \mathcal{O}\left(\eta_l^2 + \frac{\gamma}{N} + \rho\right).$$

- **Step-Update.** *If the weights are updated using the one-step projected gradient descent, additional Assumption 4.5 holds, and step sizes satisfy*

$$48\eta_l^2 K^2 L^2 \leq \min\left\{\frac{1}{4\gamma^2}, \frac{1}{1-\gamma}\right\}, \ 56S\eta_g^2 L^2\|\mathbf{A}\|_2^2 \leq [1 - 4\eta_g L\|\mathbf{A}\|_\infty - 10\eta_\lambda\rho - 5\eta_\lambda SM^2\|\mathbf{A}\|_2^2]\gamma^2,$$

*the time-averaged CA direction distance satisfies the following bound:*

$$\frac{1}{T}\sum_{t=0}^{T-1}\mathbb{E}\|\nabla\mathbf{F}(x^t)\mathbf{A}^\top\boldsymbol{\lambda}_0^*(x^t) - d^{t+1}\|^2 \leq \mathcal{O}\left(\frac{1}{\eta_g T} + \frac{1}{\gamma T} + \frac{\gamma\eta_l^2}{T} + \frac{1}{\eta_\lambda T}\right) + \mathcal{O}\left(\eta_l^2 + \frac{\gamma}{N} + \frac{\eta_g^2}{\eta_\lambda^2\rho^2} + \rho\right).$$

*Proof.* We start by analyzing a more general quantity: the discrepancy between the theoretical gradient map $\nabla\mathbf{F}(x^t)\mathbf{A}^\top\boldsymbol{\lambda}$ for an arbitrary weight $\boldsymbol{\lambda}$ and the actual direction $d^{t+1}$ generated by the algorithm. Consider the squared Euclidean norm of their difference, we have:

$$\|\nabla\mathbf{F}(x^t)\mathbf{A}^\top\boldsymbol{\lambda} - d^{t+1}\|^2 = \|\nabla\mathbf{F}(x^t)\mathbf{A}^\top\boldsymbol{\lambda}\|^2 + \|d^{t+1}\|^2 - 2\langle\nabla\mathbf{F}(x^t)\mathbf{A}^\top\boldsymbol{\lambda}, d^{t+1}\rangle$$

$$= \|\nabla\mathbf{F}(x^t)\mathbf{A}^\top\boldsymbol{\lambda}\|^2 + \|d^{t+1}\|^2 - 2\langle\bar{\boldsymbol{\Delta}}^{t+1}\mathbf{A}^\top\boldsymbol{\lambda}, d^{t+1}\rangle + 2\langle[\bar{\boldsymbol{\Delta}}^{t+1} - \nabla\mathbf{F}(x^t)]\mathbf{A}^\top\boldsymbol{\lambda}, d^{t+1}\rangle$$

$$\overset{(9b)}{\leq} \|\nabla\mathbf{F}(x^t)\mathbf{A}^\top\boldsymbol{\lambda}\|^2 + 2\|d^{t+1}\|^2 - 2\langle\bar{\boldsymbol{\Delta}}^{t+1}\mathbf{A}^\top\boldsymbol{\lambda}, d^{t+1}\rangle + \underbrace{\|[\bar{\boldsymbol{\Delta}}^{t+1} - \nabla\mathbf{F}(x^t)]\mathbf{A}^\top\boldsymbol{\lambda}\|^2}_{\leq \|\mathbf{A}\|_2^2 \sum_{s\in[S]}\mathcal{E}_s^t}. \quad (25)$$

### Analysis of CA Direction Distance for Exact-Update

We specialize the general bound by setting the weight vector $\boldsymbol{\lambda} = \boldsymbol{\lambda}_0^*(x^t)$. Summing (25) over $t = 0, \cdots, T-1$ and dividing by $T$, we obtain the average squared norm of the CA direction distance. The right-hand side is then bounded by substituting (18), (21) and (17) in sequence:

$$\frac{1}{T}\sum_{t=0}^{T-1}\mathbb{E}\|\nabla\mathbf{F}(x^t)\mathbf{A}^\top\boldsymbol{\lambda}_0^*(x^t) - d^{t+1}\|^2$$

$$\leq \frac{1}{T}\sum_{t=0}^{T-1}\mathbb{E}\|\nabla\mathbf{F}(x^t)\mathbf{A}^\top\boldsymbol{\lambda}_0^*(x^t)\|^2 + \frac{2}{T}\sum_{t=0}^{T-1}\mathbb{E}\underbrace{\left[\|d^{t+1}\|^2 - \langle\bar{\boldsymbol{\Delta}}^{t+1}\mathbf{A}^\top\boldsymbol{\lambda}_0^*(x^t), d^{t+1}\rangle\right]}_{\overset{(18)}{\leq}\rho} + \frac{\|\mathbf{A}\|_2^2}{T}\sum_{t=0}^{T-1}\sum_{s\in[S]}\mathbb{E}\left[\mathcal{E}_s^t\right]$$

$$\overset{(21)}{\leq} \frac{7\|\mathbf{A}\|_2^2}{T}\sum_{t=0}^{T-1}\sum_{s\in[S]}\mathbb{E}\left[\mathcal{E}_s^t\right] + \frac{8\mathbf{F}(x^0)\mathbf{A}^\top\mathbf{1}}{\eta_g ST} - \frac{2(1-2\eta_g L\|\mathbf{A}\|_\infty)}{T}\sum_{t=0}^{T-1}\mathbb{E}\|d^{t+1}\|^2 + 10\rho$$

$$\overset{(17)}{\leq} \frac{8\mathbf{F}(x^0)\mathbf{A}^\top\mathbf{1}}{\eta_g ST} + \frac{28\|\mathbf{A}\|_2^2\sum_{s\in[S]}\mathbb{E}\left[\mathcal{E}_s^0\right]}{\gamma T} + \frac{28L^2\|\mathbf{A}\|_2^2\sum_{s\in[S]}\mathbb{E}\left[\mathcal{C}_s^0\right]}{T}$$

$$\quad - \frac{2}{T}\underbrace{\left(1 - 2\eta_g L\|\mathbf{A}\|_\infty - \frac{28S\eta_g^2 L^2\|\mathbf{A}\|_2^2}{\gamma^2}\right)}_{\geq 0}\sum_{t=0}^{T-1}\mathbb{E}\|d^{t+1}\|^2$$

$$\quad + 336S\eta_l^2 K^2 L^2\|\mathbf{A}\|_2^2\left[\gamma\sigma^2\left(1 + \frac{16}{K}\right) + 8G^2\right] + \frac{56\gamma S\sigma^2\|\mathbf{A}\|_2^2}{KN} + 10\rho$$

$$= \mathcal{O}\left(\frac{1}{\eta_g T} + \frac{1}{\gamma T} + \frac{\gamma\eta_l^2}{T}\right) + \mathcal{O}\left(\eta_l^2 + \frac{\gamma}{N} + \rho\right).$$

**Analysis of CA Direction Distance for Step-Update**

In this case, directly setting $\boldsymbol{\lambda} = \boldsymbol{\lambda}_0^*(x^t)$ in (25) and applying the similar substitution strategy is infeasible. The reason is that the inequality (20) introduces weight tracking errors in the analysis. To control these errors, we must rely on the Lipschitz continuity of the optimal regularized weights (Lemma B.7), which relies crucially on the strong convexity of subproblem ($\rho > 0$). To circumvent this issue, we seek to bound the target distance by introducing the regularized reference $\nabla \mathbf{F}(x^t)\mathbf{A}^\top \boldsymbol{\lambda}_\rho^*(x^t)$ as an intermediate term. Specifically, by expanding the squared norm, we have the following relation:

$$\|\nabla\mathbf{F}(x^t)\mathbf{A}^\top\boldsymbol{\lambda}_0^*(x^t) - d^{t+1}\|^2$$

$$=\|\nabla\mathbf{F}(x^t)\mathbf{A}^\top\boldsymbol{\lambda}_0^*(x^t) - \nabla\mathbf{F}(x^t)\mathbf{A}^\top\boldsymbol{\lambda}_\rho^*(x^t) + \nabla\mathbf{F}(x^t)\mathbf{A}^\top\boldsymbol{\lambda}_\rho^*(x^t) - d^{t+1}\|^2$$

$$\overset{(9a)}{\leq} 2\|\nabla\mathbf{F}(x^t)\mathbf{A}^\top\boldsymbol{\lambda}_0^*(x^t) - \nabla\mathbf{F}(x^t)\mathbf{A}^\top\boldsymbol{\lambda}_\rho^*(x^t)\|^2 + 2\|\nabla\mathbf{F}(x^t)\mathbf{A}^\top\boldsymbol{\lambda}_\rho^*(x^t) - d^{t+1}\|^2$$

$$=2\|\nabla\mathbf{F}(x^t)\mathbf{A}^\top\boldsymbol{\lambda}_\rho^*(x^t) - d^{t+1}\|^2 + 2\|\nabla\mathbf{F}(x^t)\mathbf{A}^\top\boldsymbol{\lambda}_0^*(x^t)\|^2 + 2\|\nabla\mathbf{F}(x^t)\mathbf{A}^\top\boldsymbol{\lambda}_\rho^*(x^t)\|^2$$

$$- 4\underbrace{\langle\nabla\mathbf{F}(x^t)\mathbf{A}^\top\boldsymbol{\lambda}_0^*(x^t), \nabla\mathbf{F}(x^t)\mathbf{A}^\top\boldsymbol{\lambda}_\rho^*(x^t)\rangle}_{\geq\|\nabla\mathbf{F}(x^t)\mathbf{A}^\top\boldsymbol{\lambda}_0^*(x^t)\|^2}$$

$$\leq 2\|\nabla\mathbf{F}(x^t)\mathbf{A}^\top\boldsymbol{\lambda}_\rho^*(x^t) - d^{t+1}\|^2 + 2\big[\|\nabla\mathbf{F}(x^t)\mathbf{A}^\top\boldsymbol{\lambda}_\rho^*(x^t)\|^2 - \|\nabla\mathbf{F}(x^t)\mathbf{A}^\top\boldsymbol{\lambda}_0^*(x^t)\|^2\big]$$

$$\overset{\text{Lemma } B.8}{\leq} 2\|\nabla\mathbf{F}(x^t)\mathbf{A}^\top\boldsymbol{\lambda}_\rho^*(x^t) - d^{t+1}\|^2 + 2\rho,$$

where the first-order optimality condition of the convex subproblem (which implies $\langle\nabla\mathbf{F}(x^t)\mathbf{A}^\top\boldsymbol{\lambda}_0^*(x^t), \nabla\mathbf{F}(x^t)\mathbf{A}^\top\boldsymbol{\lambda}\rangle \geq \|\nabla\mathbf{F}(x^t)\mathbf{A}^\top\boldsymbol{\lambda}_0^*(x^t)\|^2$) is utilized. We can now set $\boldsymbol{\lambda} = \boldsymbol{\lambda}_\rho^*(x^t)$ in the general bound (25) to control the first term by substituting Lemma B.8, (20), (22) and (17) in sequence:

$$\frac{1}{T}\sum_{t=0}^{T-1}\mathbb{E}\|\nabla\mathbf{F}(x^t)\mathbf{A}^\top\boldsymbol{\lambda}_\rho^*(x^t) - d^{t+1}\|^2$$

$$\leq\frac{1}{T}\sum_{t=0}^{T-1}\underbrace{\mathbb{E}\|\nabla\mathbf{F}(x^t)\mathbf{A}^\top\boldsymbol{\lambda}_\rho^*(x^t)\|^2}_{\overset{\text{Lemma } B.8}{\leq}\mathbb{E}\|\nabla\mathbf{F}(x^t)\mathbf{A}^\top\boldsymbol{\lambda}_0^*(x^t)\|^2+\rho} + \frac{2}{T}\sum_{t=0}^{T-1}\mathbb{E}\big[\|d^{t+1}\|^2 - \langle\bar{\boldsymbol{\Delta}}^{t+1}\mathbf{A}^\top\boldsymbol{\lambda}_\rho^*(x^t), d^{t+1}\rangle\big] + \frac{\|\mathbf{A}\|_2^2}{T}\sum_{t=0}^{T-1}\sum_{s\in[S]}\mathbb{E}\big[\mathcal{E}_s^t\big]$$

$$\overset{(20)}{\leq}\frac{1}{T}\sum_{t=0}^{T-1}\mathbb{E}\|\nabla\mathbf{F}(x^t)\mathbf{A}^\top\boldsymbol{\lambda}_0^*(x^t)\|^2 + \frac{\|\mathbf{A}\|_2^2}{T}\sum_{t=0}^{T-1}\sum_{s\in[S]}\mathbb{E}\big[\mathcal{E}_s^t\big] + \frac{2\eta_\lambda\rho + \eta_\lambda SM^2\|\mathbf{A}\|_2^2}{T}\sum_{t=0}^{T-1}\mathbb{E}\|d^{t+1}\|^2 + \rho$$

$$+ \frac{\rho}{T}\sum_{t=0}^{T-1}\underbrace{\big[\mathbb{E}\|\boldsymbol{\lambda}_\rho^*(x^t)\|^2 - \mathbb{E}\|\boldsymbol{\lambda}_\rho^{t+1} - \boldsymbol{\lambda}_\rho^*(x^t)\|^2\big]}_{\leq 1} + \frac{1}{\eta_\lambda T}\sum_{t=0}^{T-1}\big[\mathbb{E}\|\boldsymbol{\lambda}_\rho^{t+1} - \boldsymbol{\lambda}_\rho^*(x^t)\|^2 - \mathbb{E}\|\boldsymbol{\lambda}_\rho^{t+2} - \boldsymbol{\lambda}_\rho^*(x^t)\|^2\big]$$

$$\overset{(22)}{\leq}\frac{7\|\mathbf{A}\|_2^2}{T}\sum_{t=0}^{T-1}\sum_{s\in[S]}\mathbb{E}\big[\mathcal{E}_s^t\big] + \frac{8\mathbf{F}(x^0)\mathbf{A}^\top\mathbf{1}}{\eta_g ST} - \big[2 - 10\eta_\lambda\rho - 5\eta_\lambda SM^2\|\mathbf{A}\|_2^2 - 4\eta_g L\|\mathbf{A}\|_\infty\big]\frac{1}{T}\sum_{t=0}^{T-1}\mathbb{E}\|d^{t+1}\|^2$$

$$+ \frac{8}{\eta_\lambda T} + \frac{1}{\eta_\lambda T}\underbrace{\sum_{t=0}^{T-1}\big[\mathbb{E}\|\boldsymbol{\lambda}_\rho^{t+1} - \boldsymbol{\lambda}_\rho^*(x^t)\|^2 - \mathbb{E}\|\boldsymbol{\lambda}_\rho^{t+2} - \boldsymbol{\lambda}_\rho^*(x^{t+1})\|^2\big]}_{=\mathbb{E}\|\boldsymbol{\lambda}_\rho^1 - \boldsymbol{\lambda}_\rho^*(x^0)\|^2 - \mathbb{E}\|\boldsymbol{\lambda}_\rho^{T+1} - \boldsymbol{\lambda}_\rho^*(x^T)\|^2 \leq 2} + 6\rho$$

$$+ \frac{1}{\eta_\lambda T}\sum_{t=0}^{T-1}\big[\mathbb{E}\|\boldsymbol{\lambda}_\rho^{t+2} - \boldsymbol{\lambda}_\rho^*(x^{t+1})\|^2 - \mathbb{E}\|\boldsymbol{\lambda}_\rho^{t+2} - \boldsymbol{\lambda}_\rho^*(x^t)\|^2\big]$$

$$\overset{(17)}{\leq}\frac{8\mathbf{F}(x^0)\mathbf{A}^\top\mathbf{1}}{\eta_g ST} + \frac{28\|\mathbf{A}\|_2^2\sum_{s\in[S]}\mathbb{E}\big[\mathcal{E}_s^0\big]}{\gamma T} + \frac{28L^2\|\mathbf{A}\|_2^2\sum_{s\in[S]}\mathbb{E}\big[\mathcal{C}_s^0\big]}{T} + \frac{10}{\eta_\lambda T}$$

$$- \frac{1}{T}\bigg[2 - 10\eta_\lambda\rho - 5\eta_\lambda SM^2\|\mathbf{A}\|_2^2 - 4\eta_g L\|\mathbf{A}\|_\infty - \frac{56S\eta_g^2 L^2\|\mathbf{A}\|_2^2}{\gamma^2}\bigg]\sum_{t=0}^{T-1}\mathbb{E}\|d^{t+1}\|^2 + 6\rho$$

$$+ 336S\eta_l^2 K^2 L^2\|\mathbf{A}\|_2^2\bigg[\gamma\sigma^2\Big(1 + \frac{16}{K}\Big) + 8G^2\bigg] + \frac{56\gamma S\sigma^2\|\mathbf{A}\|_2^2}{KN}$$

$$+ \frac{1}{\eta_\lambda T} \sum_{t=0}^{T-1} \left[ \mathbb{E}\|\boldsymbol{\lambda}_\rho^{t+2} - \boldsymbol{\lambda}_\rho^*(x^{t+1})\|^2 - \mathbb{E}\|\boldsymbol{\lambda}_\rho^{t+2} - \boldsymbol{\lambda}_\rho^*(x^t)\|^2 \right]$$

$$\leq \frac{8\mathbf{F}(x^0)\mathbf{A}^\top \mathbf{1}}{\eta_g ST} + \frac{28\|\mathbf{A}\|_2^2 \sum_{s\in[S]} \mathbb{E}\left[\mathcal{E}_s^0\right]}{\gamma T} + \frac{28L^2\|\mathbf{A}\|_2^2 \sum_{s\in[S]} \mathbb{E}\left[\mathcal{C}_s^0\right]}{T} + \frac{10}{\eta_\lambda T}$$

$$- \frac{1}{T} \underbrace{\left[ 1 - 10\eta_\lambda\rho - 5\eta_\lambda SM^2\|\mathbf{A}\|_2^2 - 4\eta_g L\|\mathbf{A}\|_\infty - \frac{56S\eta_g^2 L^2\|\mathbf{A}\|_2^2}{\gamma^2} \right]}_{\geq 0} \sum_{t=0}^{T-1} \mathbb{E}\|d^{t+1}\|^2 + 6\rho$$

$$+ 336S\eta_l^2 K^2 L^2\|\mathbf{A}\|_2^2 \left[ \gamma\sigma^2\left(1 + \frac{16}{K}\right) + 8G^2 \right] + \frac{56\gamma S\sigma^2\|\mathbf{A}\|_2^2}{KN} + \frac{25\eta_g^2 G^2 L^2 S^4\|\mathbf{A}\|_2^4}{\eta_\lambda^2 \rho^2},$$

where the last inequality holds due to

$$\frac{1}{\eta_\lambda} \left[ \mathbb{E}\|\boldsymbol{\lambda}_\rho^{t+2} - \boldsymbol{\lambda}_\rho^*(x^{t+1})\|^2 - \mathbb{E}\|\boldsymbol{\lambda}_\rho^{t+2} - \boldsymbol{\lambda}_\rho^*(x^t)\|^2 \right]$$

$$= \frac{1}{\eta_\lambda} \mathbb{E}\|\boldsymbol{\lambda}_\rho^*(x^{t+1}) - \boldsymbol{\lambda}_\rho^*(x^t)\|^2 + \frac{2}{\eta_\lambda} \mathbb{E}\langle \boldsymbol{\lambda}_\rho^{t+2} - \boldsymbol{\lambda}_\rho^*(x^t), \boldsymbol{\lambda}_\rho^*(x^t) - \boldsymbol{\lambda}_\rho^*(x^{t+1}) \rangle$$

$$\leq \frac{1}{\eta_\lambda} \mathbb{E}\|\boldsymbol{\lambda}_\rho^*(x^{t+1}) - \boldsymbol{\lambda}_\rho^*(x^t)\|^2 + \frac{2}{\eta_\lambda} \mathbb{E}\|\boldsymbol{\lambda}_\rho^{t+2} - \boldsymbol{\lambda}_\rho^*(x^t)\| \cdot \mathbb{E}\|\boldsymbol{\lambda}_\rho^*(x^{t+1}) - \boldsymbol{\lambda}_\rho^*(x^t)\|$$

$$= \frac{1}{\eta_\lambda} \underbrace{\left[ \mathbb{E}\|\boldsymbol{\lambda}_\rho^*(x^{t+1}) - \boldsymbol{\lambda}_\rho^*(x^t)\| + 2\mathbb{E}\|\boldsymbol{\lambda}_\rho^{t+2} - \boldsymbol{\lambda}_\rho^*(x^t)\| \right]}_{\leq 3\sqrt{2} < 5} \cdot \underbrace{\mathbb{E}\|\boldsymbol{\lambda}_\rho^*(x^{t+1}) - \boldsymbol{\lambda}_\rho^*(x^t)\|}_{\overset{\text{Lemma } B.7}{\leq} \frac{2SG}{\rho}\|\mathbf{A}\|_2^2 \cdot \mathbb{E}\|\nabla\mathbf{F}(x^{t+1}) - \nabla\mathbf{F}(x^t)\|_2}$$

$$\leq \frac{10GLS^2\|\mathbf{A}\|_2^2}{\eta_\lambda \rho} \cdot \mathbb{E}\|x^{t+1} - x^t\| \overset{(8h)}{=} 2 \cdot \frac{5\eta_g GLS^2\|\mathbf{A}\|_2^2}{\eta_\lambda \rho} \cdot \mathbb{E}\|d^{t+1}\|$$

$$\overset{(9b)}{\leq} \frac{25\eta_g^2 G^2 L^2 S^4\|\mathbf{A}\|_2^4}{\eta_\lambda^2 \rho^2} + \mathbb{E}\|d^{t+1}\|^2.$$

Finally, we arrive at the following bound:

$$\frac{1}{T} \sum_{t=0}^{T-1} \mathbb{E}\|\nabla\mathbf{F}(x^t)\mathbf{A}^\top \boldsymbol{\lambda}_0^*(x^t) - d^{t+1}\|^2$$

$$\leq \frac{2}{T} \sum_{t=0}^{T-1} \mathbb{E}\|\nabla\mathbf{F}(x^t)\mathbf{A}^\top \boldsymbol{\lambda}_\rho^*(x^t) - d^{t+1}\|^2 + 2\rho$$

$$\leq \frac{16\mathbf{F}(x^0)\mathbf{A}^\top \boldsymbol{\lambda}}{\eta_g T} + \frac{56\|\mathbf{A}\|_2^2 \sum_{s\in[S]} \mathbb{E}\left[\mathcal{E}_s^0\right]}{\gamma T} + \frac{56L^2\|\mathbf{A}\|_2^2 \sum_{s\in[S]} \mathbb{E}\left[\mathcal{C}_s^0\right]}{T} + \frac{20}{\eta_\lambda T}$$

$$+ 672S\eta_l^2 K^2 L^2\|\mathbf{A}\|_2^2 \left[ \gamma\sigma^2\left(1 + \frac{16}{K}\right) + 8G^2 \right] + \frac{112\gamma S\sigma^2\|\mathbf{A}\|_2^2}{KN} + \frac{50\eta_g^2 G^2 L^2 S^4\|\mathbf{A}\|_2^4}{\eta_\lambda^2 \rho^2} + 14\rho$$

$$= \mathcal{O}\left( \frac{1}{\eta_g T} + \frac{1}{\gamma T} + \frac{\gamma\eta_l^2}{T} + \frac{1}{\eta_\lambda T} \right) + \mathcal{O}\left( \eta_l^2 + \frac{\gamma}{N} + \frac{\eta_g^2}{\eta_\lambda^2 \rho^2} + \rho \right),$$

which completes the proof. $\qquad\square$

For the Exact-Update case, the bound shares the same structure as the convergence result; thus, applying the same parameter setting yields the optimal $\mathcal{O}(1/\sqrt{NT})$ rate, implying that the update direction aligns with the ideal CA direction as fast as stationarity is achieved. In contrast, the Step-Update case involves a trade-off between tracking error and regularization bias. To balance the constraints, we adopt the configuration $\eta_g = \mathcal{O}(T^{-3/5})$, $\gamma = \mathcal{O}(T^{-2/5})$, $\rho = \mathcal{O}(T^{-2/5})$, which secures a convergence rate of $\mathcal{O}(T^{-2/5})$ for the CA direction distance.

## G. Convergence Analysis in the Strongly Convex Setting

In this section, we analyze the convergence behavior under the strong convexity assumption. We prove Theorem 4.13 by constructing a unified Lyapunov function

$$\varphi^t = \mathbb{E}\|x^t - x^*\|^2 + \gamma \sum_{s\in[S]} \left[ \mathbb{E}\left[\mathcal{E}_s^t\right] + \gamma(1+\gamma)L^2\mathbb{E}\left[\mathcal{C}_s^t\right] \right],$$

that jointly tracks the distance to the Pareto optimal solution, the gradient estimation error, and the client drift. We demonstrate the geometric contraction of this Lyapunov function, establishing the $\tilde{\mathcal{O}}(1/\sqrt{T})$ convergence rate for the weighted sub-optimality gap.

**Theorem G.1** (DREAM in the Strongly Convex Setting). *Under Assumptions 4.1, 4.2, 4.4, suppose that parameters satisfy*

$$768(1+\gamma)\gamma^2\eta_l^2K^2L^4 + 12SL^2\eta_g^2\left(1+2SL^2\right)\|\mathbf{A}\|_2^2 \le \eta_g\mu\nu_A,$$

$$24\gamma(1-\gamma^2)\eta_l^2K^2L^2 + \frac{\eta_g\mu\nu_A}{2} + \frac{4\eta_g\|\mathbf{A}\|_2^2}{\gamma\mu\nu_A} + \frac{6\eta_g^2\left(1+2SL^2\right)\|\mathbf{A}\|_2^2}{\gamma} \le \gamma(1+\gamma),$$

$$\eta_g\mu\nu_A + 768\gamma\eta_l^2K^2L^2 \le 4.$$

*Consider the sequence generated by (8), there exists $0 \le r \le T - 1$ such that*

$$\mathbb{E}[\mathbf{F}(x^r) - \mathbf{F}(x^*)]\mathbf{A}^\top\boldsymbol{\lambda}_\rho^{r+1} \le \mathcal{O}\left(\exp\left(-\frac{\eta_g\mu\nu_AT}{4}\right)\right) + \mathcal{O}\left(\eta_g + \frac{\gamma^2\eta_l^2}{\eta_g} + \frac{\gamma^3}{\eta_g}\right).$$

*It also holds that*

$$\varphi^{t+1} \le \left(1 - \frac{\eta_g\mu\nu_A}{4}\right)\varphi^t + \mathcal{O}\left(\eta_g + \gamma^2\eta_l^2 + \gamma^3\right).$$

*Incorporating the additional Assumptions 4.3 and 4.5 allows us to refine the error term as follows:*

- **Exact-Update.** *If the weights are updated using the exact regularized QP solver, and step sizes satisfy*

$$4\sqrt{3}SG\|\nabla\mathbf{F}(x^*)\mathbf{A}^\top\|_2\|\mathbf{A}\|_2^2 \le \rho\mu\nu_A,$$

$$\frac{\eta_g\mu\nu_A}{\gamma}+48(1-\gamma^2)\eta_l^2K^2L^2+\frac{8\eta_g\|\mathbf{A}\|_2^2}{\gamma^2\mu\nu_A}\left[1+\frac{3S\left(G^2+M^2\right)\|\nabla\mathbf{F}(x^*)\mathbf{A}^\top\|_2^2\|\mathbf{A}\|_2^2}{\rho^2}+\eta_g(1+2SL^2)\mu\nu_A\right] \le 2(1+\gamma),$$

$$\eta_g\mu\nu_A + 768\gamma\eta_l^2K^2L^2 \le 4,$$

*it holds that*

$$\varphi^{t+1} \le \left(1 - \frac{\eta_g\mu\nu_A}{4}\right)\varphi^t + \mathcal{O}\left(\eta_g\rho^2 + \eta_g^2 + \gamma^2\eta_l^2 + \gamma^3\right).$$

- **Step-Update.** *If the weights are updated using the one-step projected gradient descent, and step sizes satisfy*

$$8SG\|\nabla\mathbf{F}(x^*)\mathbf{A}^\top\|_2\|\mathbf{A}\|_2^2 \le \rho\mu\nu_A,$$

$$\frac{\eta_g\mu\nu_A}{\gamma}+48(1-\gamma^2)\eta_l^2K^2L^2+\frac{8\eta_g\|\mathbf{A}\|_2^2}{\gamma^2\mu\nu_A}\left[1+\frac{4S\left(G^2+M^2\right)\|\nabla\mathbf{F}(x^*)\mathbf{A}^\top\|_2^2\|\mathbf{A}\|_2^2}{\rho^2}+\eta_g(1+2SL^2)\mu\nu_A\right] \le 2(1+\gamma),$$

$$\eta_g\mu\nu_A \le \min\{4 - 768\gamma\eta_l^2K^2L^2, \eta_\lambda^2\rho^2\},$$

$$2\eta_\lambda\left[S^2M^4\|\mathbf{A}\|_2^4 + \rho^2\right] \le \rho,$$

*it holds that*

$$\varphi^{t+1} + \frac{8\eta_g\|\nabla\mathbf{F}(x^*)\mathbf{A}^\top\|_2^2}{\eta_\lambda\rho\mu\nu_A}\mathbb{E}\left\|\boldsymbol{\lambda}_\rho^{t+1} - \boldsymbol{\lambda}_{\rho,*}^{t+1}\right\|^2 \le \left(1 - \frac{\eta_g\mu\nu_A}{4}\right)\left[\varphi^t + \frac{8\eta_g\|\nabla\mathbf{F}(x^*)\mathbf{A}^\top\|_2^2}{\eta_\lambda\rho\mu\nu_A}\mathbb{E}\left\|\boldsymbol{\lambda}_\rho^t - \boldsymbol{\lambda}_{\rho,*}^t\right\|^2\right]$$

$$+ \mathcal{O}\left(\eta_g\rho^2 + \frac{\eta_g}{\eta_\lambda^2\rho^4} + \eta_g^2 + \gamma^2\eta_l^2 + \gamma^3\right).$$

*Proof.* We begin by analyzing the distance to an arbitrary Pareto optimal solution $x^*$. From the update rules, we have:

$$\mathbb{E}\|x^{t+1} - x^*\|^2 \overset{(8h)}{=} \mathbb{E}\left\|x^t - \eta_g d^{t+1} - x^*\right\|^2$$

$$\overset{(8g)}{=} \mathbb{E}\|x^t - x^*\|^2 - 2\eta_g \mathbb{E}\left\langle x^t - x^*, \bar{\boldsymbol{\Delta}}^{t+1} \mathbf{A}^\top \boldsymbol{\lambda}_\rho^{t+1}\right\rangle + \eta_g^2 \mathbb{E}\left\|d^{t+1}\right\|^2$$

$$= \mathbb{E}\|x^t - x^*\|^2 - 2\eta_g \mathbb{E}\left\langle x^t - x^*, \nabla \mathbf{F}(x^t) \mathbf{A}^\top \boldsymbol{\lambda}_\rho^{t+1}\right\rangle$$

$$+ 2\eta_g \mathbb{E}\left\langle x^t - x^*, [\nabla \mathbf{F}(x^t) - \bar{\boldsymbol{\Delta}}^{t+1}] \mathbf{A}^\top \boldsymbol{\lambda}_\rho^{t+1}\right\rangle + \eta_g^2 \mathbb{E}\left\|d^{t+1}\right\|^2. \tag{26}$$

**Convergence of the Weighted Sub-Optimality Gap**

By applying the $\mu$-strong convexity (Assumption 4.4), we can relate the first inner product term to the sub-optimality gap:

$$-2\eta_g \left\langle x^t - x^*, \nabla \mathbf{F}(x^t) \mathbf{A}^\top \boldsymbol{\lambda}_\rho^{t+1}\right\rangle$$

$$= -2\eta_g \sum_{s\in[S]} \lambda_{s,\rho}^{t+1} \sum_{j\in[S]} A_{sj} \cdot \underbrace{\left\langle x^t - x^*, \nabla f_j(x^t)\right\rangle}_{\overset{(12a)}{\geq} f_j(x^t) - f_j(x^*) + \frac{\mu}{2}\|x^t - x^*\|^2}$$

$$\leq -2\eta_g \sum_{s\in[S]} \lambda_{s,\rho}^{t+1} \sum_{j\in[S]} A_{sj} \left[f_j(x^t) - f_j(x^*)\right] - \eta_g \mu \underbrace{\sum_{s\in[S]} \lambda_{s,\rho}^{t+1} \sum_{j\in[S]} A_{sj}}_{\geq \sum_{s\in[S]} \lambda_{s,\rho}^{t+1} \min_{s\in[S]}\left(\sum_{j\in[S]} A_{sj}\right) = \nu_A} \|x^t - x^*\|^2$$

$$\leq -2\eta_g[\mathbf{F}(x^t) - \mathbf{F}(x^*)] \mathbf{A}^\top \boldsymbol{\lambda}_\rho^{t+1} - \eta_g \mu \nu_A \|x^t - x^*\|^2.$$

Substituting this bound into (26), we further have

$$\mathbb{E}\|x^{t+1} - x^*\|^2$$

$$\leq (1 - \eta_g \mu \nu_A)\mathbb{E}\|x^t - x^*\|^2 - 2\eta_g \mathbb{E}[\mathbf{F}(x^t) - \mathbf{F}(x^*)]\mathbf{A}^\top \boldsymbol{\lambda}_\rho^{t+1}$$

$$+ 2\eta_g \mathbb{E}\left\langle x^t - x^*, [\nabla \mathbf{F}(x^t) - \bar{\boldsymbol{\Delta}}^{t+1}]\mathbf{A}^\top \boldsymbol{\lambda}_\rho^{t+1}\right\rangle + \eta_g^2 \mathbb{E}\left\|d^{t+1}\right\|^2$$

$$\overset{(9b)}{\leq} (1 - \eta_g \mu \nu_A)\mathbb{E}\|x^t - x^*\|^2 - 2\eta_g \mathbb{E}[\mathbf{F}(x^t) - \mathbf{F}(x^*)]\mathbf{A}^\top \boldsymbol{\lambda}_\rho^{t+1} + \eta_g^2 \mathbb{E}\left\|d^{t+1}\right\|^2$$

$$+ \frac{\eta_g \mu \nu_A}{2}\mathbb{E}\|x^t - x^*\|^2 + \frac{2\eta_g}{\mu \nu_A} \cdot \mathbb{E} \underbrace{\left\|[\nabla \mathbf{F}(x^t) - \bar{\boldsymbol{\Delta}}^{t+1}]\mathbf{A}^\top \boldsymbol{\lambda}_\rho^{t+1}\right\|^2}_{\leq \sum_{s\in[S]} \lambda_{s,\rho}^{t+1}\left\|\sum_{j\in[S]} A_{sj}[\nabla f_j(x^t) - \bar{\Delta}_j^{t+1}]\right\|^2 \leq \|\mathbf{A}\|_2^2 \sum_{s\in[S]} \mathcal{E}_s^t}$$

$$\leq \left(1 - \frac{\eta_g \mu \nu_A}{2}\right)\mathbb{E}\|x^t - x^*\|^2 - 2\eta_g \mathbb{E}[\mathbf{F}(x^t) - \mathbf{F}(x^*)]\mathbf{A}^\top \boldsymbol{\lambda}_\rho^{t+1} + \frac{2\eta_g\|\mathbf{A}\|_2^2}{\mu\nu_A}\sum_{s\in[S]}\mathbb{E}[\mathcal{E}_s^t] + \eta_g^2 \mathbb{E}\left\|d^{t+1}\right\|^2. \tag{27}$$

Next, by combining inequalities (13) and (15) from Lemmas C.1 and C.2, we can establish a bound for the combined error term, which is a component of function $\varphi^{t+1}$:

$$\mathbb{E}\left[\mathcal{E}_s^{t+1}\right] + \gamma(1+\gamma)L^2\mathbb{E}\left[\mathcal{C}_s^{t+1}\right]$$

$$\leq \left[\frac{2-\gamma-\gamma^2}{2} + 12\gamma(1-\gamma^2)\eta_l^2 K^2 L^2\right]\mathbb{E}\left[\mathcal{E}_s^t\right] + 192(1+\gamma)\gamma^2\eta_l^2 K^2 L^4 \mathbb{E}\left[\mathcal{C}_s^t\right]$$

$$+ \left[\frac{2}{\gamma} - \underbrace{(1 - 192\gamma^2\eta_l^2 K^2 L^2)}_{\geq 0}(1+\gamma)\right]\eta_g^2 L^2 \mathbb{E}\|d^{t+1}\|^2 + 96\gamma(1+\gamma)\eta_l^2 K^2 L^2 \mathbb{E}\underbrace{\left\|\nabla f_s(x^t)\right\|^2}_{\leq 2L^2\|x^t - x^*\|^2 + 2\|\nabla f_s(x^*)\|^2}$$

$$+ 12(1+\gamma)\gamma^2\eta_l^2 K^2 L^2 \sigma^2\left(1 + \frac{16}{K}\right) + \frac{2\gamma^2\sigma^2}{K|\mathcal{R}_s|}. \tag{28}$$

To upper bound $\mathbb{E}\|d^{t+1}\|^2$, we can use the convexity of $\ell_2$ norm and Jensen's inequality:

$$\mathbb{E}\left\|d^{t+1}\right\|^2 = \mathbb{E}\left\|[\bar{\boldsymbol{\Delta}}^{t+1} - \nabla\mathbf{F}(x^t) + \nabla\mathbf{F}(x^t) - \nabla\mathbf{F}(x^*) + \nabla\mathbf{F}(x^*)]\mathbf{A}^\top \boldsymbol{\lambda}_\rho^{t+1}\right\|^2$$

$$\leq \|\mathbf{A}\|_2^2 \cdot \mathbb{E}\left\|\bar{\boldsymbol{\Delta}}^{t+1} - \nabla\mathbf{F}(x^t) + \nabla\mathbf{F}(x^t) - \nabla\mathbf{F}(x^*) + \nabla\mathbf{F}(x^*)\right\|_F^2$$

$$\leq 3\|\mathbf{A}\|_2^2 \sum_{s\in[S]} \mathbb{E}\left[\mathcal{E}_s^t\right] + 3SL^2\|\mathbf{A}\|_2^2 \cdot \mathbb{E}\left\|x^t - x^*\right\|^2 + 3\|\mathbf{A}\|_2^2 \sum_{s\in[S]} \mathbb{E}\left\|\nabla f_s(x^*)\right\|^2. \tag{29}$$

Now, we substitute the bounds for each component back into the function $\varphi^{t+1}$. Summing (28) over $[S]$, and substituting it and (29) to (27), gives the one-step evolution of the entire Lyapunov function:

$$\varphi^{t+1} = \mathbb{E}\|x^{t+1} - x^*\|^2 + \gamma \sum_{s \in [S]} \mathbb{E}\left[\mathcal{E}_s^{t+1}\right] + (1+\gamma)\gamma^2 L^2 \sum_{s \in [S]} \mathbb{E}\left[\mathcal{C}_s^{t+1}\right]$$

$$\leq \left[1 - \frac{\eta_g \mu \nu_A}{2} + 192(1+\gamma)\gamma^2 \eta_l^2 K^2 L^4\right] \mathbb{E}\|x^t - x^*\|^2$$

$$+ \gamma \left[\frac{2 - \gamma - \gamma^2}{2} + 12\gamma(1-\gamma^2)\eta_l^2 K^2 L^2 + \frac{2\eta_g \|\mathbf{A}\|_2^2}{\gamma \mu \nu_A}\right] \sum_{s \in [S]} \mathbb{E}\left[\mathcal{E}_s^t\right]$$

$$+ 192(1+\gamma)\gamma^3 \eta_l^2 K^2 L^4 \sum_{s \in [S]} \mathbb{E}\left[\mathcal{C}_s^t\right] - 2\eta_g \mathbb{E}[\mathbf{F}(x^t) - \mathbf{F}(x^*)]\mathbf{A}^\top \boldsymbol{\lambda}_\rho^{t+1} + \eta_g^2 \left(1 + 2SL^2\right) \mathbb{E}\|d^{t+1}\|^2$$

$$+ 192(1+\gamma)\gamma^2 \eta_l^2 K^2 L^2 \sum_{s \in [S]} \mathbb{E}\|\nabla f_s(x^*)\|^2 + 12(1+\gamma)\gamma^3 \eta_l^2 S K^2 L^2 \sigma^2 \left(1 + \frac{16}{K}\right) + \sum_{s \in [S]} \frac{2\gamma^3 \sigma^2}{K|\mathcal{R}_s|}$$

$$\overset{(29)}{\leq} \underbrace{\left[1 - \frac{\eta_g \mu \nu_A}{2} + 192(1+\gamma)\gamma^2 \eta_l^2 K^2 L^4 + 3SL^2 \eta_g^2 \left(1 + 2SL^2\right) \|\mathbf{A}\|_2^2\right]}_{\leq 1 - \frac{\eta_g \mu \nu_A}{4}} \mathbb{E}\|x^t - x^*\|^2$$

$$+ \gamma \underbrace{\left[1 - \frac{\gamma + \gamma^2}{2} + 12\gamma(1-\gamma^2)\eta_l^2 K^2 L^2 + \frac{2\eta_g \|\mathbf{A}\|_2^2}{\gamma \mu \nu_A} + \frac{3\eta_g^2 \left(1 + 2SL^2\right) \|\mathbf{A}\|_2^2}{\gamma}\right]}_{\leq 1 - \frac{\eta_g \mu \nu_A}{4}} \sum_{s \in [S]} \mathbb{E}\left[\mathcal{E}_s^t\right]$$

$$+ \underbrace{192\gamma \eta_l^2 K^2 L^2}_{\leq 1 - \frac{\eta_g \mu \nu_A}{4}} \cdot (1+\gamma)\gamma^2 L^2 \sum_{s \in [S]} \mathbb{E}\left[\mathcal{C}_s^t\right] - 2\eta_g \mathbb{E}[\mathbf{F}(x^t) - \mathbf{F}(x^*)]\mathbf{A}^\top \boldsymbol{\lambda}_\rho^{t+1}$$

$$+ 3\left[64(1+\gamma)\gamma^2 \eta_l^2 K^2 L^2 + \eta_g^2 \left(1 + 2SL^2\right) \|\mathbf{A}\|_2^2\right] \sum_{s \in [S]} \mathbb{E}\|\nabla f_s(x^*)\|^2$$

$$+ 12(1+\gamma)\gamma^3 \eta_l^2 S K^2 L^2 \sigma^2 \left(1 + \frac{16}{K}\right) + \sum_{s \in [S]} \frac{2\gamma^3 \sigma^2}{K|\mathcal{R}_s|}$$

$$\leq \left(1 - \frac{\eta_g \mu \nu_A}{4}\right) \varphi^t - 2\eta_g \mathbb{E}[\mathbf{F}(x^t) - \mathbf{F}(x^*)]\mathbf{A}^\top \boldsymbol{\lambda}_\rho^{t+1}$$

$$+ 3\left[64(1+\gamma)\gamma^2 \eta_l^2 K^2 L^2 + \eta_g^2 \left(1 + 2SL^2\right) \|\mathbf{A}\|_2^2\right] \sum_{s \in [S]} \mathbb{E}\|\nabla f_s(x^*)\|^2$$

$$+ 12(1+\gamma)\gamma^3 \eta_l^2 S K^2 L^2 \sigma^2 \left(1 + \frac{16}{K}\right) + \sum_{s \in [S]} \frac{2\gamma^3 \sigma^2}{K|\mathcal{R}_s|}.$$

The parameter conditions which should be satisfied are as follows:

$$\frac{768(1+\gamma)\gamma^2 \eta_l^2 K^2 L^4}{\eta_g} + 12\eta_g SL^2 \left(1 + 2SL^2\right) \|\mathbf{A}\|_2^2 \leq \mu \nu_A,$$

$$24(1-\gamma)\eta_l^2 K^2 L^2 + \frac{\eta_g \mu \nu_A}{2\gamma(1+\gamma)} + \frac{4\eta_g \|\mathbf{A}\|_2^2}{(1+\gamma)\gamma^2 \mu \nu_A} + \frac{6\eta_g^2 \left(1 + 2SL^2\right) \|\mathbf{A}\|_2^2}{(1+\gamma)\gamma^2} \leq 1,$$

$$\eta_g \mu \nu_A + 768\gamma \eta_l^2 K^2 L^2 \leq 4.$$

Rearranging the inequality, it follows that

$$\mathbb{E}[\mathbf{F}(x^t) - \mathbf{F}(x^*)]\mathbf{A}^\top \boldsymbol{\lambda}_\rho^{t+1}$$

$$\leq \frac{1}{2\eta_g} \left[\left(1 - \frac{\eta_g \mu \nu_A}{4}\right) \varphi^t - \varphi^{t+1}\right] + \frac{6(1+\gamma)\gamma^3 \eta_l^2 S K^2 L^2 \sigma^2}{\eta_g} \left(1 + \frac{16}{K}\right) + \sum_{s \in [S]} \frac{\gamma^3 \sigma^2}{\eta_g K|\mathcal{R}_s|}$$

$$+ \frac{3}{2\eta_g} \left[64(1+\gamma)\gamma^2 \eta_l^2 K^2 L^2 + \eta_g^2 \left(1 + 2SL^2\right) \|\mathbf{A}\|_2^2\right] \sum_{s \in [S]} \mathbb{E}\|\nabla f_s(x^*)\|^2.$$

We define $\omega^t = (1 - \eta_g \mu \nu_A / 4)^{-t}$ and sum the above inequality from $t = 0$ to $T - 1$, weighted by $\omega^t$. This specific weighting allows for a telescoping sum. After summing and rearranging the terms, we arrive at a bound for the weighted cumulative sub-optimality gap:

$$\frac{1}{\sum_{t=0}^{T-1} \omega^t} \cdot \sum_{t=0}^{T-1} \omega^t \cdot \mathbb{E}[\mathbf{F}(x^t) - \mathbf{F}(x^*)]\mathbf{A}^\top \boldsymbol{\lambda}_\rho^{t+1}$$

$$\leq \frac{(2 - \eta_g \mu \nu_A)\,\varphi^0}{8\eta_g \sum_{t=0}^{T-1} \omega^t} + \frac{6(1 + \gamma)\gamma^3 \eta_l^2 SK^2 L^2 \sigma^2}{\eta_g}\left(1 + \frac{16}{K}\right) + \sum_{s \in [S]} \frac{\gamma^3 \sigma^2}{\eta_g K |\mathcal{R}_s|}$$

$$+ \frac{3}{2\eta_g}\left[64(1 + \gamma)\gamma^2 \eta_l^2 K^2 L^2 + \eta_g^2\left(1 + 2SL^2\right)\|\mathbf{A}\|_2^2\right] \sum_{s \in [S]} \mathbb{E}\|\nabla f_s(x^*)\|^2$$

$$= \mathcal{O}\left(\exp\left(-\frac{\eta_g \mu \nu_A T}{4}\right)\varphi^0\right) + \mathcal{O}\left(\eta_g + \frac{\gamma^2 \eta_l^2}{\eta_g} + \frac{\gamma^3}{\eta_g}\right).$$

**General Lyapunov Convergence via Bounded Weights**

Alternatively, instead of relating the first inner product term in (26) to the sub-optimality gap (as done in the previous derivation), we can derive a different upper bound that depends explicitly on the weight approximation error:

$$-2\eta_g \left\langle x^t - x^*, \nabla \mathbf{F}(x^t)\mathbf{A}^\top \boldsymbol{\lambda}_\rho^{t+1} \right\rangle$$

$$= -2\eta_g \left\langle x^t - x^*, [\nabla \mathbf{F}(x^t) - \nabla \mathbf{F}(x^*)]\mathbf{A}^\top \boldsymbol{\lambda}_\rho^{t+1} \right\rangle - 2\eta_g \left\langle x^t - x^*, \nabla \mathbf{F}(x^*)\mathbf{A}^\top \boldsymbol{\lambda}_\rho^{t+1} \right\rangle$$

$$= -2\eta_g \sum_{s \in [S]} \lambda_{s,\rho}^{t+1} \sum_{j \in [S]} A_{sj} \cdot \underbrace{\left\langle x^t - x^*, \nabla f_j(x^t) - \nabla f_j(x^*) \right\rangle}_{\geq \mu \|x^t - x^*\|^2} - 2\eta_g \big\langle x^t - x^*, \nabla \mathbf{F}(x^*)\mathbf{A}^\top \boldsymbol{\lambda}_\rho^{t+1} - \underbrace{\nabla \mathbf{F}(x^*)\mathbf{A}^\top \boldsymbol{\lambda}_0^*(x^*)}_{=0} \big\rangle$$

$$\leq -2\eta_g \mu \underbrace{\sum_{s \in [S]} \lambda_{s,\rho}^{t+1} \sum_{j \in [S]} A_{sj}}_{\geq \sum_{s \in [S]} \lambda_{s,\rho}^{t+1} \min_{s \in [S]}(\sum_{j \in [S]} A_{sj}) = \nu_A} \|x^t - x^*\|^2 - 2\eta_g \big\langle x^t - x^*, \nabla \mathbf{F}(x^*)\mathbf{A}^\top \left[\boldsymbol{\lambda}_\rho^{t+1} - \boldsymbol{\lambda}_0^*(x^*)\right] \big\rangle$$

$$\overset{(9b)}{\leq} -\eta_g \mu \nu_A \|x^t - x^*\|^2 + \frac{\eta_g \|\nabla \mathbf{F}(x^*)\mathbf{A}^\top\|_2^2}{\mu \nu_A}\|\boldsymbol{\lambda}_\rho^{t+1} - \boldsymbol{\lambda}_0^*(x^*)\|^2.$$

Substituting this bound back into (26) yields the following recursive inequality:

$$\mathbb{E}\|x^{t+1} - x^*\|^2 \leq (1 - \eta_g \mu \nu_A)\,\mathbb{E}\|x^t - x^*\|^2 + 2\eta_g \mathbb{E}\left\langle x^t - x^*, [\nabla \mathbf{F}(x^t) - \bar{\boldsymbol{\Delta}}^{t+1}]\mathbf{A}^\top \boldsymbol{\lambda}_\rho^{t+1}\right\rangle$$

$$+ \eta_g^2 \mathbb{E}\left\|d^{t+1}\right\|^2 + \frac{\eta_g \|\nabla \mathbf{F}(x^*)\mathbf{A}^\top\|_2^2}{\mu \nu_A}\mathbb{E}\|\boldsymbol{\lambda}_\rho^{t+1} - \boldsymbol{\lambda}_0^*(x^*)\|^2$$

$$\leq \left(1 - \frac{\eta_g \mu \nu_A}{2}\right)\mathbb{E}\|x^t - x^*\|^2 + \frac{2\eta_g \|\mathbf{A}\|_2^2}{\mu \nu_A}\sum_{s \in [S]} \mathbb{E}[\mathcal{E}_s^t]$$

$$+ \eta_g^2 \mathbb{E}\left\|d^{t+1}\right\|^2 + \frac{\eta_g \|\nabla \mathbf{F}(x^*)\mathbf{A}^\top\|_2^2}{\mu \nu_A}\mathbb{E}\|\boldsymbol{\lambda}_\rho^{t+1} - \boldsymbol{\lambda}_0^*(x^*)\|^2. \tag{30}$$

It is analogous to (27) but with a key difference: the sub-optimality gap term $\mathbb{E}[\mathbf{F}(x^t) - \mathbf{F}(x^*)]\mathbf{A}^\top \boldsymbol{\lambda}_\rho^{t+1}$ is replaced by the weight error term $\mathbb{E}\|\boldsymbol{\lambda}_\rho^{t+1} - \boldsymbol{\lambda}_0^*(x^*)\|^2$. This error term has a fundamental property: regardless of the specific update rule employed for $\boldsymbol{\lambda}$, the weight vector always resides within the probability simplex $\omega$. Consequently, the squared error term is uniformly bounded by the diameter of the simplex, i.e., $\|\boldsymbol{\lambda}_\rho^{t+1} - \boldsymbol{\lambda}_0^*(x^*)\|^2 \leq 2$. By treating this term as a bounded constant, we can establish a general convergence guarantee that applies to any valid weight update scheme.

Since inequalities (28) and (29) remain identical in this case, and the parameter constraints are unchanged, we can directly combine them with (30). By substituting the uniform bound and following the same Lyapunov construction steps, we readily arrive at the following general convergence bound:

$$\varphi^{t+1} \leq \left(1 - \frac{\eta_g \mu \nu_A}{4}\right)\varphi^t + \frac{2\eta_g \|\nabla \mathbf{F}(x^*)\mathbf{A}^\top\|_2^2}{\mu \nu_A} + 12(1 + \gamma)\gamma^3 \eta_l^2 SK^2 L^2 \sigma^2\left(1 + \frac{16}{K}\right) + \sum_{s \in [S]} \frac{2\gamma^3 \sigma^2}{K |\mathcal{R}_s|}$$

$$+ 3 \left[ 64(1+\gamma)\gamma^2\eta_l^2 K^2 L^2 + \eta_g^2 \left(1 + 2SL^2\right) \|\mathbf{A}\|_2^2 \right] \sum_{s \in [S]} \mathbb{E} \|\nabla f_s(x^*)\|^2$$

$$= \left(1 - \frac{\eta_g \mu \nu_A}{4}\right) \varphi^t + \mathcal{O}\left(\eta_g + \gamma^2\eta_l^2 + \gamma^3\right).$$

**Refined Convergence Analysis for Exact-Update**

To derive a sharper convergence rate and quantify the impact of the regularization parameter $\rho$, we now perform a fine-grained analysis of the weight approximation error $\mathbb{E}\|\boldsymbol{\lambda}_\rho^{t+1} - \boldsymbol{\lambda}_0^*(x^*)\|^2$ specifically for the Exact-Update strategy. It is important to note that the following derivation relies on Assumptions 4.3, 4.5, and the strong convexity of the subproblem (requiring $\rho > 0$), which ensures the Lipschitz continuity of the regularized weight map (Lemma B.7).

For the Exact-Update case, the algorithmic weight is the optimal solution to the current subproblem, i.e., $\boldsymbol{\lambda}_\rho^{t+1} = \boldsymbol{\lambda}_{\rho,*}^{t+1}$. We decompose the total error into three distinct components:

$$\mathbb{E} \left\| \boldsymbol{\lambda}_\rho^{t+1} - \boldsymbol{\lambda}_0^*(x^*) \right\|^2 = \mathbb{E} \left\| \boldsymbol{\lambda}_{\rho,*}^{t+1} - \boldsymbol{\lambda}_0^*(x^*) \right\|^2$$

$$= \mathbb{E} \left\| \boldsymbol{\lambda}_{\rho,*}^{t+1} - \boldsymbol{\lambda}_\rho^*(x^t) + \boldsymbol{\lambda}_\rho^*(x^t) - \boldsymbol{\lambda}_\rho^*(x^*) + \boldsymbol{\lambda}_\rho^*(x^*) - \boldsymbol{\lambda}_0^*(x^*) \right\|^2$$

$$\leq 3\mathbb{E} \left\| \boldsymbol{\lambda}_{\rho,*}^{t+1} - \boldsymbol{\lambda}_\rho^*(x^t) \right\|^2 + 3\mathbb{E} \left\| \boldsymbol{\lambda}_\rho^*(x^t) - \boldsymbol{\lambda}_\rho^*(x^*) \right\|^2 + 3 \left\| \boldsymbol{\lambda}_\rho^*(x^*) - \boldsymbol{\lambda}_0^*(x^*) \right\|^2.$$

The third term represents the regularization bias at the optimal point $x^*$. Since it depends only on the problem geometry and $\rho$, it can be treated as a controlled constant (vanishing as $\rho \to 0$). The first and second terms arise from the sensitivity of the optimal weights to changes in their input gradients. To bound these two terms, we invoke Lemmas B.6, B.7, which yields:

$$\mathbb{E} \left\| \boldsymbol{\lambda}_{\rho,*}^{t+1} - \boldsymbol{\lambda}_\rho^*(x^t) \right\|^2 \leq \frac{\left(\mathbb{E}\|\nabla\mathbf{F}(x^t)\|_2 + \mathbb{E}\|\bar{\boldsymbol{\Delta}}^{t+1}\|_2\right)^2 \|\mathbf{A}\|_2^4}{\rho^2} \cdot \mathbb{E}\|\nabla\mathbf{F}(x^t) - \bar{\boldsymbol{\Delta}}^{t+1}\|_2^2$$

$$\leq \frac{2\left(\mathbb{E}\|\nabla\mathbf{F}(x^t)\|_2^2 + \mathbb{E}\|\bar{\boldsymbol{\Delta}}^{t+1}\|_2^2\right)\|\mathbf{A}\|_2^4}{\rho^2} \cdot \mathbb{E}\|\nabla\mathbf{F}(x^t) - \bar{\boldsymbol{\Delta}}^{t+1}\|_2^2$$

$$\leq \frac{2S\left(G^2 + M^2\right)\|\mathbf{A}\|_2^4}{\rho^2} \cdot \sum_{s \in [S]} \mathbb{E}\|\nabla f_s(x^t) - \bar{\Delta}_s^{t+1}\|^2 = \frac{2S\left(G^2 + M^2\right)\|\mathbf{A}\|_2^4}{\rho^2} \cdot \sum_{s \in [S]} \mathbb{E}[\mathcal{E}_s^t],$$

$$\mathbb{E} \left\| \boldsymbol{\lambda}_\rho^*(x^t) - \boldsymbol{\lambda}_\rho^*(x^*) \right\|^2 \leq \frac{\left(\mathbb{E}\|\nabla\mathbf{F}(x^t)\|_2 + \mathbb{E}\|\nabla\mathbf{F}(x^*)\|_2\right)^2 \|\mathbf{A}\|_2^4}{\rho^2} \cdot \mathbb{E}\|\nabla\mathbf{F}(x^t) - \nabla\mathbf{F}(x^*)\|_2^2$$

$$\leq \frac{2\left(\mathbb{E}\|\nabla\mathbf{F}(x^t)\|_2^2 + \mathbb{E}\|\nabla\mathbf{F}(x^*)\|_2^2\right)\|\mathbf{A}\|_2^4}{\rho^2} \cdot \mathbb{E}\|\nabla\mathbf{F}(x^t) - \nabla\mathbf{F}(x^*)\|_2^2$$

$$\leq \frac{4SG^2\|\mathbf{A}\|_2^4}{\rho^2} \cdot \sum_{s \in [S]} \mathbb{E}\|x^t - x^*\|^2 = \frac{4S^2G^2\|\mathbf{A}\|_2^4}{\rho^2} \cdot \mathbb{E}\|x^t - x^*\|^2.$$

Substituting them back into (30) yields

$$\mathbb{E}\|x^{t+1} - x^*\|^2 \leq \left(1 - \frac{\eta_g \mu \nu_A}{2} + \frac{12\eta_g S^2 G^2 \|\nabla\mathbf{F}(x^*)\mathbf{A}^\top\|_2^2 \|\mathbf{A}\|_2^4}{\mu \nu_A \rho^2}\right) \mathbb{E}\|x^t - x^*\|^2$$

$$+ \frac{\eta_g \|\mathbf{A}\|_2^2}{\mu \nu_A} \left[2 + \frac{6S\left(G^2 + M^2\right)\|\nabla\mathbf{F}(x^*)\mathbf{A}^\top\|_2^2 \|\mathbf{A}\|_2^2}{\rho^2}\right] \sum_{s \in [S]} \mathbb{E}[\mathcal{E}_s^t]$$

$$+ \eta_g^2 \mathbb{E}\left\|d^{t+1}\right\|^2 + \frac{3\eta_g \|\nabla\mathbf{F}(x^*)\mathbf{A}^\top\|_2^2}{\mu \nu_A} \left\|\boldsymbol{\lambda}_\rho^*(x^*) - \boldsymbol{\lambda}_0^*(x^*)\right\|^2.$$

By combining the above inequality and (28), we have

$$\varphi^{t+1} = \mathbb{E}\|x^{t+1} - x^*\|^2 + \gamma \sum_{s \in [S]} \mathbb{E}\left[\mathcal{E}_s^{t+1}\right] + (1+\gamma)\gamma^2 L^2 \sum_{s \in [S]} \mathbb{E}\left[\mathcal{C}_s^{t+1}\right]$$

$$
\leq \left[ 1 - \frac{\eta_g \mu \nu_A}{2} + \frac{12 \eta_g S^2 G^2 \|\nabla \mathbf{F}(x^*) \mathbf{A}^\top\|_2^2 \|\mathbf{A}\|_2^4}{\mu \nu_A \rho^2} \right] \mathbb{E}\|x^t - x^*\|^2
$$

$$
+ \gamma \left[ \frac{2 - \gamma - \gamma^2}{2} + 12 \gamma (1 - \gamma^2) \eta_l^2 K^2 L^2 + \frac{2 \eta_g \|\mathbf{A}\|_2^2}{\gamma \mu \nu_A} + \frac{6 \eta_g S \left( G^2 + M^2 \right) \|\nabla \mathbf{F}(x^*) \mathbf{A}^\top\|_2^2 \|\mathbf{A}\|_2^4}{\gamma \mu \nu_A \rho^2} \right] \sum_{s \in [S]} \mathbb{E}\left[\mathcal{E}_s^t\right]
$$

$$
+ 192 (1 + \gamma) \gamma^3 \eta_l^2 K^2 L^4 \sum_{s \in [S]} \mathbb{E}\left[\mathcal{C}_s^t\right] + \eta_g^2 \left( 1 + 2SL^2 \right) \mathbb{E}\|d^{t+1}\|^2 + \frac{3 \eta_g \|\nabla \mathbf{F}(x^*) \mathbf{A}^\top\|_2^2}{\mu \nu_A} \left\| \boldsymbol{\lambda}_\rho^*(x^*) - \boldsymbol{\lambda}_0^*(x^*) \right\|^2
$$

$$
+ 96 (1 + \gamma) \gamma^2 \eta_l^2 S K^2 L^2 G^2 + 12 (1 + \gamma) \gamma^3 \eta_l^2 S K^2 L^2 \sigma^2 \left( 1 + \frac{16}{K} \right) + \sum_{s \in [S]} \frac{2 \gamma^3 \sigma^2}{K |\mathcal{R}_s|}
$$

$$
\leq \underbrace{\left[ 1 - \frac{\eta_g \mu \nu_A}{2} + \frac{12 \eta_g S^2 G^2 \|\nabla \mathbf{F}(x^*) \mathbf{A}^\top\|_2^2 \|\mathbf{A}\|_2^4}{\mu \nu_A \rho^2} \right]}_{\leq 1 - \frac{\eta_g \mu \nu_A}{4}} \mathbb{E}\|x^t - x^*\|^2
$$

$$
+ \gamma \underbrace{\left[ 1 - \frac{\gamma + \gamma^2}{2} + 12 \gamma (1 - \gamma^2) \eta_l^2 K^2 L^2 + \frac{2 \eta_g \|\mathbf{A}\|_2^2}{\gamma \mu \nu_A} \left[ 1 + \frac{3S \left( G^2 + M^2 \right) \|\nabla \mathbf{F}(x^*) \mathbf{A}^\top\|_2^2 \|\mathbf{A}\|_2^2}{\rho^2} + \eta_g (1 + 2SL^2) \mu \nu_A \right] \right]}_{\leq 1 - \frac{\eta_g \mu \nu_A}{4}}
$$

$$
\cdot \sum_{s \in [S]} \mathbb{E}\left[\mathcal{E}_s^t\right] + \underbrace{192 \gamma \eta_l^2 K^2 L^2}_{\leq 1 - \frac{\eta_g \mu \nu_A}{4}} \cdot (1 + \gamma) \gamma^2 L^2 \sum_{s \in [S]} \mathbb{E}\left[\mathcal{C}_s^t\right] + \frac{3 \eta_g \|\nabla \mathbf{F}(x^*) \mathbf{A}^\top\|_2^2}{\mu \nu_A} \left\| \boldsymbol{\lambda}_\rho^*(x^*) - \boldsymbol{\lambda}_0^*(x^*) \right\|^2
$$

$$
+ 2 \eta_g^2 (1 + 2SL^2) S G^2 \|\mathbf{A}\|_2^2 + 96 (1 + \gamma) \gamma^2 \eta_l^2 S K^2 L^2 G^2 + 12 (1 + \gamma) \gamma^3 \eta_l^2 S K^2 L^2 \sigma^2 \left( 1 + \frac{16}{K} \right) + \sum_{s \in [S]} \frac{2 \gamma^3 \sigma^2}{K |\mathcal{R}_s|}
$$

$$
\leq \left( 1 - \frac{\eta_g \mu \nu_A}{4} \right) \varphi^t + \frac{3 \eta_g \|\nabla \mathbf{F}(x^*) \mathbf{A}^\top\|_2^2}{\mu \nu_A} \left\| \boldsymbol{\lambda}_\rho^*(x^*) - \boldsymbol{\lambda}_0^*(x^*) \right\|^2 + 2 \eta_g^2 (1 + 2SL^2) S G^2 \|\mathbf{A}\|_2^2
$$

$$
+ 96 (1 + \gamma) \gamma^2 \eta_l^2 S K^2 L^2 G^2 + 12 (1 + \gamma) \gamma^3 \eta_l^2 S K^2 L^2 \sigma^2 \left( 1 + \frac{16}{K} \right) + \sum_{s \in [S]} \frac{2 \gamma^3 \sigma^2}{K |\mathcal{R}_s|}
$$

$$
= \left( 1 - \frac{\eta_g \mu \nu_A}{4} \right) \varphi^t + \mathcal{O}\left( \eta_g \rho^2 + \eta_g^2 + \gamma^2 \eta_l^2 + \gamma^3 \right),
$$

where the second inequality holds by using

$$
\begin{aligned}
\mathbb{E}\left\| d^{t+1} \right\|^2 &= \mathbb{E}\left\| [\bar{\boldsymbol{\Delta}}^{t+1} - \nabla \mathbf{F}(x^t) + \nabla \mathbf{F}(x^t)] \mathbf{A}^\top \boldsymbol{\lambda}_\rho^{t+1} \right\|^2 \\
&\leq \|\mathbf{A}\|_2^2 \cdot \mathbb{E}\left\| \bar{\boldsymbol{\Delta}}^{t+1} - \nabla \mathbf{F}(x^t) + \nabla \mathbf{F}(x^t) \right\|_F^2 \\
&\leq 2\|\mathbf{A}\|_2^2 \sum_{s \in [S]} \mathbb{E}\left[\mathcal{E}_s^t\right] + 2\|\mathbf{A}\|_2^2 \sum_{s \in [S]} \mathbb{E}\left\| \nabla f_s(x^t) \right\|^2 \leq 2\|\mathbf{A}\|_2^2 \sum_{s \in [S]} \mathbb{E}\left[\mathcal{E}_s^t\right] + 2SG^2 \|\mathbf{A}\|_2^2.
\end{aligned}
$$

The parameter conditions are summarized as follows:

$$
4\sqrt{3} SG \|\nabla \mathbf{F}(x^*) \mathbf{A}^\top\|_2 \|\mathbf{A}\|_2^2 \leq \rho \mu \nu_A,
$$

$$
\frac{\eta_g \mu \nu_A}{\gamma} + 48 (1 - \gamma^2) \eta_l^2 K^2 L^2 + \frac{8 \eta_g \|\mathbf{A}\|_2^2}{\gamma^2 \mu \nu_A} \left[ 1 + \frac{3S \left( G^2 + M^2 \right) \|\nabla \mathbf{F}(x^*) \mathbf{A}^\top\|_2^2 \|\mathbf{A}\|_2^2}{\rho^2} + \eta_g (1 + 2SL^2) \mu \nu_A \right] \leq 2(1 + \gamma),
$$

$$
\eta_g \mu \nu_A + 768 \gamma \eta_l^2 K^2 L^2 \leq 4.
$$

### Refined Convergence Analysis for Step-Update

In the Step-Update scenario, the algorithmic weights $\boldsymbol{\lambda}_\rho^{t+1}$ are obtained via a single gradient step and may not fully converge to the current subproblem optimum $\boldsymbol{\lambda}_{\rho,*}^{t+1}$. To account for this additional discrepancy, we extend the error decomposition by

introducing this optimization tracking error as a new component:

$$\mathbb{E}\left\|\boldsymbol{\lambda}_\rho^{t+1} - \boldsymbol{\lambda}_0^*(x^*)\right\|^2 = \mathbb{E}\left\|\boldsymbol{\lambda}_\rho^{t+1} - \boldsymbol{\lambda}_{\rho,*}^{t+1} + \boldsymbol{\lambda}_{\rho,*}^{t+1} - \boldsymbol{\lambda}_\rho^*(x^t) + \boldsymbol{\lambda}_\rho^*(x^t) - \boldsymbol{\lambda}_0^*(x^t) + \boldsymbol{\lambda}_0^*(x^t) - \boldsymbol{\lambda}_0^*(x^*)\right\|^2$$

$$\leq 4\mathbb{E}\|\boldsymbol{\lambda}_\rho^{t+1} - \boldsymbol{\lambda}_{\rho,*}^{t+1}\|^2 + 4\mathbb{E}\left\|\boldsymbol{\lambda}_{\rho,*}^{t+1} - \boldsymbol{\lambda}_\rho^*(x^t)\right\|^2 + 4\mathbb{E}\left\|\boldsymbol{\lambda}_\rho^*(x^t) - \boldsymbol{\lambda}_\rho^*(x^*)\right\|^2 + 4\left\|\boldsymbol{\lambda}_\rho^*(x^*) - \boldsymbol{\lambda}_0^*(x^*)\right\|^2.$$

The last three terms are identical in structure to those analyzed in the Exact-Update case (scaled by a factor of 4 instead of 3), and thus share the same upper bounds. The tracking error can be decoupled by using Young's inequality:

$$\mathbb{E}\left\|\boldsymbol{\lambda}_\rho^{t+1} - \boldsymbol{\lambda}_{\rho,*}^{t+1}\right\|^2 \leq \left(1 + \frac{\eta_\lambda\rho}{2}\right)\mathbb{E}\left\|\boldsymbol{\lambda}_\rho^{t+1} - \boldsymbol{\lambda}_{\rho,*}^t\right\|^2 + \left(1 + \frac{2}{\eta_\lambda\rho}\right)\mathbb{E}\left\|\boldsymbol{\lambda}_{\rho,*}^{t+1} - \boldsymbol{\lambda}_{\rho,*}^t\right\|^2.$$

The first term represents the optimization error after one projected gradient step. Since $\boldsymbol{\lambda}_{\rho,*}^t$ is a fixed point and $J_\rho(\boldsymbol{\lambda}; \bar{\boldsymbol{\Delta}}^t)$ is $\rho$-strongly convex, the update contracts the distance to the target $\boldsymbol{\lambda}_{\rho,*}^t$:

$$\mathbb{E}\left\|\boldsymbol{\lambda}_\rho^{t+1} - \boldsymbol{\lambda}_{\rho,*}^t\right\|^2$$

$$=\mathbb{E}\left\|\Pi_{\boldsymbol{\omega}}\left(\boldsymbol{\lambda}_\rho^t - \eta_\lambda\left[\mathbf{A}(\bar{\boldsymbol{\Delta}}^t)^\top\bar{\boldsymbol{\Delta}}^t\mathbf{A}^\top + \rho\mathbf{I}\right]\boldsymbol{\lambda}_\rho^t\right) - \Pi_{\boldsymbol{\omega}}\left(\boldsymbol{\lambda}_{\rho,*}^t - \eta_\lambda\left[\mathbf{A}(\bar{\boldsymbol{\Delta}}^t)^\top\bar{\boldsymbol{\Delta}}^t\mathbf{A}^\top + \rho\mathbf{I}\right]\boldsymbol{\lambda}_{\rho,*}^t\right)\right\|^2$$

$$\leq\mathbb{E}\left\|\boldsymbol{\lambda}_\rho^t - \boldsymbol{\lambda}_{\rho,*}^t - \eta_\lambda\left[\mathbf{A}(\bar{\boldsymbol{\Delta}}^t)^\top\bar{\boldsymbol{\Delta}}^t\mathbf{A}^\top + \rho\mathbf{I}\right](\boldsymbol{\lambda}_\rho^t - \boldsymbol{\lambda}_{\rho,*}^t)\right\|^2$$

$$=\mathbb{E}\left\|\boldsymbol{\lambda}_\rho^t - \boldsymbol{\lambda}_{\rho,*}^t\right\|^2 - 2\eta_\lambda\mathbb{E}\underbrace{\left\langle\boldsymbol{\lambda}_\rho^t - \boldsymbol{\lambda}_{\rho,*}^t, \left[\mathbf{A}(\bar{\boldsymbol{\Delta}}^t)^\top\bar{\boldsymbol{\Delta}}^t\mathbf{A}^\top + \rho\mathbf{I}\right](\boldsymbol{\lambda}_\rho^t - \boldsymbol{\lambda}_{\rho,*}^t)\right\rangle}_{=\langle\boldsymbol{\lambda}_\rho^t - \boldsymbol{\lambda}_{\rho,*}^t, \nabla J_\rho(\boldsymbol{\lambda}_\rho^t;\bar{\boldsymbol{\Delta}}^t) - \nabla J_\rho(\boldsymbol{\lambda}_{\rho,*}^t;\bar{\boldsymbol{\Delta}}^t)\rangle \overset{(12b)}{\geq} \rho\|\boldsymbol{\lambda}_\rho^t - \boldsymbol{\lambda}_{\rho,*}^t\|^2}$$

$$+ \eta_\lambda^2 \cdot \mathbb{E}\Big[\underbrace{\left\|\left[\mathbf{A}(\bar{\boldsymbol{\Delta}}^t)^\top\bar{\boldsymbol{\Delta}}^t\mathbf{A}^\top + \rho\mathbf{I}\right](\boldsymbol{\lambda}_\rho^t - \boldsymbol{\lambda}_{\rho,*}^t)\right\|^2}_{\leq\|\mathbf{A}(\bar{\boldsymbol{\Delta}}^t)^\top\bar{\boldsymbol{\Delta}}^t\mathbf{A}^\top + \rho\mathbf{I}\|_2^2\|\boldsymbol{\lambda}_\rho^t - \boldsymbol{\lambda}_{\rho,*}^t\|^2 \leq 2\left(\|\mathbf{A}\|_2^4\|\bar{\boldsymbol{\Delta}}^t\|_2^4 + \rho^2\right)\|\boldsymbol{\lambda}_\rho^t - \boldsymbol{\lambda}_{\rho,*}^t\|^2}\Big]$$

$$\leq\Big[1 - 2\eta_\lambda\rho + \underbrace{2\eta_\lambda^2\left(S^2 M^4\|\mathbf{A}\|_2^4 + \rho^2\right)}_{\leq\eta_\lambda\rho}\Big]\mathbb{E}\left\|\boldsymbol{\lambda}_\rho^t - \boldsymbol{\lambda}_{\rho,*}^t\right\|^2$$

$$\leq(1 - \eta_\lambda\rho)\mathbb{E}\left\|\boldsymbol{\lambda}_\rho^t - \boldsymbol{\lambda}_{\rho,*}^t\right\|^2.$$

The second term is bounded by the Lipschitz continuity of $\boldsymbol{\lambda}$ with respect to the input $\bar{\boldsymbol{\Delta}}$:

$$\mathbb{E}\left\|\boldsymbol{\lambda}_{\rho,*}^{t+1} - \boldsymbol{\lambda}_{\rho,*}^t\right\|^2 \leq \frac{\left(\mathbb{E}\|\bar{\boldsymbol{\Delta}}^{t+1}\|_2 + \mathbb{E}\|\bar{\boldsymbol{\Delta}}^t\|_2\right)^2\|\mathbf{A}\|_2^4}{\rho^2} \cdot \mathbb{E}\|\bar{\boldsymbol{\Delta}}^{t+1} - \bar{\boldsymbol{\Delta}}^t\|_2^2$$

$$\leq\frac{2\left(\mathbb{E}\|\bar{\boldsymbol{\Delta}}^{t+1}\|_2^2 + \mathbb{E}\|\bar{\boldsymbol{\Delta}}^t\|_2^2\right)\|\mathbf{A}\|_2^4}{\rho^2} \cdot 2\left(\mathbb{E}\|\bar{\boldsymbol{\Delta}}^{t+1}\|_2^2 + \mathbb{E}\|\bar{\boldsymbol{\Delta}}^t\|_2^2\right)$$

$$\leq\frac{16 S^2 M^4\|\mathbf{A}\|_2^4}{\rho^2}.$$

Combining these bounds yields the recursive relation for the tracking error:

$$\mathbb{E}\left\|\boldsymbol{\lambda}_\rho^{t+1} - \boldsymbol{\lambda}_{\rho,*}^{t+1}\right\|^2 \leq \left(1 - \frac{\eta_\lambda\rho}{2}\right)\mathbb{E}\left\|\boldsymbol{\lambda}_\rho^t - \boldsymbol{\lambda}_{\rho,*}^t\right\|^2 + \frac{16(\eta_\lambda\rho + 2)S^2 M^4\|\mathbf{A}\|_2^4}{\eta_\lambda\rho^3}. \tag{31}$$

Substituting all the bounds and (31) back into (30) yields

$$\mathbb{E}\|x^{t+1} - x^*\|^2 + \frac{8\eta_g\|\nabla\mathbf{F}(x^*)\mathbf{A}^\top\|_2^2}{\eta_\lambda\rho\mu\nu_A}\mathbb{E}\left\|\boldsymbol{\lambda}_\rho^{t+1} - \boldsymbol{\lambda}_{\rho,*}^{t+1}\right\|^2$$

$$\leq\left(1 - \frac{\eta_g\mu\nu_A}{2} + \frac{16\eta_g S^2 G^2\|\nabla\mathbf{F}(x^*)\mathbf{A}^\top\|_2^2\|\mathbf{A}\|_2^4}{\mu\nu_A\rho^2}\right)\mathbb{E}\|x^t - x^*\|^2 + \frac{8\eta_g\|\nabla\mathbf{F}(x^*)\mathbf{A}^\top\|_2^2}{\eta_\lambda\rho\mu\nu_A}\left(1 - \frac{\eta_\lambda^2\rho^2}{4}\right)\mathbb{E}\left\|\boldsymbol{\lambda}_\rho^t - \boldsymbol{\lambda}_{\rho,*}^t\right\|^2$$

$$+ \frac{\eta_g\|\mathbf{A}\|_2^2}{\mu\nu_A}\left[2 + \frac{8S\left(G^2 + M^2\right)\|\nabla\mathbf{F}(x^*)\mathbf{A}^\top\|_2^2\|\mathbf{A}\|_2^2}{\rho^2}\right]\sum_{s\in[S]}\mathbb{E}[\mathcal{E}_s^t] + \eta_g^2\mathbb{E}\left\|d^{t+1}\right\|^2$$

$$+ \frac{64\eta_g(\eta_\lambda\rho + 2)^2 S^2 M^4\|\nabla\mathbf{F}(x^*)\mathbf{A}^\top\|_2^2\|\mathbf{A}\|_2^4}{\mu\nu_A\eta_\lambda^2\rho^4} + \frac{4\eta_g\|\nabla\mathbf{F}(x^*)\mathbf{A}^\top\|_2^2}{\mu\nu_A}\left\|\boldsymbol{\lambda}_\rho^*(x^*) - \boldsymbol{\lambda}_0^*(x^*)\right\|^2.$$

Combining the above inequality and (28) yielding

$$\varphi^{t+1} + \frac{8\eta_g\|\nabla\mathbf{F}(x^*)\mathbf{A}^\top\|_2^2}{\eta_\lambda\rho\mu\nu_A}\mathbb{E}\left\|\boldsymbol{\lambda}_\rho^{t+1} - \boldsymbol{\lambda}_{\rho,*}^{t+1}\right\|^2$$

$$=\mathbb{E}\|x^{t+1} - x^*\|^2 + \gamma\sum_{s\in[S]}\mathbb{E}\left[\mathcal{E}_s^{t+1}\right] + (1+\gamma)\gamma^2 L^2\sum_{s\in[S]}\mathbb{E}\left[\mathcal{C}_s^{t+1}\right] + \frac{8\eta_g\|\nabla\mathbf{F}(x^*)\mathbf{A}^\top\|_2^2}{\eta_\lambda\rho\mu\nu_A}\mathbb{E}\left\|\boldsymbol{\lambda}_\rho^{t+1} - \boldsymbol{\lambda}_{\rho,*}^{t+1}\right\|^2$$

$$\leq\left[1 - \frac{\eta_g\mu\nu_A}{2} + \frac{16\eta_g S^2 G^2\|\nabla\mathbf{F}(x^*)\mathbf{A}^\top\|_2^2\|\mathbf{A}\|_2^4}{\mu\nu_A\rho^2}\right]\mathbb{E}\|x^t - x^*\|^2$$

$$+ \gamma\left[\frac{2-\gamma-\gamma^2}{2} + 12\gamma(1-\gamma^2)\eta_l^2 K^2 L^2 + \frac{2\eta_g\|\mathbf{A}\|_2^2}{\gamma\mu\nu_A} + \frac{8\eta_g S\left(G^2+M^2\right)\|\nabla\mathbf{F}(x^*)\mathbf{A}^\top\|_2^2\|\mathbf{A}\|_2^4}{\gamma\mu\nu_A\rho^2}\right]\sum_{s\in[S]}\mathbb{E}\left[\mathcal{E}_s^t\right]$$

$$+ 192(1+\gamma)\gamma^3\eta_l^2 K^2 L^4\sum_{s\in[S]}\mathbb{E}\left[\mathcal{C}_s^t\right] + \frac{8\eta_g\|\nabla\mathbf{F}(x^*)\mathbf{A}^\top\|_2^2}{\eta_\lambda\rho\mu\nu_A}\left(1 - \frac{\eta_\lambda^2\rho^2}{4}\right)\mathbb{E}\left\|\boldsymbol{\lambda}_\rho^t - \boldsymbol{\lambda}_{\rho,*}^t\right\|^2$$

$$+ \eta_g^2\left(1 + 2SL^2\right)\underbrace{\mathbb{E}\|d^{t+1}\|^2}_{\leq 2\|\mathbf{A}\|_2^2\sum_{s\in[S]}\mathbb{E}[\mathcal{E}_s^t] + 2SG^2\|\mathbf{A}\|_2^2} + \frac{4\eta_g\|\nabla\mathbf{F}(x^*)\mathbf{A}^\top\|_2^2}{\mu\nu_A}\left\|\boldsymbol{\lambda}_\rho^*(x^*) - \boldsymbol{\lambda}_0^*(x^*)\right\|^2$$

$$+ 96(1+\gamma)\gamma^2\eta_l^2 SK^2 L^2 G^2 + 12(1+\gamma)\gamma^3\eta_l^2 SK^2 L^2\sigma^2\left(1 + \frac{16}{K}\right) + \sum_{s\in[S]}\frac{2\gamma^3\sigma^2}{K|\mathcal{R}_s|}$$

$$+ \frac{64\eta_g(\eta_\lambda\rho+2)^2 S^2 M^4\|\nabla\mathbf{F}(x^*)\mathbf{A}^\top\|_2^2\|\mathbf{A}\|_2^4}{\mu\nu_A\eta_\lambda^2\rho^4}$$

$$\leq\underbrace{\left[1 - \frac{\eta_g\mu\nu_A}{2} + \frac{16\eta_g S^2 G^2\|\nabla\mathbf{F}(x^*)\mathbf{A}^\top\|_2^2\|\mathbf{A}\|_2^4}{\mu\nu_A\rho^2}\right]}_{\leq 1 - \frac{\eta_g\mu\nu_A}{4}}\mathbb{E}\|x^t - x^*\|^2$$

$$+ \gamma\underbrace{\left[1 - \frac{\gamma+\gamma^2}{2} + 12\gamma(1-\gamma^2)\eta_l^2 K^2 L^2 + \frac{2\eta_g\|\mathbf{A}\|_2^2}{\gamma\mu\nu_A}\left[1 + \frac{4S\left(G^2+M^2\right)\|\nabla\mathbf{F}(x^*)\mathbf{A}^\top\|_2^2\|\mathbf{A}\|_2^2}{\rho^2} + \eta_g(1+2SL^2)\mu\nu_A\right]\right]}_{\leq 1 - \frac{\eta_g\mu\nu_A}{4}}$$

$$\cdot\sum_{s\in[S]}\mathbb{E}\left[\mathcal{E}_s^t\right] + \underbrace{192\gamma\eta_l^2 K^2 L^2}_{\leq 1 - \frac{\eta_g\mu\nu_A}{4}}\cdot(1+\gamma)\gamma^2 L^2\sum_{s\in[S]}\mathbb{E}\left[\mathcal{C}_s^t\right] + \frac{8\eta_g\|\nabla\mathbf{F}(x^*)\mathbf{A}^\top\|_2^2}{\eta_\lambda\rho\mu\nu_A}\underbrace{\left(1 - \frac{\eta_\lambda^2\rho^2}{4}\right)}_{\leq 1 - \frac{\eta_g\mu\nu_A}{4}}\mathbb{E}\left\|\boldsymbol{\lambda}_\rho^t - \boldsymbol{\lambda}_{\rho,*}^t\right\|^2$$

$$+ \frac{4\eta_g\|\nabla\mathbf{F}(x^*)\mathbf{A}^\top\|_2^2}{\mu\nu_A}\left\|\boldsymbol{\lambda}_\rho^*(x^*) - \boldsymbol{\lambda}_0^*(x^*)\right\|^2 + 2\eta_g^2(1+2SL^2)SG^2\|\mathbf{A}\|_2^2 + 96(1+\gamma)\gamma^2\eta_l^2 SK^2 L^2 G^2$$

$$+ 12(1+\gamma)\gamma^3\eta_l^2 SK^2 L^2\sigma^2\left(1 + \frac{16}{K}\right) + \sum_{s\in[S]}\frac{2\gamma^3\sigma^2}{K|\mathcal{R}_s|} + \frac{64\eta_g(\eta_\lambda\rho+2)^2 S^2 M^4\|\nabla\mathbf{F}(x^*)\mathbf{A}^\top\|_2^2\|\mathbf{A}\|_2^4}{\mu\nu_A\eta_\lambda^2\rho^4}$$

$$\leq\left(1 - \frac{\eta_g\mu\nu_A}{4}\right)\left[\varphi^t + \frac{8\eta_g\|\nabla\mathbf{F}(x^*)\mathbf{A}^\top\|_2^2}{\eta_\lambda\rho\mu\nu_A}\mathbb{E}\left\|\boldsymbol{\lambda}_\rho^t - \boldsymbol{\lambda}_{\rho,*}^t\right\|^2\right] + \frac{4\eta_g\|\nabla\mathbf{F}(x^*)\mathbf{A}^\top\|_2^2}{\mu\nu_A}\left\|\boldsymbol{\lambda}_\rho^*(x^*) - \boldsymbol{\lambda}_0^*(x^*)\right\|^2$$

$$+ 2\eta_g^2(1+2SL^2)SG^2\|\mathbf{A}\|_2^2 + 96(1+\gamma)\gamma^2\eta_l^2 SK^2 L^2 G^2 + 12(1+\gamma)\gamma^3\eta_l^2 SK^2 L^2\sigma^2\left(1 + \frac{16}{K}\right)$$

$$+ \sum_{s\in[S]}\frac{2\gamma^3\sigma^2}{K|\mathcal{R}_s|} + \frac{64\eta_g(\eta_\lambda\rho+2)^2 S^2 M^4\|\nabla\mathbf{F}(x^*)\mathbf{A}^\top\|_2^2\|\mathbf{A}\|_2^4}{\mu\nu_A\eta_\lambda^2\rho^4}$$

$$=\left(1 - \frac{\eta_g\mu\nu_A}{4}\right)\left[\varphi^t + \frac{8\eta_g\|\nabla\mathbf{F}(x^*)\mathbf{A}^\top\|_2^2}{\eta_\lambda\rho\mu\nu_A}\mathbb{E}\left\|\boldsymbol{\lambda}_\rho^t - \boldsymbol{\lambda}_{\rho,*}^t\right\|^2\right] + \mathcal{O}\left(\eta_g\rho^2 + \eta_g^2 + \gamma^2\eta_l^2 + \gamma^3 + \frac{\eta_g}{\eta_\lambda^2\rho^4}\right).$$

The parameter conditions are summarized as follows:

$$8SG\|\nabla\mathbf{F}(x^*)\mathbf{A}^\top\|_2\|\mathbf{A}\|_2^2 \leq \rho\mu\nu_A,$$

$$\frac{\eta_g \mu \nu_A}{\gamma} + 48(1-\gamma^2)\eta_l^2 K^2 L^2 + \frac{8\eta_g \|\mathbf{A}\|_2^2}{\gamma^2 \mu \nu_A} \left[ 1 + \frac{4S\left(G^2 + M^2\right)\|\nabla \mathbf{F}(x^*)\mathbf{A}^\top\|_2^2 \|\mathbf{A}\|_2^2}{\rho^2} + \eta_g(1+2SL^2)\mu \nu_A \right] \le 2(1+\gamma),$$

$$\eta_g \mu \nu_A \le \min\{4 - 768\gamma \eta_l^2 K^2 L^2, \eta_\lambda^2 \rho^2\},$$

$$2\eta_\lambda \left[ S^2 M^4 \|\mathbf{A}\|_2^4 + \rho^2 \right] \le \rho.$$

This completes the proof of Theorem 4.13. $\qquad\qquad\square$

By setting the server learning rate as $\eta_g = \Theta(\log(T)/T)$ to ensure linear decay of the exponential term, the constraint $\eta_g \le \mathcal{O}(\gamma^2)$ necessitates that the momentum parameter $\gamma$ scales as $\Omega(\sqrt{\eta_g})$. By setting $\gamma = \Theta(\sqrt{\log(T)/T})$ and $\eta_l = \Theta(1/\sqrt{T})$, the weighted sub-optimality gap $\mathbb{E}[\mathbf{F}(x^r) - \mathbf{F}(x^*)]\mathbf{A}^\top \boldsymbol{\lambda}_\rho^{r+1}$ converges at a rate of $\tilde{\mathcal{O}}(1/\sqrt{T})$. Simultaneously, the Lyapunov function $\varphi^{t+1}$ exhibits a distinct convergence behavior characterized by geometric contraction. The general recursive bound implies that the system converges linearly to a steady-state error neighborhood, whose magnitude is determined by the ratio of the noise terms to the contraction coefficient $\Theta(\eta_g)$. Specific weight update strategies further dictate the structure of this error floor. In the Exact-Update scenario, the stability condition imposes a hard lower bound on the regularization parameter, requiring $\rho = \Theta(1)$. Consequently, the regularization bias becomes irreducible, causing the algorithm to linearly converge to an error floor dominated by $\mathcal{O}(\rho^2)$. The Step-Update scenario faces a more severe precision trade-off, where the weight update step size is tightly constrained by $\eta_\lambda \le \mathcal{O}(\rho)$ to ensure tracking stability. This limitation amplifies the tracking error term $\mathcal{O}(1/(\eta_\lambda^2 \rho^4))$, resulting in a significantly larger steady-state error floor (when $\rho < 1$) of order $\mathcal{O}(\rho^{-6})$ compared to the exact counterpart.

# H. Additional Experiments

In this section, we provide detailed descriptions of the experimental environment, datasets, and hyperparameter settings used in evaluations. We then present additional experimental results, including performance evaluations on large-scale benchmarks, to further substantiate the effectiveness of DREAM.

## H.1. Detailed Experimental Setup

All experiments are conducted on a computing platform equipped with an Intel Xeon Gold 6242R CPU, 512GB of RAM, and three NVIDIA Tesla P100 GPUs. The software stack is built upon Ubuntu 20.04 LTS, with Python 3.8 and PyTorch 1.13 forming the core of our implementation.

**Datasets and Tasks**. We evaluate the framework on three multi-task image classification benchmarks:

- **Multi-MNIST**: This dataset is constructed by overlaying two MNIST digits into a single $28 \times 28$ image. The goal is to simultaneously classify the left digit (Task 1) and the right digit (Task 2). It contains 60,000 training samples and 10,000 test samples. This serves as a standard benchmark for analyzing gradient conflicts in MOO (Yang et al., 2023; Sabour et al., 2017). Sample images are shown in Figure 4.

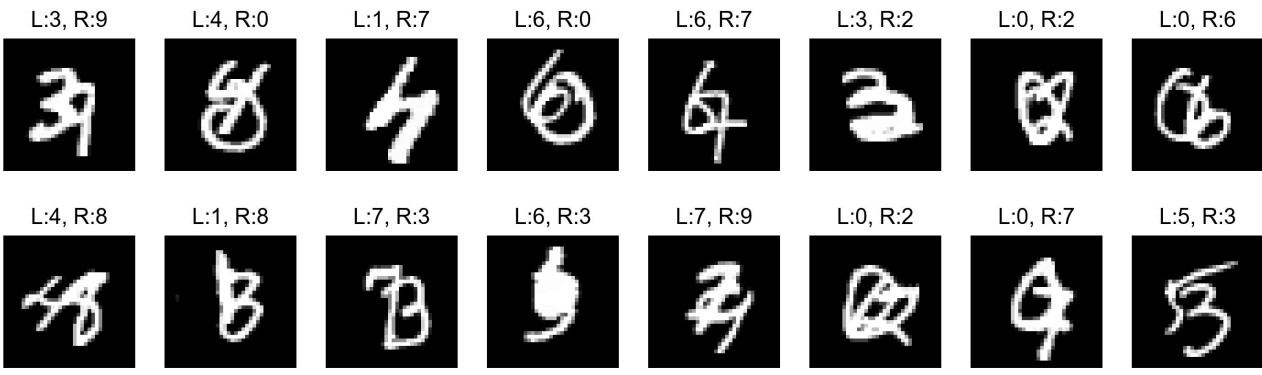

*Figure 4.* Samples from Multi-MNIST Dataset: Each image contains two overlapping digits.

- **2MNIST+FMNIST**: To introduce task-type heterogeneity, we create a mixed-domain dataset combining MNIST and Fashion-MNIST. Each sample consists of three stacked images. Task 1 and Task 2 are digit classification tasks (MNIST), while Task 3 is a clothing classification task (Fashion-MNIST). This setup challenges the algorithm to balance tasks with different feature distributions and difficulties. Sample images are shown in Figure 5.

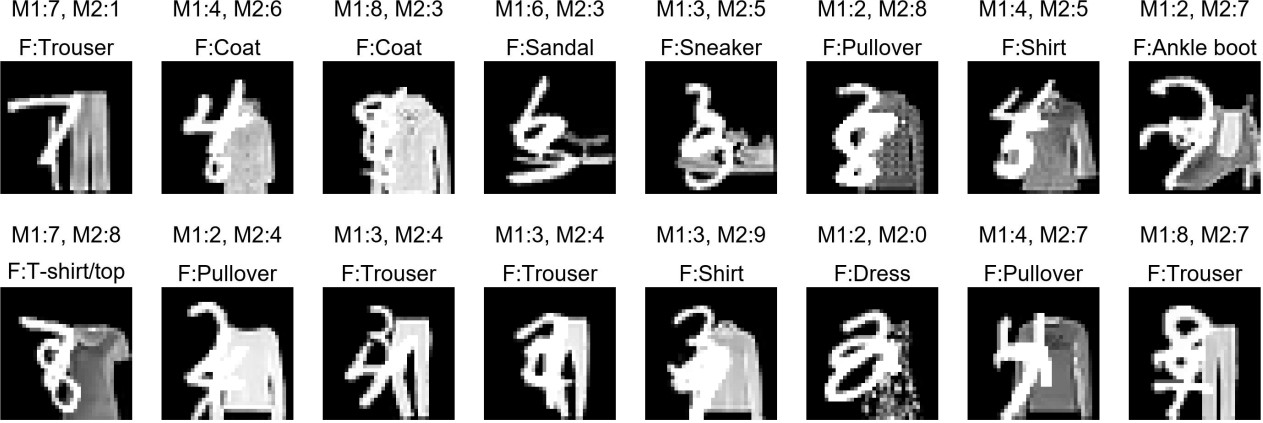

*Figure 5.* Samples from 2MNIST+FMNIST Dataset: Each sample is a composite of two MNIST digits and one Fashion-MNIST item.

- **CelebA** (Liu et al., 2015): A large-scale face attributes dataset containing over 200,000 celebrity images, each annotated with 40 binary attributes (e.g., Smiling, Eyeglasses, Wavy Hair). We treat the prediction of each attribute as a separate binary classification task, resulting in a 40-objective optimization problem. This represents a challenging real-world scenario with a massive number of objectives. Sample images are shown in Figure 6.

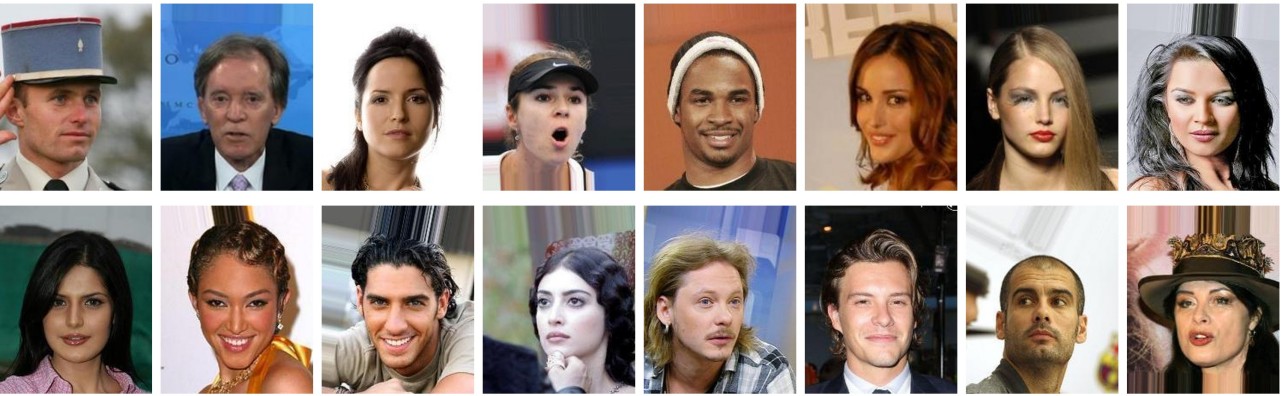

*Figure 6.* Samples from CelebA Dataset: Representative images with diverse facial attributes.

**Model Architectures**. We adopt the Hard Parameter Sharing paradigm across all tasks, employing shared encoders to extract common features followed by task-specific heads. For the digit-based benchmarks (Multi-MNIST and 2MNIST+FMNIST), we utilize lightweight LeNet-style CNNs (LeCun et al., 1998) with a shared encoder of two convolutional layers and a fully connected layer, branching into 2 or 3 independent heads, respectively. For the more complex CIFAR-100 dataset, we implement a deeper VGG-style network. Crucially, this architecture replaces standard Batch Normalization with Group Normalization to mitigate statistical instability. Finally, for the 40-task CelebA benchmark, we employ a compact 3-layer CNN encoder to maintain computational efficiency suitable for edge devices, feeding into 40 independent binary classification heads.

**Hyperparameters.** Unless otherwise stated, we use the following default hyperparameters: momentum factor $\gamma = 0.5$, regularization parameter $\rho = 0.1$, client learning rate $\eta_l = 0.05$, local update steps $K = 20$, batch size $B = 16$. For the Step-Update strategy, the weight learning rate is set to $\eta_\lambda = 0.05$. The task correction matrix $\mathbf{A}$ is set to Identity $\mathbf{I}$ by default.

A federated environment is simulated with $N = 10$ clients, where data heterogeneity is introduced by partitioning samples using a Dirichlet distribution with concentration parameter $\alpha = 0.2$, creating a highly non-i.i.d. setting. Results are reported as the average over 5 independent runs with different random seeds.

## H.2. Performance on Multi-MNIST Dataset

To evaluate the impact of the momentum smoothing mechanism, we conduct an experiment on the momentum factor $\gamma$ using the Multi-MNIST dataset.

Figure 7 illustrates the test accuracy and loss trajectories for both Exact-Update and Step-Update strategies as with $\gamma \in \{0.3, 0.5, 0.8\}$. The results reveal that an appropriately large momentum factor is crucial for fast convergence. Specifically, the setting with $\gamma = 0.3$ exhibits a noticeably slower learning curve compared to larger values. Increasing $\gamma$ to $0.5$ brings a speedup, while further increasing it to $0.8$ yields diminishing returns, with the convergence trajectory closely overlapping that of $\gamma = 0.5$. This suggests that once the momentum is sufficient to stabilize the update direction, further increasing it provides limited additional acceleration. Furthermore, we observe that the Exact-Update and Step-Update strategies deliver virtually identical performance across all $\gamma$ settings. This consistency supports the practical utility of the Step-Update strategy, confirming that it can serve as a proxy for the exact solution while reducing computational overhead.

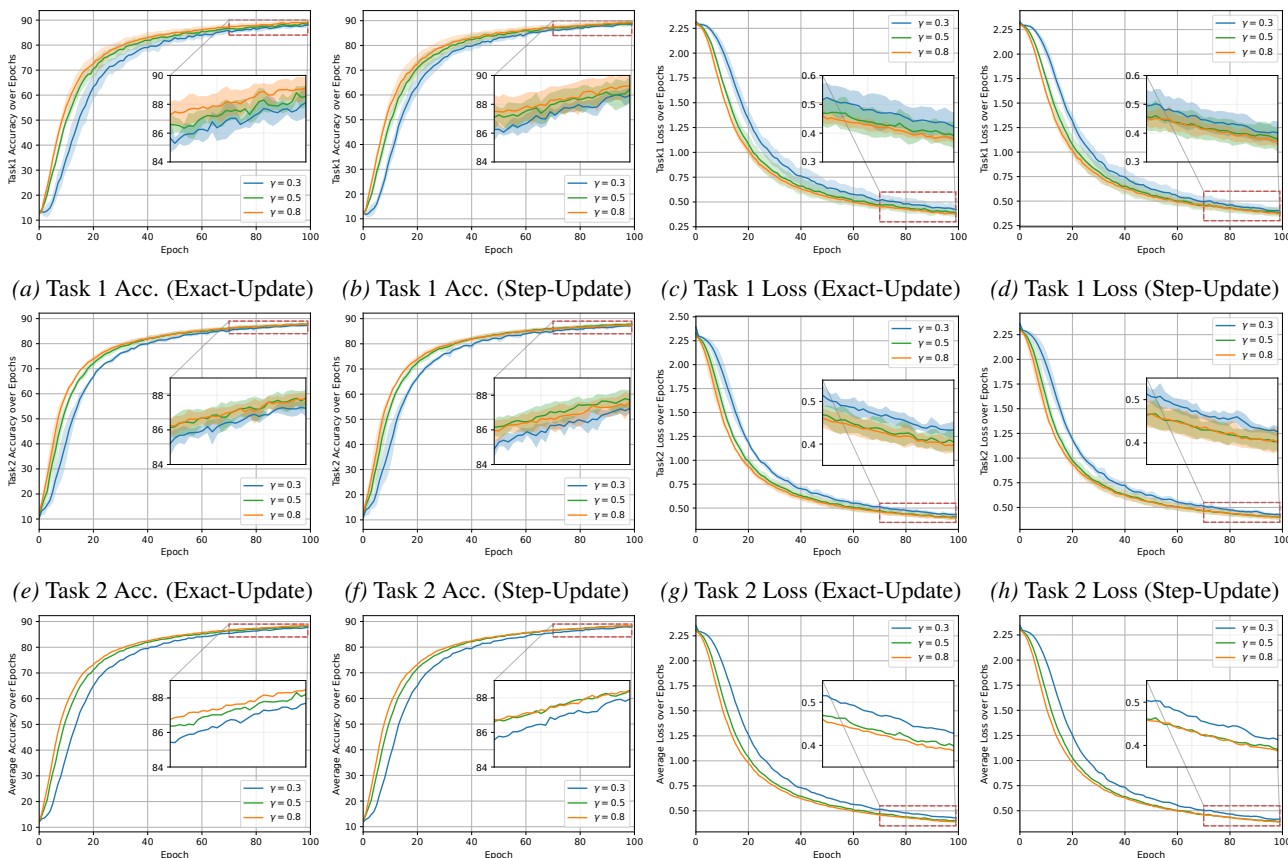

*(a)* Task 1 Acc. (Exact-Update)  *(b)* Task 1 Acc. (Step-Update)  *(c)* Task 1 Loss (Exact-Update)  *(d)* Task 1 Loss (Step-Update)

*(e)* Task 2 Acc. (Exact-Update)  *(f)* Task 2 Acc. (Step-Update)  *(g)* Task 2 Loss (Exact-Update)  *(h)* Task 2 Loss (Step-Update)

*(i)* Average Acc. (Exact-Update)  *(j)* Average Acc. (Step-Update)  *(k)* Average Loss (Exact-Update)  *(l)* Average Loss (Step-Update)

*Figure 7.* Impact of momentum parameter $\gamma$ on the Multi-MNIST Dataset.

## H.3. Performance on 2MNIST+FMNIST Dataset

We evaluate DREAM against several FL baselines: FSMGDA (Yang et al., 2023), and the traditional single-objective algorithms SCAFFOLD and FedAvg, which use fixed weights of $[0.1, 0.3, 0.6]$ for scalarization. We set communication rounds $T = 300$ and local updates $K = 50$ for all methods. This setup introduces heterogeneity not only in data distribution across $N = 50$ clients but also in the task types themselves.

As shown in Figure 8, DREAM (Exact-Update) demonstrates the best overall performance, achieving the highest average accuracy (approx. 70%). More importantly, it maintains a fair and balanced performance across three tasks. In contrast, the baseline methods struggle with the compounded challenges of data and task heterogeneity. FSMGDA exhibits a clear preference for Task 1. SCAFFOLD and FedAvg are heavily skewed by their pre-defined fixed weights, leading to an over-emphasis on Task 3. In terms of average performance, SCAFFOLD and FSMGDA perform comparably, both notably outperforming FedAvg, but falling short of DREAM. These findings yield a clear conclusion: both data heterogeneity and improper fixed-weighting can severely degrade multi-objective training. DREAM successfully overcomes both of these challenges, demonstrating its robustness and effectiveness in complex, real-world FMOL scenarios.

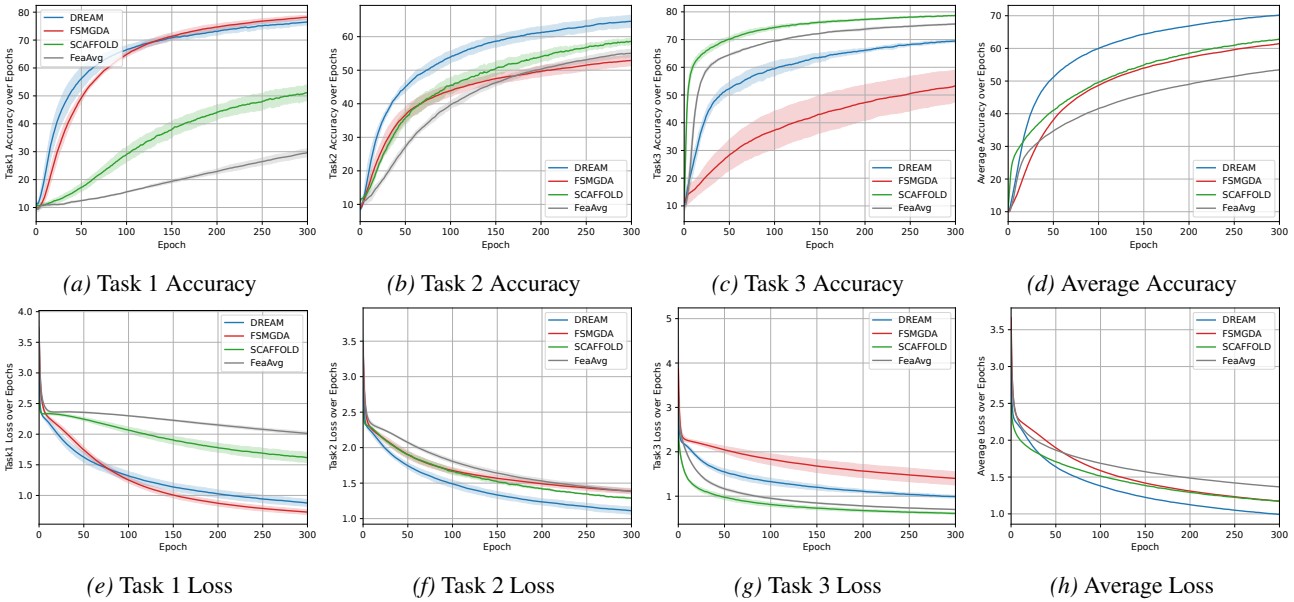

*(a)* Task 1 Accuracy      *(b)* Task 2 Accuracy      *(c)* Task 3 Accuracy      *(d)* Average Accuracy

*(e)* Task 1 Loss      *(f)* Task 2 Loss      *(g)* Task 3 Loss      *(h)* Average Loss

*Figure 8.* Performance comparison on the 2MNIST+FMNIST dataset under a non-i.i.d. setting.

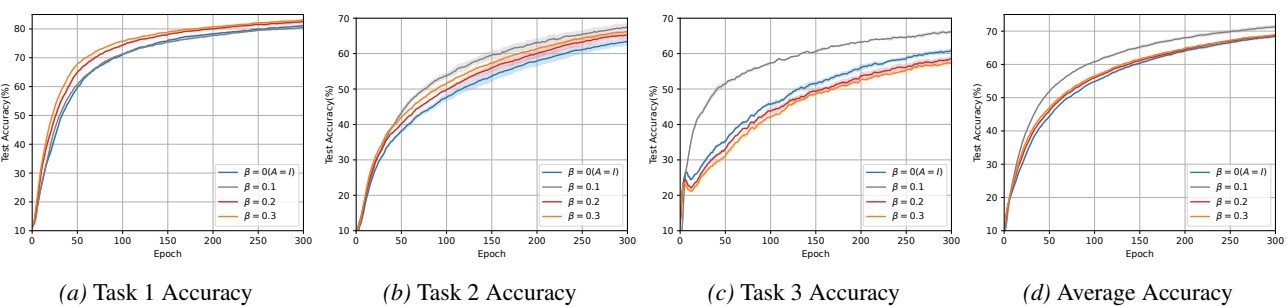

*(a)* Task 1 Accuracy      *(b)* Task 2 Accuracy      *(c)* Task 3 Accuracy      *(d)* Average Accuracy

*Figure 9.* Effect of modeling task interactions via off-diagonal entries in $\mathbf{A}$ on the 2MNIST+FMNIST dataset.

Beyond baseline comparisons, we use the same 2MNIST+FMNIST benchmark to evaluate whether the task correction matrix $\mathbf{A}$ can encode task-interaction priors. Since the first two tasks are MNIST-related and are naturally more correlated with each other than with the FMNIST task, we introduce an off-diagonal coupling between them:

$$\mathbf{A} = \begin{bmatrix} 1 & \beta & 0 \\ \beta & 1 & 0 \\ 0 & 0 & 1 \end{bmatrix},$$

where $\beta$ controls the interaction strength between the two MNIST tasks while leaving the FMNIST task independent. Figure 9 shows that a mild positive coupling improves the optimization outcome compared with the standard MGDA setting $\mathbf{A} = \mathbf{I}$. In particular, $\beta = 0.1$ yields a $+2.95\%$ gain in average accuracy over the identity baseline, while stronger couplings ($\beta = 0.2, 0.3$) still provide consistent gains of around $+0.4\%$. This suggests that off-diagonal entries in $\mathbf{A}$ can translate

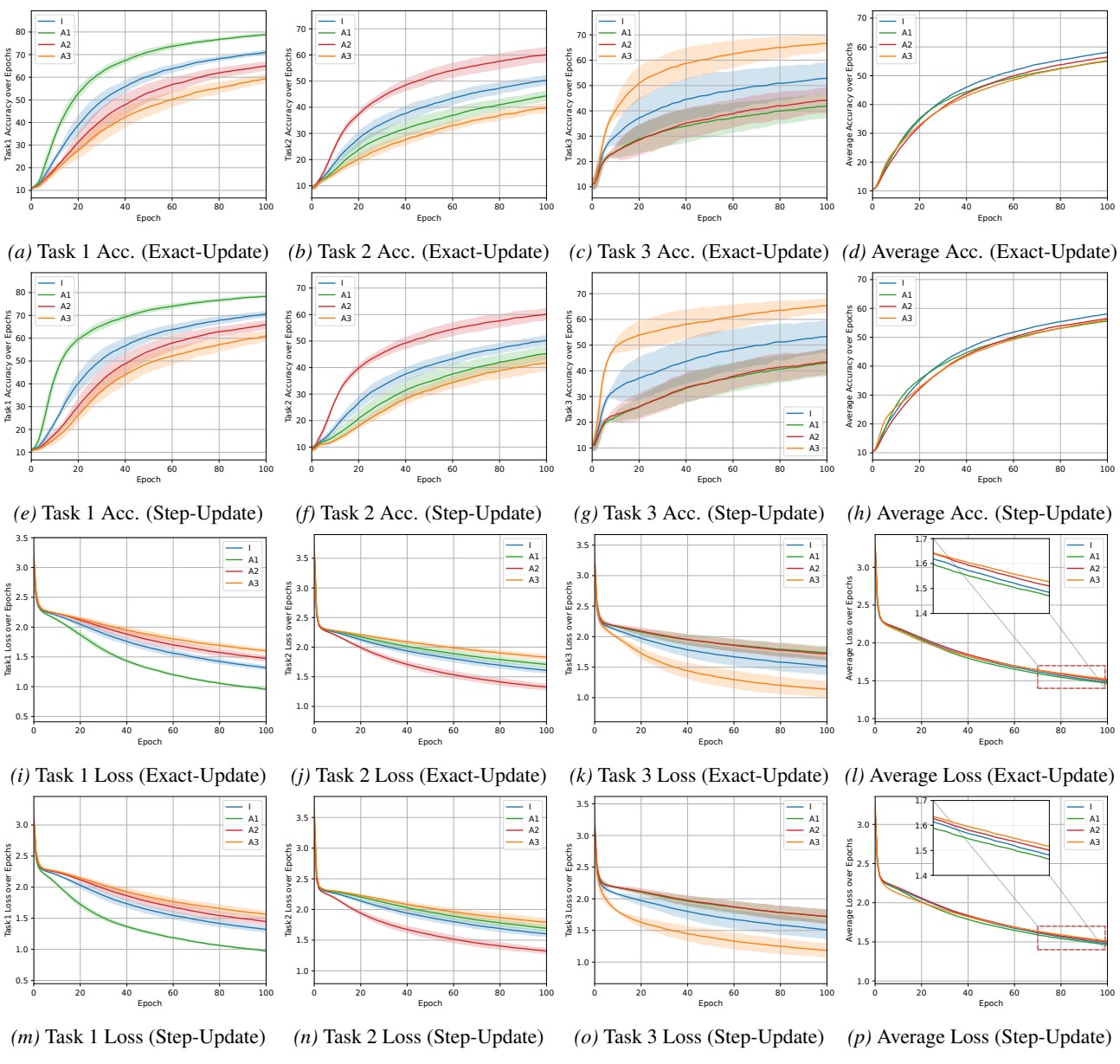

*(a)* Task 1 Acc. (Exact-Update)    *(b)* Task 2 Acc. (Exact-Update)    *(c)* Task 3 Acc. (Exact-Update)    *(d)* Average Acc. (Exact-Update)

*(e)* Task 1 Acc. (Step-Update)    *(f)* Task 2 Acc. (Step-Update)    *(g)* Task 3 Acc. (Step-Update)    *(h)* Average Acc. (Step-Update)

*(i)* Task 1 Loss (Exact-Update)    *(j)* Task 2 Loss (Exact-Update)    *(k)* Task 3 Loss (Exact-Update)    *(l)* Average Loss (Exact-Update)

*(m)* Task 1 Loss (Step-Update)    *(n)* Task 2 Loss (Step-Update)    *(o)* Task 3 Loss (Step-Update)    *(p)* Average Loss (Step-Update)

*Figure 10.* Task Prioritization on the 2MNIST+FMNIST dataset.

task-structure priors into empirical gains: moderate coupling allows related tasks to share directional information, whereas overly strong coupling may partially blur task-specific signals.

We further validate the flexibility of $\mathbf{A}$ by investigating its response to explicit task prioritization. We configure three diagonal task correction matrices to target specific objectives:

$$\mathbf{A}_1 = \begin{bmatrix} 1.5 & 0 & 0 \\ 0 & 0.75 & 0 \\ 0 & 0 & 0.75 \end{bmatrix}, \quad \mathbf{A}_2 = \begin{bmatrix} 0.75 & 0 & 0 \\ 0 & 1.5 & 0 \\ 0 & 0 & 0.75 \end{bmatrix}, \quad \mathbf{A}_3 = \begin{bmatrix} 0.75 & 0 & 0 \\ 0 & 0.75 & 0 \\ 0 & 0 & 1.5 \end{bmatrix},$$

which prioritize Task 1, Task 2, and Task 3, respectively, and compare them against the standard identity matrix $\mathbf{I}$. Figure 10 visualizes the resulting trajectories for accuracy and loss, respectively. The empirical results demonstrate a precise alignment with the specified preferences: setting $\mathbf{A} = \mathbf{A}_1$ yields the fastest convergence and highest final accuracy for Task 1, while $\mathbf{A} = \mathbf{A}_2$ and $\mathbf{A} = \mathbf{A}_3$ successfully shift the performance advantage to Task 2 and Task 3, respectively. Importantly, this

prioritization reflects a controlled trade-off, confirming that DREAM provides a robust mechanism to incorporate user preferences into the optimization process.

## H.4. Performance on CelebA Dataset

To demonstrate the scalability and robustness of DREAM in large-scale scenarios, we evaluate it on the CelebA dataset, which involves simultaneous optimization of 40 distinct classification tasks. Figure 11 visualizes the final test accuracy for each of the 40 tasks using radar charts, providing a holistic view of the performance landscape. Quantitative results confirm the superiority of our framework. DREAM (Exact-Update) achieves the highest average accuracy of 76.26%, followed closely by the Step-Update variant at 75.67%. In comparison, the baseline FSMGDA struggles significantly in this complex setting, reaching an average accuracy of only 69.06%. FedCMOO performs better with an average accuracy of 75.79%. This highlights DREAM's robustness and its superior capability in finding a high-quality, well-balanced Pareto solution in large-scale FMOL settings.

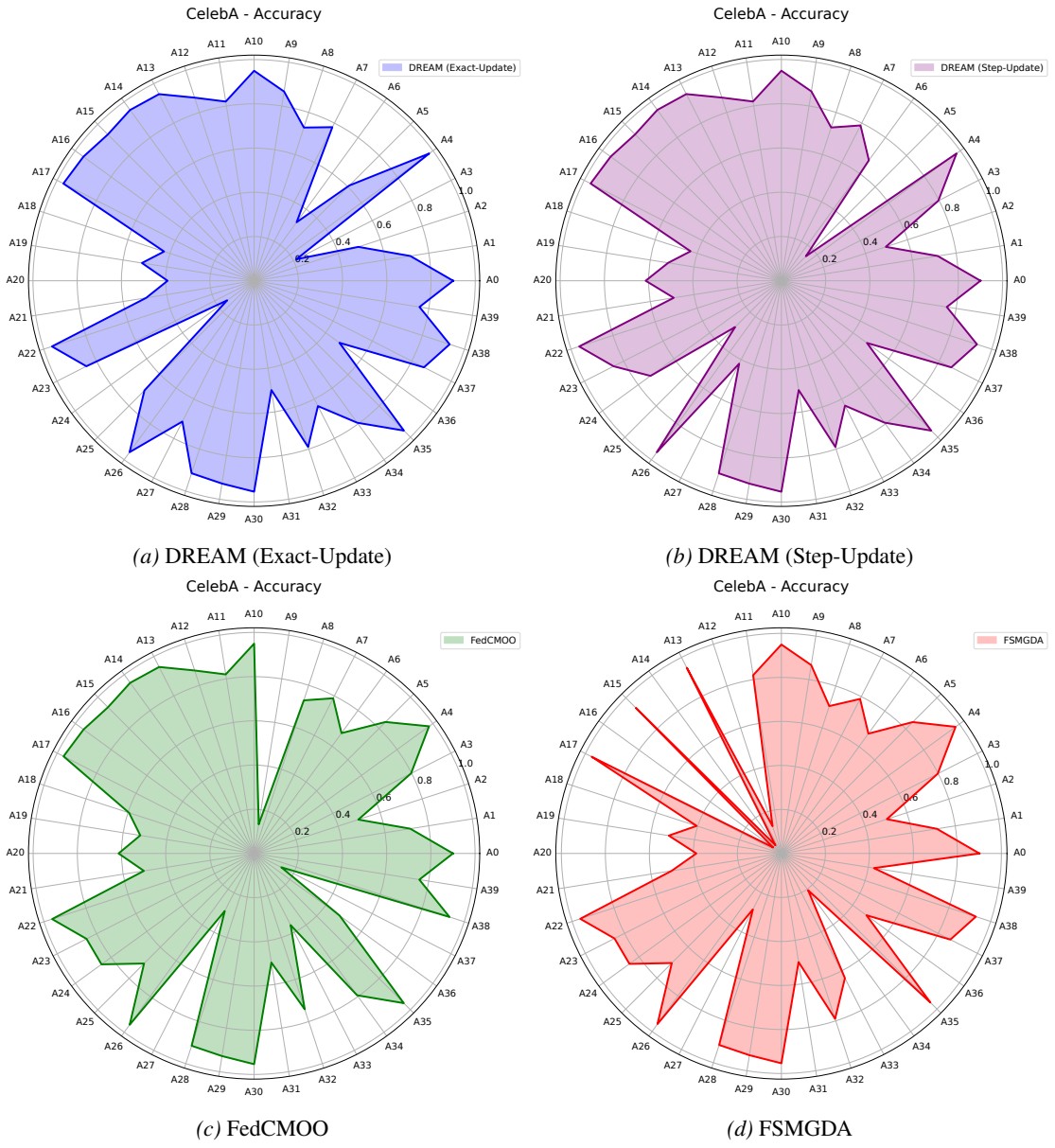

*(a)* DREAM (Exact-Update)  *(b)* DREAM (Step-Update)

*(c)* FedCMOO  *(d)* FSMGDA

*Figure 11.* Test accuracy on 40 tasks of the CelebA Dataset.

