# OpenReview forum: "DREAM: A Unified Framework for Drift-Corrected Federated Multi-Objective Learning"
_ICML.cc/2026/Conference — ICML 2026 regular_

### Official Review · Reviewer_ddqE · 2026-03-01

**Soundness:** 3
**Presentation:** 3
**Significance:** 2
**Originality:** 3
**Overall Recommendation:** 4
**Confidence:** 3

**Summary:**

The paper proposed DREAM, a federated multi-objective learning framework in which clients use a correction term with momentum to mitigate local training drift and stochastic noise, while the server determines task weights via a regularized multi-objective QP/MGDA and introduces a task correction matrix to encode task preferences for weighted aggregation.

**Compliance With Llm Reviewing Policy:**

Affirmed.

**Final Justification:**

Thanks for the authors' response. Overall, the author's response addressed some of my concerns, but the theory still relies on BG, the overhead discussion is mostly qualitative, and the task-correction matrix $A$ is now positioned as optional and typically set to $I$, which undermines its role as a meaningful contribution. Based on the clarification provided by the current response, I will only raise my overall recommendation by 1.

**Key Questions For Authors:**

1. Compared to the simple combination of drift correction, momentum, and regularized MGDA/QP, what is the core novelty of DREAM?

2. Which one is the most critical component in your method? If only two or one of them are retained, what would theoretically and experimentally happen? Could you please provide a clear conclusion?

**Limitations:**

Same as Weaknesses.

**Strengths And Weaknesses:**

### Strengths:

1. The paper separates the two key difficulties in FMOL for discussion, writing clearly and with coherent motivation.

2. The client and server are designed separately, making it easy to replace different multi-objective aggregators or client update rules.

3. The paper provides convergence analysis under non-convex and strongly convex settings.

### Weaknesses:

1. The novelty of the paper is limited. The core method is to combine the existing methods together. It feels more like integrating mature components rather than proposing a new algorithm.

2. Convergence analysis relies on a set of strong assumptions, such as bounded gradients.

3. The experiments are only focused on a small number of classic multi-task datasets, and lack ablation experiments to verify the effectiveness of each module.

---

> ### Author Rebuttal · Authors · 2026-03-30
>
> We sincerely thank the reviewer for the rigorous evaluation and address each concern below.
>
> **Q: Core novelty beyond combining existing components**
>
> We respectfully clarify that our contribution is not a heuristic combination of known ingredients, but an **analysis-driven framework** that identifies and resolves a recursive error coupling unique to FMOL, dictating why naïve combinations provably fail and how the mechanisms must jointly interact.
>
> The central obstacle is a **recursive error coupling unique to FMOL**: client drift corrupts task-gradient estimates → biased estimates enter the non-linear MOO solver → the solver's non-linearity amplifies the error into a wrong descent direction → the wrong global update worsens client drift next round. This vicious cycle is absent in single-objective FL (linear aggregation preserves unbiasedness) and centralized MOO (no local drift). Prior FMOL works bypass it via stronger assumptions to cap the error externally.
>
> Therefore, **the most critical part of DREAM is the joint client-side mechanism** to break this loop. Each mechanism alone is provably insufficient: drift correction without momentum retains non-vanishing $O(\sigma^2)$ in gradient estimation error $\mathcal{E}_s^t$, which propagates through aggregation and prevents the CA direction distance from converging; momentum without drift correction cannot remove systematic shifts, requiring restrictive conditions to bound. The joint design is not additively helpful but multiplicatively necessary.
>
> **Ablations** on Multi-MNIST (details are in our response to **R-57ab Q1**: https://imgur.com/a/K37psde):
>
> - **No drift correction** (→FSMGDA): avg accuracy drops ~**6%**, inter-task gap widens 1.3%→8.7%.
> - **No momentum** ($\gamma=1$): avg accuracy drops ~**2%**; direction norm and task weights oscillate strongly. Theoretically, $\gamma=1$ makes the variance in Thm 4.8 non-vanishing as $T\to\infty$.
>
> The server-side components serve structurally different roles: $\rho>0$ ensures Lipschitz continuity of the weight mapping, a mathematical prerequisite for proofs, not a heuristic; the matrix $A$ generalizes beyond standard MGDA to enable interaction modeling (empirical evidence in R-57ab Q2).
>
> **Specific technical contributions beyond individual components:**
>
> **(i) Coupled-error recurrence.** We formally derive the $\mathcal{E}_s^t \leftrightarrow \mathcal{C}_s^t$ feedback loop and prove both errors are jointly controllable through the proposed correction mechanism. Prior FMOL analyses sever this coupling by invoking bounded heterogeneity (BH).
>
> **(ii) Cross-coupled Lyapunov construction.** The SC analysis constructs $\phi^t$ that simultaneously couples optimality gap, gradient estimation error, client drift, and weight tracking error (for Step-Update) with carefully calibrated coefficients, going well beyond standard Lyapunov designs in FL.
>
> **(iii) Linear speedup in FMOL.** We establish $O(1/\sqrt{NT})$ convergence without bounded heterogeneity (Exact-Update). To our knowledge, no prior FMOL work achieves this under comparable assumptions.
>
> **(iv) CA direction distance guarantee.** We provide convergence bounds for the CA direction distance in FMOL, via a decomposition separating weight-mapping bias from input estimation error, achieving $O(1/\sqrt{NT})$ (Exact) and $O(T^{-2/5})$ (Step). No prior FMOL work offers such geometric guarantees.
>
> **(v) Exact vs. Step-Update cost characterization.** We explicitly quantify the theoretical price of replacing the exact QP solver with a single projected gradient step: the steady-state error inflates from $O(\rho^2)$ to $O(\rho^2+\rho^{-6})$, with the bottleneck ($\eta_\lambda\lesssim\rho$) precisely characterized.
>
> **(vi) Unified task-correction matrix.** The PSD matrix $A$ generalizes server-side aggregation, establishing a continuous interpolation between FedAvg ($A=J$) and MGDA ($A=I$). All convergence results hold for general $A$.
>
> In summary, DREAM constitutes a genuinely new analytical framework. To our knowledge, prior FMOL works do not simultaneously provide these.
>
> **On assumption strength**
>
> We appreciate this concern and want to clarify: our assumptions are actually weaker than existing FMOL baselines, not stronger. A detailed comparison table is at: https://imgur.com/a/IXS7WTI.
>
> - Bounded Gradient (BG), while seemingly strong in single-objective optimization, is standard across stochastic MOO (Chen et al., 2025). For SC objectives, it can be derived from iterate boundedness (Remark 4.6); we adopt it directly to streamline derivations.
> - DREAM (Exact): only Smoothness + BG — the minimal set among all stochastic MOO methods. FedCMOO adds BH; FSMGDA requires $(\alpha,\beta)$-LCSG implying mega-batch sampling.
> - DREAM (Step): adds BH solely for weight tracking, still avoiding FSMGDA's stronger conditions, despite FSMGDA requiring exact MGDA solves.
>
> We thank the reviewer again for the thoughtful feedback and would greatly appreciate any further discussion!

---

> > ### Author Rebuttal · Reviewer_ddqE · 2026-04-01
> >
> > Thank you for your reply, part of my concerns has been resolved. However, there still be some points need further clarification so that now I'm not fully convinced to raise my score.
> >
> > 1. The authors mention that removing momentum prevents variance from vanishing. However, in stochastic optimization, the noise can also be handled by using decaying learning rate. So, under decaying learning rate scheduling, does DREAM still have a better convergence result that other baselines cannot achieve? Could you please provide some theoretical results about this?
> >
> > 2. Although the authors compared it with other FMOL papers, in the broader FL field, getting rid of the bounded gradient assumption is already a standard requirement. The authors still rely on BG , which is somewhat inconsistent with their claimed theoretical contributions. If DREAM's design goal is to handle instability, then it should exhibit robustness under more stringent conditions. Moreover, some existing unified work, e.g., SPARKLE: A Unified Single-Loop Primal-Dual Framework for Decentralized Bilevel Optimization, can already remove the BG assumption even in more complex settings.
> >
> > 3. The introduction of matrix $A$ increases the cost of hyperparameter tuning. It is unclear how can $A$ be automatically learned or determined in large-scale or dynamic tasks? Without an automated determination mechanism, $A$ may simply be obtained in experiments through posterior tuning, raising issues about its generalizability.
> >
> > 4. The real challenge of FMOL lies in tasks with strong conflicts. Even a 2%-6% performance improvement on simple tasks like Multi-MNIST is insufficient to demonstrate the superiority of DREAM's complex design when handling large-scale heterogeneous data in the real world. Furthermore, the introduction of multiple components would incorporate additional computation and communication cost,  it lacks the report of the communication overhead and wall-clock time analysis.

---

> > > ### Author Response · Authors · 2026-04-04
> > >
> > > Thank you for the thoughtful follow-up. We agree these are the key concerns and address them below.
> > >
> > > Q1
> > >
> > > We agree that in standard stochastic optimization, decaying LRs reduce noise via bounds like $O(1/(\eta T))+O(\eta\sigma^2)$. Our main results already use decaying schedules (e.g., Cor. 4.9/4.12/4.14 set $\eta_g,\eta_l$ to decay with $T$). However, in mini-batch FMOL, decaying LR alone is still insufficient.
> > >
> > > The key difference is that stochastic MOO/FMOL often involves an error term **not directly scaled by LR** in the same way. One intermediate bound in our non-convex analysis (for simplicity, with $A=I$; Page 26) is:
> > > $$
> > > \mathbb{E}||\nabla F^t\lambda_0^t||^2\le6\sum_s\mathbb{E}||\nabla f_s^t-\bar{\Delta}^{t+1}_s||^2-\frac{8}{\eta_g}\mathbb{E}[F^{t+1}-F^t]\lambda
> > > +...
> > > $$
> > > The first term is the input error entering the nonlinear MOO solver, and its coefficient is a **constant**, not $\eta$. Without extra mechanisms, this term typically only reduces to a non-vanishing floor such as $O(\sigma^2/B)$ in standard stochastic MOO/FMOL analyses (unless one uses mega-batch sampling, e.g., SMG).
> > >
> > > This is why related works also go beyond decaying LR: e.g., double sampling in MoDo/SDMGrad, VR in MoCo+. In DREAM, momentum reduces this error to $O(\gamma\sigma^2/B)$. Thus, if momentum is removed ($\gamma=1$), the corresponding variance term in Theorem 4.8 remains non-vanishing in the mini-batch regime; with $\gamma=O(\sqrt{N/T})$, it becomes $O(1/\sqrt{NT})$, enabling a vanishing bound.
> > >
> > > Therefore, even under the same decaying LR schedule, DREAM still has a theoretical advantage over FSMGDA-style baselines: decaying LR alone does not control the input bias, while momentum does.
> > >
> > > Q2
> > >
> > > We appreciate the reviewer’s pointer to SPARKLE, where BG can be removed under different frameworks. We agree this is a limitation and will soften our claim accordingly. Our point is not that BG is fundamentally unavoidable, but that stochastic MOO/FMOL has an additional obstacle beyond standard single-objective FL.
> > >
> > > The same inequality above reflects this mismatch: the target is the weighted Pareto-stationarity measure $||\nabla F^t\lambda_{0}^t\|^2$, whereas the RHS error term $\sum_s||\nabla f_s^t-\bar{\Delta}_s^{t+1}||^2$ is controlled through unweighted per-task gradients, e.g., $\sum_s||\nabla f_s^t||^2$ ((17), Page 22). Under our current proof, removing BG would require a relation like $$\sum_s||\nabla f_s^t||^2
> > > \le C||\nabla F^t\lambda_0^t||^2,$$ which generally does not hold in MOO, since simplex weights may down-weight tasks whose individual gradients remain large. So the issue is not simply failure to import a BG-free FL technique; rather, there is a genuine **MOO-specific weighted/unweighted mismatch** in the recursion. This is also why BG-type assumptions remain common in stochastic MOO analyses.
> > >
> > > That said, BG-free FMOL theory is clearly important. Our contribution should be understood relative to prior FMOL works, and we will explicitly discuss BG as a current limitation. A plausible next step is to replace BG with assumptions implying effective boundedness along the trajectory, such as bounded iterates or compact level sets.
> > >
> > > Q3
> > >
> > > The core gains of DREAM do not rely on tuning $A$. In practice, $A=I$ is the default, and already outperforms FMOL baselines, so $A$ should be viewed as an **optional way to encode prior structure**, not a required hyperparameter. We do not tune a dense $S\times S$ matrix; instead, one typically uses simple structured forms (e.g. diagonal scaling, or block-diagonal grouping), summarized in Table A (https://imgur.com/a/idTREdd). Thus, we do not depend on posterior tuning.
> > >
> > > We agree that automatically adapting $A$ online would be valuable future work, but that is beyond the current scope.
> > >
> > > Q4
> > >
> > > Our empirical evidence is not limited to Multi-MNIST: the paper also includes CIFAR-100 and CelebA (40 attributes) under heterogeneous data, which are substantially larger and more realistic. DREAM improves over FSMGDA on CIFAR-100 by about +2.4%/+3.5%, and on CelebA by +7.3% avg. accuracy.
> > >
> > > Regarding overhead: DREAM's extra operations (vector add/scale for control variates and momentum) are $O(p)$ arithmetic per task, negligible compared to backpropagation's $O(pB)$ cost. The measured per-round wall-clock times showed no significant difference in ablation experiments. DREAM keeps the same communication rounds as FSMGDA. Its extra communication is the broadcast of $S$ task-wise aggregated vectors $\{\bar\Delta_s\}$ from the server, while the client-side overhead is only lightweight $O(Sp)$ vector operations. A concise comparison is in Table B (https://imgur.com/a/idTREdd).
> > >
> > > Therefore, DREAM achieves improvements on large-scale benchmarks without incurring additional significant communication or computational overhead.
> > >
> > > Thank you again for the constructive feedback. We hope these clarifications help address the remaining concerns, and we sincerely appreciate the reviewer’s consideration in the final evaluation.

---

### Official Review · Reviewer_ss2d · 2026-03-09

**Soundness:** 2
**Presentation:** 2
**Significance:** 2
**Originality:** 3
**Overall Recommendation:** 4
**Confidence:** 1

**Summary:**

This paper investigates the federated multi-objective learning (FMOL) problem. It develops a unified framework named DREAM to address two coupled error in this problem. It provides comprehensive theoretical analyses for non-convex and strongly convex loss functions. Furthermore, empirical evaluations are conducted to demonstrate the effectiveness of the proposed method.

**Compliance With Llm Reviewing Policy:**

Affirmed.

**Final Justification:**

The authors have fully addressed my concerns during the rebuttal, so I will maintain my score of 4. However, because this submission falls outside my primary area of expertise, I set my confidence score as 1.

**Key Questions For Authors:**

Refer to __Weaknesses__.

**Limitations:**

yes.

**Strengths And Weaknesses:**

__Strengths__

1. The paper introduces DREAM, a unified framework for Federated Multi-Objective Learning (FMOL). The approach of addressing two coupled errors within a single framework is well-motivated.
2. The authors provide a detailed theoretical analysis.
3. The effectiveness of the proposed DREAM framework is supported by experimental results across several tasks.



__Weaknesses__



1. The authors claim that FMOL exacerbates the optimization complexity due to "intra-task client drift."  I would appreciate a more rigorous explanation or a toy example showing how multiple objectives specifically intensify this drift.
2. In standard optimization literature, the convergence rate for strongly convex functions is typically superior compared to non-convex settings. However, in this paper, the derived rate for the strongly convex case seems weaker or less favorable than the non-convex one. I hope the authors could clarify this.
3. Although the paper provides convergence guarantees for non-convex and strongly convex settings, the general convex case is notably absent. Could the authors clarify whether the DREAM framework can be extended to this class?

---

> ### Author Rebuttal · Authors · 2026-03-29
>
> We are grateful for the reviewer's positive assessment and the opportunity to clarify the theoretical intuitions. We address each question below.
>
> **Q1: Why multiple objectives make drift more harmful in FMOL**
>
> We respectfully clarify that multiple objectives don't create additional raw heterogeneity by themselves. Rather, in FMOL, the **non-linear multi-objective aggregation step amplifies the effect of existing client drift**, making the optimization significantly more sensitive than in single-objective FL.
>
> **Why non-linearity is the key.** In single-objective FL, the server linearly averages local updates. Under unbiased stochasticity, this preserves the correct descent direction in expectation. In contrast, MOO methods such as MGDA determine task weights through a non-linear optimization subproblem. Because the optimal weights depend non-linearly on stochastic inputs, even unbiased gradient estimates introduce aggregation bias:
>
> $$\mathbb{E}[\sum_s\lambda_s^{g,\*}g_s(x)]\neq\sum_s\lambda_s^\*\nabla f_s(x).$$
>
> In FMOL, the inputs to the server are not merely noisy; they are additionally distorted by local client drift caused by heterogeneous local training. The non-linear solver is highly sensitive to such shifts, and the resulting error feeds back: a distorted descent direction worsens the next round's client drift, which further corrupts future gradient estimates (formalized in Section 4 and Appendix C).
>
> - A simple 2-task **toy example**. Suppose the true global gradients are $\nabla f_1=(1,0.1)$ and $\nabla f_2=(-1,0.1)$. The ideal optimal MOO direction is $(0,0.1)$ with equal weights. Now suppose client drift mildly corrupts the aggregated updates to $\bar\Delta_1=(1,-0.05)$ and $\bar\Delta_2=(-1,0.2)$.
>   - **Linear Scalarization** with fixed equal weights yields $d_{lin}=(0,0.075)$. Checking: $\langle d_{lin},\nabla f_1\rangle=\langle d_{lin},\nabla f_2\rangle=0.0075>0$. It remains a valid common descent direction.
>   - **MGDA** minimize $||\lambda\bar\Delta_1+(1-\lambda)\bar\Delta_2||^2$.  The conflicting $x$-components dominate the norm, so the solver "over-fits" to cancel this axis, yielding $\lambda_1\approx0.505$ and $\lambda_2\approx0.495$. This produces the direction $d_{MOO}\approx(0.01,0.074)$. Checking against Task 2: $\langle d_{MOO},\nabla f_2\rangle=-0.0026<0$. The non-linear solver's sensitivity to the drift has flipped the update into an **ascent** direction for Task 2, violating Pareto improvement.
>
> This example illustrates that under the same mild drift, linear aggregation remains safe while the non-linear MOO solver catastrophically amplifies the perturbation. DREAM breaks this feedback loop by explicitly smoothing task-wise updates (via control variates and momentum) before they enter the sensitive non-linear solver.
>
> **Q2: Clarification on the SC vs. NC convergence rates**
>
> The apparent discrepancy mainly comes from the fact that the two results are stated in **different evaluation metrics**, which are not directly comparable.
>
> The $O(1/\sqrt{NT})$ rate in the NC setting bounds the **gradient norm** (Pareto stationarity, i.e., $\min_{0\le r\le T-1}\mathbb{E}||\nabla\mathbf{F}(x^r)\mathbf{A}^\top\lambda_{0}^\*(x^r)||^2$). In contrast, the $\tilde{O}(1/\sqrt{T})$ rate in the SC setting measures a much stricter **weighted sub-optimality gap** ($\min_{0\le r\le T-1}\mathbb{E}[\mathbf{F}(x^r)-\mathbf{F}(x^\*)] \mathbf{A}^\top{\lambda}^{r+1}_\rho$). These are standard but fundamentally different criteria in MOO, making the two rates not directly comparable. Moreover, the SC analysis in Theorem 4.13 does contains a geometric contraction term $O(\exp(-\eta_g\mu T/4))$, meaning the algorithm converges exponentially fast until it hits a steady-state noise floor.
>
> The bottleneck limiting the SC rate to $\tilde{O}(1/\sqrt{T})$ is the persistent stochastic noise. Existing works like FSMGDA claim a faster $\tilde{O}(1/T)$ rate in SC FMOL under a much stronger $(\alpha,\beta)$-Lipschitz continuity condition on stochastic gradients, which effectively requires mega-batch sampling to control variance. DREAM avoids such assumptions to reflect practical, constant-batch settings. Under comparable variance-decay mechanisms, the noise floor vanishes and DREAM naturally recovers the $\tilde{O}(1/T)$ rate.
>
> **Q3: Extension to the general convex case.**
>
> Yes, the DREAM framework can be extended to the general convex setting ($\mu=0$), we omitted it because doing so requires introducing additional boundedness assumptions. In our strongly convex analysis, $\mu>0$ provides a crucial negative term to control the accumulation of the distance term $\mathbb{E}||x^t-x^*||^2$ in the recurrence. For general convex objectives, this control vanishes, and convergence can only be established by assuming a **bounded feasible set** to cap the iterates, as adopted in MoCo, CR-MOGM, PSMGD. We prioritize presenting the unconstrained NC and SC analyses to maintain a cleaner and more practical theoretical foundation.

---

> > ### Author Rebuttal · Reviewer_ss2d · 2026-04-01
> >
> > The authors have solved all my concerns. Hence, I will maintain my positive score for this work.

---

> > > ### Author Response · Authors · 2026-04-02
> > >
> > > We are grateful that our rebuttal has addressed your concerns, and we sincerely appreciate your positive assessment of our work. It means a great deal to us.

---

### Official Review · Reviewer_57ab · 2026-03-14

**Soundness:** 3
**Presentation:** 2
**Significance:** 3
**Originality:** 3
**Overall Recommendation:** 4
**Confidence:** 3

**Summary:**

In this paper, the authors propose a general framework for federated multi-objective learning (FMOL) that accommodates fine-grained manipulation of task-specific gradients. The framework subsumes several multi-objective descent direction computations and allows the incorporation of prior knowledge on task priority and interactions. The paper introduces a momentum-inspired error correction technique for controlling errors due to gradient aggregation across different clients and tasks, and provides a theoretical convergence analysis establishing its convergence. The paper also provides some empirical results to verify the practical applicability of the proposed method.

**Compliance With Llm Reviewing Policy:**

Affirmed.

**Final Justification:**

The authors have sufficiently addressed my concerns, so I maintain my positive score.

**Key Questions For Authors:**

* Could the authors provide ablation studies isolating the effects of the global task-specific gradient aggregation correction and the per-task client-side gradient correction to clarify their individual contributions?
* Can the authors provide empirical evidence demonstrating the practical benefit of modeling task interactions through the proposed task-correction matrix?
* Could the authors include additional real-world benchmarks / federated multi-objective optimization baselines to strengthen the empirical evaluation?

**Strengths And Weaknesses:**

Strengths:
* This paper provides a deep analysis of how to correct the errors introduced due to gradient aggregation across clients and tasks in the FOML setting, which is an important area of study for the machine learning community.
* The paper is mostly self-contained, and the contributions are outlined clearly.
* The paper provides theoretical convergence guarantees for the proposed method, along with empirical verification of the applicability of the proposed method in real-world FOML settings.

Weaknesses:
* Some ablation of the components of the proposed algorithm, such as correction for global task-specific gradient aggregation and correction for per-task client-side gradient, may be needed to show the effect of each type of error on the performance of FOML.
* Some empirical results for task interaction modeling seem needed to show the practical usefulness of the proposed framework in that setting.
* The numerical experiments section can benefit from more real-world benchmarks with more multi-objective optimization baselines applied in the federated learning setting to bolster the importance of the proposed method in the practical FOML setting.

---

> ### Author Rebuttal · Authors · 2026-03-29
>
> We sincerely thank the reviewer for the constructive feedback. We agree that clearer ablations, stronger evidence for task-interaction modeling, and broader empirical comparisons would better demonstrate the practical value of the framework. In response, we conducted additional experiments during the rebuttal period and summarize them below.
>
> **Q1: Ablation studies isolating the correction mechanisms.** Anonymous figures: https://imgur.com/a/K37psde
>
> We add new ablations on Multi-MNIST to isolate the two client-side mechanisms in DREAM: drift correction and momentum smoothing. This dataset is suitable here because, although the two tasks are relatively similar, the client data are highly heterogeneous, making the effect of drift particularly visible. All runs are repeated across multiple random seeds.
>
> - **Removing control variate**. In this case, DREAM reduces to a baseline without explicit drift correction (denoted “FSMGDA” in the figure). The result is a strong directional bias toward **Task 1**, and the average accuracy drops by about **6%**. Under heterogeneous local training, the aggregated task updates are no longer balanced enough for the server-side MOO solver to identify a good common descent direction (also consistent with Fig.2.c, Page 8). Restoring the drift correction significantly improves task balance and yields a clearly better Pareto trade-off across both tasks.
> - **Removing momentum smoothing** by setting $\gamma=1$. This keeps the control variate but disables the smoothing of noisy updates before they enter the server’s non-linear weight solver. Empirically, without momentum, the average accuracy drops by about **2%**. Moreover, the descent direction norm $\|d\|$ becomes larger and both $\|d\|$ and the dynamically solved task weights fluctuate much more strongly across rounds. This indicates that the server-side MOO subproblem becomes more sensitive to stochastic noise, leading to less stable convergence. With momentum enabled, these trajectories are much smoother and convergence is more stable.
>
> Overall, the ablations show that the two mechanisms play **complementary roles** rather than redundant ones:
>
> - the **control variate / per-task client-side correction** mainly mitigates the heterogeneous shift caused by local training;
> - the **momentum smoothing / global task-update stabilization** mainly reduces stochastic oscillation before non-linear multi-objective aggregation.
>
> This is consistent with our theoretical discussion in Section 4 and Appendix C, where the key challenge is the recursive coupling between **client drift** and **gradient estimation error** in FMOL. Together, these two client-side mechanisms improve the quality of the task-wise updates before they reach the server.
>
> **Q2: Empirical evidence for modeling task interactions via matrix A.** Anonymous figures: https://imgur.com/a/u00Rr9s
>
> We provide a new experiment on 2MNIST+FMNIST dataset to test whether the task-correction matrix $A$ can effectively encode prior task structure. In this benchmark, the two MNIST-related tasks are naturally more related to each other than to FMNIST. To encode this prior, we use
> $$
> A=[1,\beta,0; \beta,1,0; 0,0,1]
> $$
> where the off-diagonal parameter $\beta$ controls the interaction strength between the two MNIST tasks, while leaving the FMNIST task independent.
>
> Compared with the standard MGDA setting $A=I$, introducing a mild positive correlation improves the optimization outcome. In particular, $\beta=0.1$ yields a +2.95% gain in average accuracy over the baseline. Even for stronger couplings ($\beta=0.2,0.3$), we still observe consistent gains of around +0.4%. These results suggest that $A$ is not merely a theoretical generalization: it can translate meaningful structural prior knowledge into measurable empirical benefit.
>
> Intuitively, $A$ changes the geometry of the server-side aggregation problem by allowing related tasks to share directional information. A moderate interaction strength appears most effective: it alleviates unnecessary gradient conflict between related tasks, while overly strong coupling can partially blur task-specific signals. We will clarify this practical interpretation in the revision.
>
> **Q3: Additional real-world benchmarks and baselines.** Anonymous figures: https://imgur.com/a/RLVNVPV
>
> We agree this would further strengthen the paper. Due to the short rebuttal timeline, we prioritize expanding baseline comparisons on the existing real-world benchmarks rather than introducing an entirely new dataset pipeline. In particular, we augment our evaluations on the large-scale CelebA (40 attributes) benchmark with additional baselines.
>
> The current FMOL literature remains quite limited: to the best of our knowledge, FSMGDA and FedCMOO are among the very few methods that explicitly study FMOL. To broaden the comparison, we therefore adapt several strong centralized stochastic MOO baselines as supplementary references (with average accuracy ~$80\%$ for CelebA).

---

> > ### Author Rebuttal · Reviewer_57ab · 2026-04-04
> >
> > Thank you for the clarifications and the detailed responses; most of my concerns were addressed.  I encourage the authors to incorporate these clarifications into the final version to better illustrate the proposed method, especially the necessity of each component. I will maintain my positive score.

---

> > > ### Author Response · Authors · 2026-04-04
> > >
> > > Thank you very much for your thoughtful feedback and encouraging support! We are glad that our clarifications were helpful, and we sincerely appreciate your positive assessment.

---

### Decision · Program_Chairs · 2026-04-30

**Decision:**

Accept (regular)

**Comment:**

This paper proposes DREAM, a unified framework for federated multi-objective learning that jointly addresses client drift and inter-task aggregation bias. The theoretical analysis is rigorous, covering both non-convex and strongly convex settings, and experiments on Multi-MNIST, CIFAR-100, and CelebA demonstrate consistent improvements over baselines.

Strengths include a novel error-coupling insight, complementary client-side corrections, and a flexible server-side task-correction matrix. Weaknesses—such as limited ablation and reliance on bounded gradient assumptions—were largely addressed in rebuttal with new results and clarifications.

Given  solid technical contribution, and effective author responses, this paper merits acceptance.